# The Geometric Mechanics of Contrastive Representation Learning: Alignment Potentials, Entropic Dispersion, and Cross-modal Divergence

**Yichao Cai** [1]  **Zhen Zhang** [1 2]  **Yuhang Liu** [1 2]  **Javen Qinfeng Shi** [1 2]

## Abstract

While InfoNCE underlies modern contrastive learning, its geometric mechanisms remain under-characterized beyond the canonical alignment–uniformity decomposition. We develop a measure-theoretic framework in which representation measures evolve on a fixed embedding manifold. In the large-batch limit, we prove value and gradient consistency, linking the stochastic objective to explicit deterministic energy landscapes and revealing a geometric bifurcation between unimodal and symmetric multimodal regimes. In the unimodal case, the intrinsic energy is strictly convex and admits a unique Gibbs equilibrium, showing that entropy acts as a tie-breaker within the aligned basin. In the multimodal case, the intrinsic geometry becomes cross-coupled and contains a persistent negative symmetric divergence term: each modality's marginal reshapes the effective landscape of the other, allowing strong pairwise alignment to coexist with a persistent modality gap. Controlled synthetic experiments and analyses of pretrained CLIP representations support these predictions. Overall, our results shift the analytical lens from pointwise discrimination to population geometry, showing that pairwise alignment alone is insufficient to control cross-modal marginal structure. [1]

## 1. Introduction

InfoNCE-based contrastive learning is a central objective in self-supervised and multimodal representation learning (Oord et al., 2018; Radford et al., 2021; Jia et al., 2021),

yet its representational geometry remains poorly understood. In multimodal systems such as CLIP, training can achieve strong *pairwise* alignment while the modality *marginals* remain separated, an empirically observed phenomenon known as the modality gap (Liang et al., 2022; Levi & Gilboa, 2025), for which a mechanistic explanation is still lacking. Even in the unimodal setting, it remains unclear what population-level energy InfoNCE induces and how its geometry should be understood beyond the alignment–uniformity view (Wang & Isola, 2020). This raises a key geometric question for representation learning: why does matching positive pairs fail to match the induced population distributions? More broadly, what population energies do unimodal and symmetric multimodal InfoNCE induce, and what equilibria does their geometry favor?

Existing theory still lacks a first-principles account of these questions. Density-ratio analyses characterize the optimal critic as pointwise mutual information (Gutmann & Hyvärinen, 2010; Oord et al., 2018), but do not directly describe the deterministic descent directions induced by the softmax denominator or how these directions reshape the induced representations. The alignment–uniformity view (Wang & Isola, 2020) clarifies important asymptotic trade-offs, but leaves open what population structure the InfoNCE objective itself favors. Identifiability results offer recovery guarantees under explicit generative assumptions (Zimmermann et al., 2021; Liu et al., 2026), addressing what is learnable in principle rather than the objective-induced geometry of the learned marginals. This lack of a geometric account is particularly consequential in multimodal settings, where the symmetric InfoNCE objective is formally an average of two directional terms, yet its induced geometry is not merely the simple average of their individual geometries.

We therefore adopt a measure-theoretic viewpoint that makes the population geometry of InfoNCE explicit (see Fig. 1 for an illustration). In the canonical feature normalization setting, we regard the embedding space $\mathcal{Z}$ as a fixed compact manifold equipped with a volume measure $\mu$. Encoders push data distributions forward onto $\mathcal{Z}$, thereby inducing both representation laws and positive-pair laws directly in the embedding space. Under the exponential similarity kernel, the InfoNCE denominator converges, in the

---

[1]Australian Institute for Machine Learning, Adelaide University. [2]Responsible AI Research Centre, Australia. Correspondence to: Yichao Cai <yichao.cai@adelaide.edu.au>.

*Proceedings of the 43$^{rd}$ International Conference on Machine Learning*, Seoul, South Korea. PMLR 306, 2026. Copyright 2026 by the author(s).

[1]The project page is available at: `https://yichaocai.com/nce_geo.github.io`

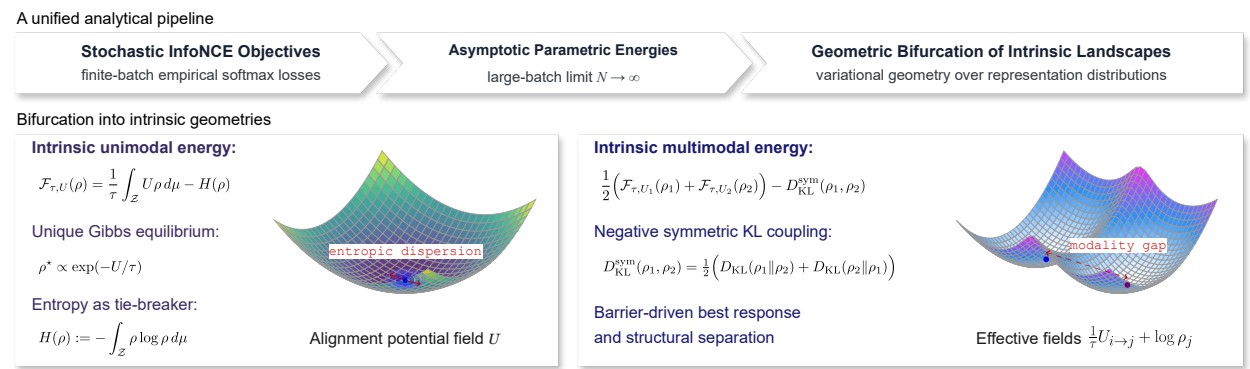

*Figure 1.* A unified analytical pipeline for contrastive learning geometry. Starting from stochastic InfoNCE losses, the large-batch limit yields deterministic parametric energies, which in turn lift to intrinsic landscapes over representation densities. At this intrinsic level, the geometry bifurcates: in the unimodal case, the landscape is strictly convex, yielding a Gibbs equilibrium; multimodally, a negative symmetric KL coupling induces barrier-driven best responses and a modality gap. The landscape visualizations are illustrative schematics.

large-batch limit, to a population *partition field*: a kernel-averaged functional of the current representation law that determines the softmax gradient. We show that stochastic InfoNCE is consistent with a closed-form deterministic energy over representation measures, both in value and in gradient. This passage from finite-batch objectives to population energies allows us to separate intrinsic geometric effects from parametrization or training artifacts.

From this intrinsic view, unimodal and symmetric multimodal contrastive learning exhibit a fundamental *geometric bifurcation*. In the unimodal regime, the population energy is strictly convex and admits a unique Gibbs equilibrium: at low temperature, mass concentrates in highly aligned regions, while entropy controls dispersion *within* those regions. This reframes "uniformity" (Wang & Isola, 2020) not as a global force opposing alignment, but as entropy-driven spreading inside the alignment basin. In contrast, symmetric multimodal InfoNCE induces a cross-coupled geometry: heterogeneous modalities generally define different conditional laws and hence different alignment fields, producing a negative symmetric divergence coupling between their marginals. Rather than driving the modalities toward a shared equilibrium, this coupling makes each modality act as a barrier in the effective landscape of the other, leading to barrier-driven co-adaptation. Exact marginal matching thus requires a knife-edge compatibility condition, and the modality gap arises generically in the analyzed regime rather than as a mere optimization failure.

These intrinsic results are not merely formal: under suitable encoder expressivity, the corresponding parametric energies inherit the same equilibrium and barrier structures up to controllable KDE mismatch, linking the population-level theory back to practical encoder training. We validate the resulting predictions through controlled synthetic experiments and measurements on pretrained CLIP representations. The results show that strong retrieval performance can coexist with

a measurable modality gap, and that weakening cross-modal compatibility systematically enlarges this gap even when category-level semantics are preserved. Together, these findings suggest that the modality gap is not merely a failure of pairwise alignment, but reflects a population-level geometric effect; reducing it therefore requires controlling cross-modal divergence beyond positive-pair matching alone.

**Contributions.** Our main contributions are: **(i)** large-batch value- and gradient-consistency results connecting stochastic InfoNCE to deterministic population energies; **(ii)** a characterization of unimodal InfoNCE as a strictly convex population energy with a unique Gibbs equilibrium, together with a reinterpretation of "uniformity" as entropy-driven dispersion within the alignment basin; **(iii)** a derivation of the multimodal bifurcation, in which a negative symmetric divergence coupling induces barrier-driven co-adaptation and can sustain a modality gap under conditional heterogeneity; and **(iv)** synthetic and real-data evidence supporting these predictions through controlled experiments and measurements on pretrained OpenCLIP models across CNN- and ViT-based backbones.

## 2. Problem Setup

We introduce the population-level objects used throughout the analysis for unimodal and symmetric multimodal InfoNCE. For ease of reference, Tab. 1 summarizes the principal notations used throughout the paper.

**Data, encoders, and induced laws.** Let $\mathcal{P}(\mathcal{S})$ denote the set of probability measures on a measurable space $\mathcal{S}$. In the unimodal setting, $r_{\mathrm{um}} \in \mathcal{P}(\mathcal{X} \times \mathcal{X})$ governs positive observational pairs $(\mathbf{x}, \tilde{\mathbf{x}})$, typically two views of the same underlying example; its two $\mathcal{X}$-marginals therefore coincide, and we denote the common marginal by $p_{\mathbf{x}}$. In the multimodal setting, $r_{\mathrm{mm}} \in \mathcal{P}(\mathcal{X} \times \mathcal{Y})$ governs positive cross-modal pairs $(\mathbf{x}, \mathbf{y})$, where $\mathcal{X}$ and $\mathcal{Y}$ are measurable sample spaces;

we write $p_{\mathbf{x}}$ and $p_{\mathbf{y}}$ for its $\mathcal{X}$- and $\mathcal{Y}$-marginals.

We consider two encoder families $\{f_\theta\}_{\theta \in \Theta}$ and $\{g_\phi\}_{\phi \in \Phi}$ with compact parameter spaces $\Theta$ and $\Phi$, mapping into a common representation space $\mathcal{Z} \subset \mathbb{R}^n$. We model $\mathcal{Z}$ as a Riemannian manifold (e.g., $\mathbb{S}^{n-1}$) equipped with volume measure $\mu$. These encoders induce the pushforward laws

$$q_\theta := (f_\theta)_\# p_{\mathbf{x}}, \qquad \pi_{\theta\theta} := (f_\theta \times f_\theta)_\# r_{\mathrm{um}},$$
$$q_\phi := (g_\phi)_\# p_{\mathbf{y}}, \qquad \pi_{\theta\phi} := (f_\theta \times g_\phi)_\# r_{\mathrm{mm}}.$$

Here, $q_\theta$ and $q_\phi$ are the encoded marginal laws, while $\pi_{\theta\theta}$ and $\pi_{\theta\phi}$ are the encoded positive-pair laws.

**InfoNCE objectives in contrastive learning.** We consider InfoNCE objectives based on the exponential kernel

$$\kappa_\tau(\mathbf{z}, \mathbf{w}) = \kappa_\tau(\mathbf{w}, \mathbf{z}) := \exp\big(s(\mathbf{z}, \mathbf{w})/\tau\big),$$

where $s$ is a similarity critic and $\tau > 0$ is the temperature. The unimodal and symmetric multimodal objectives are defined as follows.

*(i) Unimodal objective.* Given a training batch $\mathcal{B}$ composed of a positive pair $(\mathbf{x}, \tilde{\mathbf{x}}) \sim r_{\mathrm{um}}$ and $N$ negative samples $\{\mathbf{x}'_j\}_{j=1}^N \overset{\text{iid}}{\sim} p_{\mathbf{x}}$, the empirical InfoNCE loss is[2]

$$\mathcal{L}_{\mathrm{NCE}}(\theta) :=$$
$$\mathbb{E}_{\mathcal{B}}\left[ -\log \frac{\kappa_\tau(f_\theta(\mathbf{x}), f_\theta(\tilde{\mathbf{x}}))}{\kappa_\tau(f_\theta(\mathbf{x}), f_\theta(\tilde{\mathbf{x}})) + \sum_{j=1}^N \kappa_\tau(f_\theta(\mathbf{x}), f_\theta(\mathbf{x}'_j))} \right]. \tag{1}$$

*(ii) Symmetric multimodal objective.* Given a cross-modal positive pair $(\mathbf{x}, \mathbf{y}) \sim r_{\mathrm{mm}}$ and negatives $\{\mathbf{y}'_j\}_{j=1}^N \overset{\text{iid}}{\sim} p_{\mathbf{y}}$, define the directional loss

$$\mathcal{L}_{\mathrm{NCE}}^{x \to y}(\theta, \phi) := \mathbb{E}_{\mathcal{B}_{x \to y}}$$
$$\left[ -\log \frac{\kappa_\tau\big(f_\theta(\mathbf{x}), g_\phi(\mathbf{y})\big)}{\kappa_\tau\big(f_\theta(\mathbf{x}), g_\phi(\mathbf{y})\big) + \sum_{j=1}^N \kappa_\tau\big(f_\theta(\mathbf{x}), g_\phi(\mathbf{y}'_j)\big)} \right],$$

where $\mathcal{B}_{x \to y} := \{\mathbf{x}, \mathbf{y}, \mathbf{y}'_1, \ldots, \mathbf{y}'_N\}$. Define $\mathcal{L}_{\mathrm{NCE}}^{y \to x}(\phi, \theta)$ analogously by swapping modalities. The symmetric multimodal InfoNCE loss is

$$\mathcal{L}_{\mathrm{Sym}}(\theta, \phi) := \tfrac{1}{2}\big(\mathcal{L}_{\mathrm{NCE}}^{x \to y}(\theta, \phi) + \mathcal{L}_{\mathrm{NCE}}^{y \to x}(\phi, \theta)\big). \tag{2}$$

**Population partition fields and smoothed densities.** At the population level, the InfoNCE denominator induces kernel-averaged fields on $\mathcal{Z}$. We use these fields to define the corresponding smoothed representation densities. To normalize them, we impose the following constant-volume condition.

---

[2]We analyze an iid negative-sampling formulation. Although in-batch sampling introduces finite-sample dependencies, both regimes converge to the same population limit as $N \to \infty$.

**Assumption 2.1** (Constant kernel volume). The representation space $\mathcal{Z}$ is compact with $\mu(\mathcal{Z}) < \infty$. For each $\tau > 0$, there exists $V_\kappa(\tau) \in (0, \infty)$ such that

$$\int_{\mathcal{Z}} \kappa_\tau(\mathbf{z}, \mathbf{w}) \, \mathrm{d}\mu(\mathbf{w}) = V_\kappa(\tau) \qquad \text{for all } \mathbf{z} \in \mathcal{Z}.$$

*Remark* 2.1 (When Asm. 2.1 holds). A sufficient condition for Asm. 2.1 is that $\mathcal{Z}$ is a compact homogeneous Riemannian manifold, $\mu$ is its invariant volume measure, and $\kappa_\tau$ is isotropic, i.e., depends only on geodesic distance. In that case, $V_\kappa(\tau, \mathbf{z}) := \int_{\mathcal{Z}} \kappa_\tau(\mathbf{z}, \mathbf{w}) \, \mathrm{d}\mu(\mathbf{w})$ is independent of $\mathbf{z}$, so Asm. 2.1 holds exactly.

**Definition 2.1** (Population partition fields). Given representation laws $q_\theta$ and $q_\phi$, define the population *partition fields*

$$\Gamma_{\theta,\tau}(\mathbf{z}) := \int_{\mathcal{Z}} \kappa_\tau(\mathbf{z}, \mathbf{w}) \, \mathrm{d}q_\theta(\mathbf{w}),$$
$$\Gamma_{\phi,\tau}(\mathbf{z}) := \int_{\mathcal{Z}} \kappa_\tau(\mathbf{z}, \mathbf{w}) \, \mathrm{d}q_\phi(\mathbf{w}).$$

If $s$ is bounded above on $\mathcal{Z} \times \mathcal{Z}$, then these quantities are finite and well-defined.

**Definition 2.2** (Kernel-smoothed representation density). Under Asm. 2.1, define the kernel-smoothed representation densities $\tilde{\rho}_{\theta,\tau}, \tilde{\rho}_{\phi,\tau} : \mathcal{Z} \to \mathbb{R}_+$ by

$$\tilde{\rho}_{\theta,\tau}(\mathbf{z}) := \Gamma_{\theta,\tau}(\mathbf{z})/V_\kappa(\tau), \quad \tilde{\rho}_{\phi,\tau}(\mathbf{z}) := \Gamma_{\phi,\tau}(\mathbf{z})/V_\kappa(\tau).$$

Equivalently, $\tilde{\rho}_{\theta,\tau} \, \mathrm{d}\mu$, $\tilde{\rho}_{\phi,\tau} \, \mathrm{d}\mu$ are the corresponding kernel-smoothed representation measures on $\mathcal{Z}$.

*Remark* 2.2 (Finite-$\tau$ smoothing and positivity). For any $\tau > 0$, $\tilde{\rho}_{\theta,\tau}$ is a kernel smoothing of $q_\theta$ via $\Gamma_{\theta,\tau}$. In particular, it defines a $\mu$-density even when $q_\theta$ is singular with respect to $\mu$. If $\kappa_\tau(\mathbf{z}, \mathbf{w})$ is continuous and strictly positive on $\mathcal{Z} \times \mathcal{Z}$, then $\tilde{\rho}_{\theta,\tau}$ is continuous and strictly positive on $\mathcal{Z}$; the same holds for $\tilde{\rho}_{\phi,\tau}$.

Thus $\Gamma_{\theta,\tau}(\mathbf{z})$ is the population softmax normalizer associated with a query at $\mathbf{z}$, and $\tilde{\rho}_{\theta,\tau}$ is the corresponding kernel-smoothed density of $q_\theta$ at scale $\tau$. In this way, the empirical InfoNCE denominator can be viewed as a Monte Carlo estimator of $\Gamma_{\theta,\tau}$ (and symmetrically of $\Gamma_{\phi,\tau}$ in the multimodal regime), while the population dynamics are governed by the induced densities $\tilde{\rho}_{\theta,\tau}$ and $\tilde{\rho}_{\phi,\tau}$.

## 3. Unimodal InfoNCE: Convex Geometry and Gibbs Equilibrium

This section develops the unimodal baseline that underlies the later multimodal bifurcation. We first show that, in the large-batch regime, unimodal InfoNCE is governed by an explicit deterministic energy depending on $\theta$ only through the induced laws $(q_\theta, \pi_{\theta\theta})$ on $\mathcal{Z}$ and $\mathcal{Z} \times \mathcal{Z}$. Proofs for all formal statements in this section are deferred to App. C.

### 3.1. Large-Batch Deterministic Energy

We begin by identifying the population-level energy optimized by unimodal InfoNCE in the large-batch limit. The resulting closed-form parametric energy $\mathcal{J}_\tau(\theta)$ has a gradient that matches the InfoNCE gradient up to vanishing error as $N \to \infty$.

**Assumption 3.1** (Optimization regularity). Assume the following conditions hold:

(i) (Encoder regularity). For every $\mathbf{x} \in \mathcal{X}$, the map $\theta \mapsto f_\theta(\mathbf{x})$ is $C^1$ on $\Theta$, and its parameter Jacobian is uniformly bounded: $\sup_{\theta \in \Theta} \sup_{\mathbf{x} \in \mathcal{X}} \|J_\theta f_\theta(\mathbf{x})\| < \infty$.[3]

(ii) (Critic regularity). The critic $s$ is $C^1$ on an open neighborhood of $\mathcal{Z} \times \mathcal{Z}$, and its input gradient is uniformly bounded on $\mathcal{Z} \times \mathcal{Z}$: $\sup_{(\mathbf{z},\mathbf{w}) \in \mathcal{Z} \times \mathcal{Z}} \|\nabla s(\mathbf{z}, \mathbf{w})\| < \infty$.

*Remark* 3.1 (Optimization stability). Asm. 3.1 ensures that InfoNCE gradients remain uniformly controlled over $\Theta$, which is needed to justify exchanging the large-batch limit with differentiation. Further discussion and the formal uniform bounds are given in App. A.1 and Lem. B.1.

For fixed $\tau$, the empirical softmax denominator concentrates around the population partition field $\Gamma_{\theta,\tau}$ from Def. 2.1. This yields a closed-form population energy with both value and gradient consistency in the large-batch limit.

**Definition 3.1** (Alignment potential field). Let $\pi_{\theta\theta} \in \mathcal{P}(\mathcal{Z} \times \mathcal{Z})$ be the encoded positive-pair law with first marginal $q_\theta \in \mathcal{P}(\mathcal{Z})$. By disintegration, there exists a measurable family of probability measures $\{\nu_{\theta,\mathbf{z}}\}_{\mathbf{z} \in \mathcal{Z}} \subset \mathcal{P}(\mathcal{Z})$ such that $\mathrm{d}\pi_{\theta\theta}(\mathbf{z}, \mathbf{w}) = \mathrm{d}q_\theta(\mathbf{z})\,\mathrm{d}\nu_{\theta,\mathbf{z}}(\mathbf{w})$ as measures on $\mathcal{Z} \times \mathcal{Z}$. We define the alignment potential field by

$$U_\theta(\mathbf{z}) := -\int_{\mathcal{Z}} s(\mathbf{z}, \mathbf{w})\,\mathrm{d}\nu_{\theta,\mathbf{z}}(\mathbf{w}), \qquad q_\theta\text{-a.e. } \mathbf{z}. \quad (3)$$

**Definition 3.2** (Unimodal parametric energy). Fix $\tau > 0$. Let $\tilde{\rho}_{\theta,\tau}$ be the kernel-smoothed $\mu$-density from Def. 2.2, and let $U_\theta$ be the alignment potential field from Def. 3.1. Define the unimodal parametric energy $\mathcal{J}_\tau : \Theta \to \mathbb{R}$ by

$$\mathcal{J}_\tau(\theta) := \frac{1}{\tau} \int_{\mathcal{Z}} U_\theta(\mathbf{z})\,\mathrm{d}q_\theta(\mathbf{z}) - H_\times\big(q_\theta, \tilde{\rho}_{\theta,\tau}\big), \quad (4)$$

where $H_\times(q, \tilde{\rho}) = -\int_{\mathcal{Z}} \log \tilde{\rho}(\mathbf{z})\,\mathrm{d}q(\mathbf{z})$ denotes the cross-entropy of $q \in \mathcal{P}(\mathcal{Z})$ against a strictly positive density $\tilde{\rho}$.

**Theorem 3.1** (Large-batch unimodal dynamics). *Consider the unimodal InfoNCE objective $\mathcal{L}_{\mathrm{NCE}}(\theta)$ in Eq. (1) and the parametric energy $\mathcal{J}_\tau(\theta)$ in Def. 3.2. Assume Asms. 2.1 and 3.1. Then for any fixed $\tau > 0$ and $\theta \in \Theta$, as $N \to \infty$,*

$$\big|\mathcal{L}_{\mathrm{NCE}}(\theta) - \mathcal{J}_\tau(\theta) - \log\big(NV_\kappa(\tau)\big)\big| \to 0, \quad (5)$$

$$\text{and } \big\|\nabla_\theta \mathcal{L}_{\mathrm{NCE}}(\theta) - \nabla_\theta \mathcal{J}_\tau(\theta)\big\| \to 0. \quad (6)$$

---

[3]Throughout this paper, $\|\cdot\|$ denotes the $\ell_2$ norm for vectors unless otherwise specified.

Thm. 3.1 shows that large-batch unimodal InfoNCE follows gradient descent on the deterministic energy $\mathcal{J}_\tau$ up to the additive constant $\log\big(NV_\kappa(\tau)\big)$ and vanishing error. Thus, the population dynamics are governed by the partition field $\Gamma_{\theta,\tau}$, or equivalently by the smoothed density $\tilde{\rho}_{\theta,\tau}$.

### 3.2. Intrinsic Free Energy and Gibbs Equilibrium

The parametric energy $\mathcal{J}_\tau(\theta)$ is generally nonconvex in $\theta$, since the map $\theta \mapsto (q_\theta, \pi_{\theta\theta})$ is implicit. To isolate the intrinsic geometry, we fix a potential field $U$ on $\mathcal{Z}$ and optimize directly over distributions on $\mathcal{Z}$. This yields a strictly convex free-energy functional with a unique Gibbs equilibrium. In the low-temperature regime, $\mathcal{J}_\tau$ approaches this intrinsic landscape up to a controllable KDE mismatch.

**Definition 3.3** (Intrinsic unimodal energy). Fix $\tau > 0$ and a Borel measurable potential $U : \mathcal{Z} \to \mathbb{R}$. Let $\mathcal{P}_\mu(\mathcal{Z})$ denote the set of $\mu$-densities on $\mathcal{Z}$. We define the intrinsic unimodal energy $\mathcal{F}_{\tau,U} : \mathcal{P}_\mu(\mathcal{Z}) \to (-\infty, +\infty]$ by

$$\mathcal{F}_{\tau,U}(\rho) := \frac{1}{\tau} \int_{\mathcal{Z}} U(\mathbf{z})\,\rho(\mathbf{z})\,\mathrm{d}\mu(\mathbf{z}) - H(\rho), \quad (7)$$

where $H(\rho) := -\int_{\mathcal{Z}} \rho \log \rho\,\mathrm{d}\mu$[4] is the differential entropy of $\mu$-density $\rho$, with the convention that $0 \log 0 := 0$ and $\mathcal{F}_{\tau,U}(\rho) = +\infty$ if either integral is $+\infty$.

**Proposition 3.1** (Unique Gibbs equilibrium). *Assume $\mathcal{Z}$ is compact with $\mu(\mathcal{Z}) < \infty$. Fix $\tau > 0$ and let $U : \mathcal{Z} \to \mathbb{R}$ be Borel measurable and bounded below $\mu$-a.e. Then, $\mathcal{F}_{\tau,U}$ is strictly convex on $\mathcal{P}_\mu(\mathcal{Z})$ with a unique minimizer*

$$\rho^*(\mathbf{z}) = \frac{\exp(-U(\mathbf{z})/\tau)}{Z_\tau}, \quad Z_\tau := \int_{\mathcal{Z}} \exp\left(\frac{-U(\mathbf{z})}{\tau}\right)\mathrm{d}\mu(\mathbf{z}).$$

**Proposition 3.2** (Low-temperature concentration). *In the setting of Prop. 3.1, fix $\sigma > 0$ and define*

$$\mathcal{W}^\sigma := \big\{\mathbf{z} \in \mathcal{Z} : U(\mathbf{z}) \leq \operatorname{ess\,inf}_\mu U + \sigma\big\}.$$

*Then as $\tau \to 0^+$, the Gibbs measure $\rho^*(\mathbf{z})\,\mathrm{d}\mu(\mathbf{z})$ concentrates on $\mathcal{W}^\sigma$ in the sense that, for every open $\mathcal{O} \supset \mathcal{W}^\sigma$, $\rho^*\mu(\mathcal{Z} \setminus \mathcal{O}) \to 0$.*

In the unimodal regime, the intrinsic landscape admits a unique Gibbs equilibrium for each $\tau$, and as $\tau \to 0^+$, the equilibrium concentrates on near-minimizers of the alignment potential $U$. Prop. 3.2 formalizes this low-temperature concentration, while Fig. 2a illustrates it by tracking the mass inside a fixed near-minimizer region as $\tau$ decreases.

**Connecting back to $\mathcal{J}_\tau$: sharp kernels control KDE bias.** To connect the intrinsic geometry back to the parametric energy, we consider a sharp-kernel regime in which the kernel-smoothed density $\tilde{\rho}_{\theta,\tau}$ approximates the induced representation density $\rho_\theta$.

---

[4]We sometimes write $\int_{\mathcal{Z}} h\mathrm{d}\mu$ for $\int_{\mathcal{Z}} h(\mathbf{z})\mathrm{d}\mu(\mathbf{z})$ for brevity.

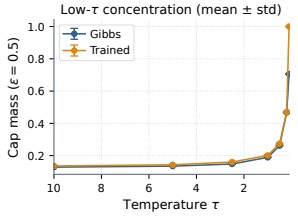
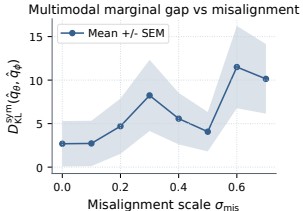

*(a)* Unimodal concentration.     *(b)* Multimodal marginal gap.

*Figure 2.* Numerical illustration of the geometric bifurcation. (a) In the unimodal intrinsic problem, the energy is Gibbs-like and concentrates toward low-potential regions as $\tau \downarrow$. (b) In the multimodal setting, cross-modal coupling induces persistent marginal separation, whose magnitude grows with latent misalignment.

**Assumption 3.2** (Sharp diagonal peak). Assume Asm. 2.1. Let $d_{\mathcal{Z}}(\cdot, \cdot)$ denote the geodesic distance on $\mathcal{Z}$. There exist constants $r > 0$ and $0 < m_1 \leq m_2 < \infty$ such that for all $\mathbf{z} \in \mathcal{Z}$ and all $\mathbf{w} \in \{\mathbf{w} \in \mathcal{Z} : d_{\mathcal{Z}}(\mathbf{z}, \mathbf{w}) < r\}$,

$$-m_2 d_{\mathcal{Z}}(\mathbf{z}, \mathbf{w})^2 \leq s(\mathbf{z}, \mathbf{w}) - s(\mathbf{z}, \mathbf{z}) \leq -m_1 d_{\mathcal{Z}}(\mathbf{z}, \mathbf{w})^2, \tag{8}$$

and the similarity peaks strictly at the identity such that $\arg\max_{\mathbf{w} \in \mathcal{Z}} s(\mathbf{z}, \mathbf{w}) = \mathbf{z}$ for each $\mathbf{z} \in \mathcal{Z}$.

*Remark* 3.2. Geometrically, Asm. 3.2 forces the normalized kernel mass to concentrate within an $O(\sqrt{\tau})$ geodesic neighborhood of the diagonal; see App. A.2 for discussion. As a result, $\tilde{\rho}_{\theta,\tau}$ acts as a local smoothing of $\rho_\theta$, which is quantified in the low-temperature regime by Lem. B.3.

**Theorem 3.2** (Low-temperature unimodal energy consistency). *Assume Asms. 2.1, 3.1 and 3.2. Fix $\theta \in \Theta$ and assume $q_\theta \ll \mu$ with a strictly positive, continuous density $\rho_\theta$, i.e., $\inf_{\mathbf{z} \in \mathcal{Z}} \rho_\theta(\mathbf{z}) \geq \underline{\rho}_\theta > 0$.[5] Define the KDE mismatch $\varepsilon_{\mathrm{kde}}^{(\theta)}(\tau) := \|\tilde{\rho}_{\theta,\tau} - \rho_\theta\|_\infty$. Then there exists $\tau_0(\theta) > 0$ such that*

$$\left|\mathcal{J}_\tau(\theta) - \mathcal{F}_{\tau, U_\theta}(\rho_\theta)\right| \leq 2\varepsilon_{\mathrm{kde}}^{(\theta)}(\tau)/\underline{\rho}_\theta, \quad \forall 0 < \tau \leq \tau_0(\theta),$$

*and in particular, as $\tau \to 0^+$, $\left|\mathcal{J}_\tau(\theta) - \mathcal{F}_{\tau, U_\theta}(\rho_\theta)\right| \to 0$.*

**Uniformity as a tie-breaker.** Thm. 3.2 bridges the intrinsic analysis back to training: in the sharp-kernel regime, the parametric energy $\mathcal{J}_\tau(\theta)$ is close to the intrinsic energy $\mathcal{F}_{\tau, U_\theta}(\rho_\theta)$ up to the KDE mismatch between $\tilde{\rho}_{\theta,\tau}$ and $\rho_\theta$. If the encoder family is expressive enough to realize the relevant intrinsic Gibbs equilibrium at the best attainable potential level, then minimizers of $\mathcal{J}_\tau$ inherit the same ground-state concentration behavior. Within an aligned basin where $U_\theta$ is already near-minimized, the remaining degree of freedom is how probability mass is distributed, and the entropy term in Eq. (7) selects the most internally

---

[5]See App. A.3 for a detailed discussion on the strict positivity assumption and the role of density floors.

dispersed near-optimal configuration. This is the sense in which "uniformity" emerges. Cor. B.1 formalizes the expressivity condition and the resulting parametric inheritance; see App. B.3 for extended formal statements.

> ☞ **Punchline 1** (Unimodal InfoNCE geometry). The defining feature of unimodal contrastive learning is structural cohesion. Because the modality evaluates its cross-entropy against its *own* smoothed density field, the intrinsic energy is strictly convex. Alignment determines the ground-state basin, while entropy selects a unique, internally dispersed equilibrium within it.

# 4. Symmetric Multimodal InfoNCE: Divergence-Coupled Geometry

We analyze the symmetric multimodal InfoNCE objective used in CLIP-style training (Eq. (2)). Unlike the unimodal setting, the geometry is *cross-coupled*: each modality induces the effective field seen by the other. This coupling yields a qualitatively different geometry and can sustain a population-level modality gap. Proofs for all formal statements in this section are deferred to App. D.

## 4.1. Cross-Coupled Multimodal Deterministic Energy

We now identify the deterministic population energy optimized by symmetric multimodal InfoNCE in the large-batch limit. As in the unimodal case, the stochastic objective converges at both value and gradient level to a closed-form parametric energy. The key difference is cross-coupling: each modality evaluates its cross-entropy against the other modality's kernel-smoothed density field.

**Assumption 4.1** (Multimodal optimization regularity). In multimodal contrastive learning, assume Asm. 3.1 holds for both encoder families $\{f_\theta\}_{\theta \in \Theta}$ and $\{g_\phi\}_{\phi \in \Phi}$, uniformly over $\Theta \times \Phi$.

*Remark* 4.1. Under Asm. 4.1, the symmetric InfoNCE objective has a uniformly bounded joint gradient over $\Theta \times \Phi$, which is needed to exchange the large-batch limit with differentiation in the multimodal consistency proof. See Lem. B.2.

**Definition 4.1** (Directional alignment potential fields). Let $\pi_{\theta\phi}$ denote the encoded positive-pair law on $\mathcal{Z} \times \mathcal{Z}$ with marginals $q_\theta$ and $q_\phi$. By disintegration, there exist measurable families of probability measures $\{\nu_{\theta\phi,\mathbf{z}}\}_{\mathbf{z} \in \mathcal{Z}}$ and $\{\nu_{\phi\theta,\mathbf{w}}\}_{\mathbf{w} \in \mathcal{Z}}$ such that $d\pi_{\theta\phi}(\mathbf{z}, \mathbf{w}) = dq_\theta(\mathbf{z}) \, d\nu_{\theta\phi,\mathbf{z}}(\mathbf{w}) = dq_\phi(\mathbf{w}) \, d\nu_{\phi\theta,\mathbf{w}}(\mathbf{z})$, as measures on $\mathcal{Z} \times \mathcal{Z}$. Define the directional alignment potential fields

$$U_{\theta \to \phi}(\mathbf{z}) := -\int_{\mathcal{Z}} s(\mathbf{z}, \mathbf{w}) \, d\nu_{\theta\phi,\mathbf{z}}(\mathbf{w}), \ q_\theta\text{-a.e. } \mathbf{z},$$

$$U_{\phi \to \theta}(\mathbf{w}) := -\int_{\mathcal{Z}} s(\mathbf{z}, \mathbf{w}) \, d\nu_{\phi\theta,\mathbf{w}}(\mathbf{z}), \ q_\phi\text{-a.e. } \mathbf{w}.$$

**Definition 4.2** (Symmetric multimodal parametric energy). Fix $\tau > 0$. Recall the kernel-smoothed $\mu$-densities $\tilde{\rho}_{\theta,\tau}, \tilde{\rho}_{\phi,\tau}$ from Def. 2.2. Define the directional multimodal energies by

$$\mathcal{J}_\tau^{x \to y}(\theta, \phi) := \frac{1}{\tau} \int_{\mathcal{Z}} U_{\theta \to \phi} \, dq_\theta - H_\times(q_\theta, \tilde{\rho}_{\phi,\tau}),$$

$$\mathcal{J}_\tau^{y \to x}(\theta, \phi) := \frac{1}{\tau} \int_{\mathcal{Z}} U_{\phi \to \theta} \, dq_\phi - H_\times(q_\phi, \tilde{\rho}_{\theta,\tau}),$$

and the symmetric multimodal parametric energy by

$$\mathcal{J}_\tau^{\mathrm{Sym}}(\theta, \phi) := \frac{1}{2} \left( \mathcal{J}_\tau^{x \to y}(\theta, \phi) + \mathcal{J}_\tau^{y \to x}(\theta, \phi) \right). \quad (9)$$

*Remark* 4.2 (Directional asymmetry is generic). Although both directions are induced by the same pair law $\pi_{\theta\phi}$, the two regular conditional laws $\nu_{\theta\phi,\mathbf{z}}$ and $\nu_{\phi\theta,\mathbf{w}}$ need not coincide. Consequently, $U_{\theta \to \phi}$ and $U_{\phi \to \theta}$ are generally distinct: there is no single unimodal-style potential field unless an additional compatibility condition is imposed on the conditional laws.

**Theorem 4.1** (Large-batch multimodal dynamics). *Consider the symmetric multimodal loss $\mathcal{L}_{\mathrm{Sym}}(\theta, \phi)$ in Eq. (2) and the energy $\mathcal{J}_\tau^{\mathrm{Sym}}(\theta, \phi)$ in Def. 4.2. Assume Asms. 2.1 and 4.1. Then, for any fixed $\tau > 0$ and $(\theta, \phi) \in \Theta \times \Phi$, as $N \to \infty$,*

$$\left| \mathcal{L}_{\mathrm{Sym}}(\theta, \phi) - \mathcal{J}_\tau^{\mathrm{Sym}}(\theta, \phi) - \log\left(N V_\kappa(\tau)\right) \right| \to 0,$$

$$\text{and } \left\| \nabla_{(\theta,\phi)} \mathcal{L}_{\mathrm{Sym}}(\theta, \phi) - \nabla_{(\theta,\phi)} \mathcal{J}_\tau^{\mathrm{Sym}}(\theta, \phi) \right\| \to 0.$$

*Remark* 4.3. Since the additive $\log\left(N V_\kappa(\tau)\right)$ is constant in $(\theta, \phi)$, large-batch symmetric InfoNCE follows gradient descent on $\mathcal{J}_\tau^{\mathrm{Sym}}$ up to vanishing error. Unlike the unimodal case, the energy is intrinsically cross-coupled: each modality evaluates its cross-entropy against the other's smoothed density field.

### 4.2. Cross-Modal Divergence Coupling and Barrier-Driven Co-Adaptation

To interpret the coupled parametric energy $\mathcal{J}_\tau^{\mathrm{Sym}}$, we lift it to an intrinsic functional over modality *densities* on $\mathcal{Z}$. Relative to the unimodal free energy, the key new term is a *negative* symmetric divergence coupling. This coupling persists at fixed temperature and qualitatively changes the coordinate-wise geometry.

**Definition 4.3** (Intrinsic multimodal energy). Fix $\tau > 0$ and let $U_{1 \to 2}, U_{2 \to 1} : \mathcal{Z} \to \mathbb{R}$ be fixed Borel measurable potential fields. Fix density floors $\underline{\rho}_1, \underline{\rho}_2 > 0$ satisfying $\underline{\rho}_1 \mu(\mathcal{Z}) < 1$, $\underline{\rho}_2 \mu(\mathcal{Z}) < 1$, and define $\mathcal{P}_{\mu, \underline{\rho}_i}(\mathcal{Z}) := \left\{ \rho \in L^1(\mathcal{Z}) : \rho \geq \underline{\rho}_i \ \mu\text{-a.e.}, \int_{\mathcal{Z}} \rho \, d\mu = 1 \right\}$. For $(\rho_1, \rho_2) \in \mathcal{P}_{\mu, \underline{\rho}_1}(\mathcal{Z}) \times \mathcal{P}_{\mu, \underline{\rho}_2}(\mathcal{Z})$, define the intrinsic multimodal energy by

$$\mathcal{F}_{\tau, \mathbf{U}_{1,2}}^{\mathrm{Sym}}(\rho_1, \rho_2) :=$$
$$\frac{1}{2}\left( \mathcal{F}_{\tau, U_{1 \to 2}}(\rho_1) + \mathcal{F}_{\tau, U_{2 \to 1}}(\rho_2) \right) - D_{\mathrm{KL}}^{\mathrm{Sym}}(\rho_1, \rho_2), \quad (10)$$

where $\mathbf{U}_{1,2} := (U_{1 \to 2}, U_{2 \to 1})$, $\mathcal{F}_{\tau, U_{1 \to 2}}$ and $\mathcal{F}_{\tau, U_{2 \to 1}}$ are the unimodal free energies from Eq. (7), and

$$D_{\mathrm{KL}}^{\mathrm{Sym}}(\rho_1, \rho_2) := \frac{1}{2}\left( D_{\mathrm{KL}}(\rho_1 \| \rho_2) + D_{\mathrm{KL}}(\rho_2 \| \rho_1) \right),$$

with $D_{\mathrm{KL}}(\rho_1 \| \rho_2) := \int_{\mathcal{Z}} \rho_1 \log(\rho_1/\rho_2) \, d\mu$ the KL divergence between the $\mu$-densities $\rho_1$ and $\rho_2$.

*Remark* 4.4. The constraint $\rho_i \geq \underline{\rho}_i$ is a technical device that rules out degenerate $-\infty$ energies and makes the best-response geometry explicit. See App. A.3 for a discussion.

**Proposition 4.1** (Barrier best response and extremal convergence). *Assume $\mathcal{Z}$ is compact with $\mu(\mathcal{Z}) < \infty$, and let $\mathcal{F}_{\tau, \mathbf{U}_{1,2}}^{\mathrm{Sym}}$ be as in Def. 4.3. Fix $\rho_2 \in \mathcal{P}_{\mu, \underline{\rho}_2}(\mathcal{Z})$ and define the effective potential field $V_{1|2}(\mathbf{z}) := \frac{1}{\tau} U_{1 \to 2}(\mathbf{z}) + \log \rho_2(\mathbf{z})$. Then the coordinate map $\rho_1 \mapsto \mathcal{F}_{\tau, \mathbf{U}_{1,2}}^{\mathrm{Sym}}(\rho_1, \rho_2)$ is concave on $\mathcal{P}_{\mu, \underline{\rho}_1}(\mathcal{Z})$, and its infimum is approached by densities that place the excess mass $M_{\mathrm{ex}}^{(1)} := 1 - \underline{\rho}_1 \mu(\mathcal{Z})$ on approximate minimizers of $V_{1|2}$ while leaving only the floor $\underline{\rho}_1$ elsewhere. An analogous statement for $M_{\mathrm{ex}}^{(2)} := 1 - \underline{\rho}_2 \mu(\mathcal{Z})$ holds when optimizing in $\rho_2$ with $\rho_1$ fixed, with effective potential field $V_{2|1}(\mathbf{z}) := \frac{1}{\tau} U_{2 \to 1}(\mathbf{z}) + \log \rho_1(\mathbf{z})$. The explicit extremal construction is given in App. D.2.*

*Remark* 4.5 (Barrier-driven co-adaptation). Unlike the intrinsic unimodal Gibbs-like free energy, the multimodal functional does not admit Gibbs-type coordinate minimization. The term $\log \rho_2$ enters $V_{1|2} = U_{1 \to 2}/\tau + \log \rho_2$ as an occupied-mass barrier: regions where modality 2 already assigns high density become more expensive for modality 1, whereas low-density regions of modality 2 are comparatively favored, subject to the alignment potential. Hence the best response is extremal rather than Gibbs-like: after maintaining the density floor, modality 1 places its excess mass near minimizers of $V_{1|2}$, and symmetrically for modality 2. This mutual avoidance of occupied high-density regions, together with alignment-dependent basin selection, induces a winner-take-all form of co-adaptation.

**Theorem 4.2** (Low-temperature multimodal energy consistency). *Assume Asms. 2.1, 3.2 and 4.1. Fix any $(\theta, \phi) \in \Theta \times \Phi$, and assume $q_\theta \ll \mu$, $q_\phi \ll \mu$ with continuous densities $\rho_\theta, \rho_\phi$ bounded below: $\inf_{\mathbf{z} \in \mathcal{Z}} \rho_\theta(\mathbf{z}) \geq \underline{\rho}_\theta > 0$, $\inf_{\mathbf{z} \in \mathcal{Z}} \rho_\phi(\mathbf{z}) \geq \underline{\rho}_\phi > 0$. Let $\varepsilon_{\mathrm{kde}}^{(\theta)}(\tau) := \|\tilde{\rho}_{\theta,\tau} - \rho_\theta\|_\infty$ and $\varepsilon_{\mathrm{kde}}^{(\phi)}(\tau) := \|\tilde{\rho}_{\phi,\tau} - \rho_\phi\|_\infty$. Then there exists $\tau_0(\theta, \phi) > 0$ such that for all $0 < \tau \leq \tau_0(\theta, \phi)$,*

$$\left| \mathcal{J}_\tau^{\mathrm{Sym}}(\theta, \phi) - \mathcal{F}_{\tau, \mathbf{U}_{\theta,\phi}}^{\mathrm{Sym}}(\rho_\theta, \rho_\phi) \right| \leq \frac{\varepsilon_{\mathrm{kde}}^{(\theta)}(\tau) + \varepsilon_{\mathrm{kde}}^{(\phi)}(\tau)}{\min\{\underline{\rho}_\theta, \underline{\rho}_\phi\}},$$

*where $\mathcal{F}_{\tau, \mathbf{U}_{\theta,\phi}}^{\mathrm{Sym}}(\rho_\theta, \rho_\phi)$ is the intrinsic multimodal energy from Def. 4.3, evaluated on $\mathcal{P}_{\mu, \underline{\rho}_\theta}(\mathcal{Z}) \times \mathcal{P}_{\mu, \underline{\rho}_\phi}(\mathcal{Z})$, with $\mathbf{U}_{\theta,\phi} := (U_{\theta \to \phi}, U_{\phi \to \theta})$. In particular,*

$$\left| \mathcal{J}_\tau^{\mathrm{Sym}}(\theta, \phi) - \mathcal{F}_{\tau, \mathbf{U}_{\theta,\phi}}^{\mathrm{Sym}}(\rho_\theta, \rho_\phi) \right| \to 0, \qquad \text{as } \tau \to 0^+.$$

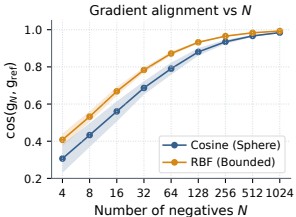 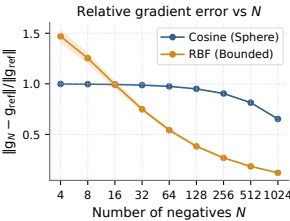

*Figure 3.* Large-batch gradient consistency across critics. Finite-batch InfoNCE gradients are compared against a high-fidelity large-batch reference as the number of negatives $N$ increases. Left: gradient alignment with the reference ($\uparrow$). Right: relative gradient error ($\downarrow$). Results are shown for both the cosine critic in the spherical regime and the RBF critic in the compact Euclidean regime, with mean $\pm$ std over 20 seeds.

The limit $\tau \to 0^+$ controls only the KDE mismatches, driving $\tilde{\rho}_{.,\tau} \to \rho_.$; it does not remove the divergence term $D_{\mathrm{KL}}^{\mathrm{Sym}}(\rho_\theta, \rho_\phi)$. Because this term enters Def. 4.3 with a negative sign, the intrinsic landscape favors separation rather than matching, even in the sharp-kernel regime. Fig. 2b illustrates this persistent repulsion by estimating the symmetric KL divergence of the induced distributions under increasing levels of cross-modal pairing misalignment.

**Parametric inheritance under model expressivity.** Combining Thm. 4.2 with the barrier best-response geometry in Prop. 4.1 yields a parametric inheritance statement: under a suitable model-expressivity condition, minimizers of the practical energy $\mathcal{J}_\tau^{\mathrm{Sym}}$ concentrate their excess mass near minimizers of the effective barrier potentials $V_{\theta|\phi}$ and $V_{\phi|\theta}$. Cor. B.2 formalizes this condition and establishes the inheritance result; see App. B.4.

> ☞ **Punchline 2** (Multimodal InfoNCE geometry)**.** Symmetric multimodal InfoNCE is pairwise attractive yet population-level repulsive. While individual cross-modal pairs align, the cross-coupled geometry can make each modality's marginal act as a repulsive *population barrier* against the other. In the analyzed regime, this makes modality gap a natural geometric consequence.

## 5. Experiments

We validate the theory at two levels. First, controlled numerical experiments isolate the core mechanisms predicted by the analysis: large-batch gradient consistency, unimodal Gibbs-type concentration, and structural modality gap under controlled cross-modal misalignment. Second, on MS-COCO (Lin et al., 2014), we test whether the same signatures appear in pretrained CLIP-like models, and whether weakening image–text compatibility systematically enlarges the representational modality gap.

**Large-batch gradient consistency.** We first test the large-batch consistency predicted by Thm. 3.1. On a synthetic Gaussian-mixture dataset, we compare finite-batch InfoNCE gradients with a high-fidelity large-batch reference under both a spherical cosine regime and a compact Euclidean RBF regime. We vary the number of negatives $N$ and report two diagnostics: gradient alignment with the reference and relative gradient error. As shown in Fig. 3, agreement with the large-batch reference improves steadily as $N$ increases in both regimes: alignment rises, while relative error falls toward zero. This matches the consistency prediction of Thm. 3.1, showing that the deterministic large-batch energy is already a good proxy for training dynamics at moderate $N$. Full experimental details are deferred to App. E.1.

**Unimodal equilibrium and low-$\tau$ concentration.** We next test the intrinsic unimodal picture from § 3. On a smooth two-well potential on $\mathbb{S}^2$, we optimize a particle-based surrogate of $\mathcal{F}_{\tau,U}$ and compare the resulting particle distribution with the Gibbs law $\rho_\tau^\star \propto \exp(-U/\tau)$ across temperatures. As shown in Fig. 4, the trained particles closely follow the Gibbs trend: at high temperature, both remain diffuse, while at low temperature, both concentrate near the wells. This visual agreement is quantitatively summarized by the cap-mass trend in Fig. 2a, which increases monotonically as $\tau$ decreases for both the Gibbs baseline and the trained particle system. Together, these results support the unique-equilibrium and low-temperature concentration predictions of the intrinsic theory. Further experimental details are deferred to App. E.2.

**Structural modality gap under controlled misalignment.** We next test the central multimodal prediction from § 4: under heterogeneous conditional laws, exact marginal matching should be fragile rather than generic. We construct a latent-angle experiment in which both modalities share the same underlying semantic variable, while one modality is perturbed by controlled angular misalignment noise $\sigma_{\mathrm{mis}}$. We train symmetric CLIP-style linear encoders on $\mathbb{S}^1$ and examine both the learned joint coupling and the induced marginal discrepancy. As shown in Fig. 5, increasing $\sigma_{\mathrm{mis}}$ progressively broadens and deforms the diagonal coupling structure, indicating weaker cross-modal alignment despite preserved coarse latent structure. This is accompanied by the increasing marginal discrepancy reported in Fig. 2b. Together, these results support the theoretical claim that the modality gap is a structural consequence of cross-modal heterogeneity rather than merely a finite-sample artifact or optimization failure. Further details are deferred to App. E.3.

**MS-COCO studies with pretrained OpenCLIP models.** We finally test whether the multimodal mechanism appears in realistic image–text representations on MS-COCO. In an observational study, we evaluate frozen OpenCLIP check-

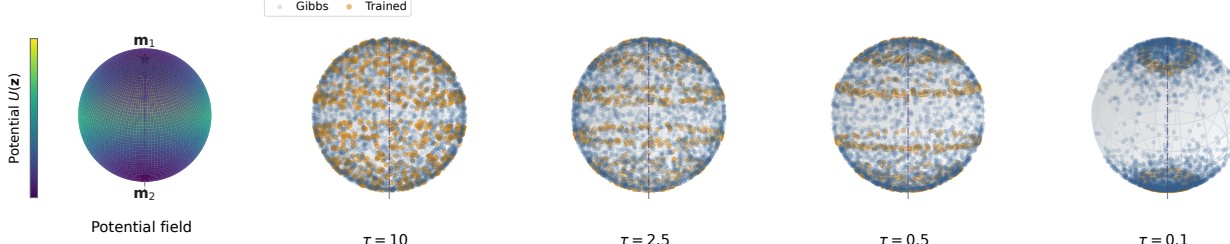

*Figure 4.* Unimodal potential landscape on $\mathbb{S}^2$ and equilibria across temperature. Left: the two-well potential $U$, with minima centers $\mathbf{m}_1$ and $\mathbf{m}_2$ marked. Right: Gibbs samples (blue) and trained particles (orange) for several temperatures $\tau$. At high temperature, both distributions remain diffuse due to the stronger role of entropy; as $\tau$ decreases, both concentrate around the low-energy wells.

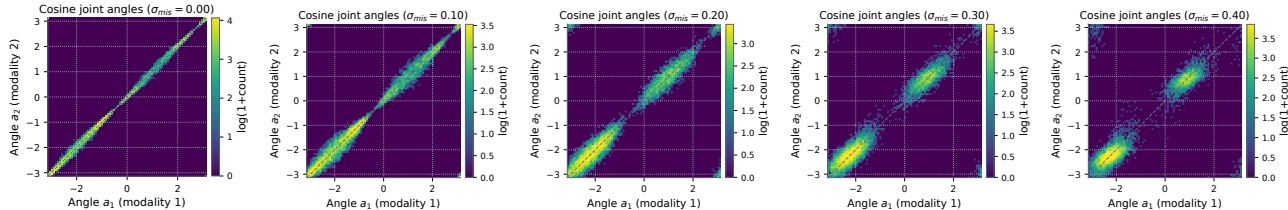

*Figure 5.* Joint-angle coupling under controlled misalignment. Each panel shows a histogram of joint angles $(a_1, a_2)$ for $\sigma_{\mathrm{mis}} \in \{0.0, 0.1, 0.2, 0.3, 0.4\}$. When $\sigma_{\mathrm{mis}} = 0$, the learned coupling is sharply concentrated near the diagonal, indicating a small modality gap. As $\sigma_{\mathrm{mis}} \uparrow$, the diagonal band broadens and deforms, revealing a noisier cross-modal coupling while preserving the coarse latent modes.

points on `val2017` using average Recall@1 together with two geometric diagnostics, the centroid gap $\|\mu_I - \mu_T\|$ and the energy distance between image and text marginals. As shown in Fig. 6a, stronger retrieval does not necessarily imply a smaller modality gap: models with similar AvgR@1 can exhibit substantially different centroid and energy gaps. We then perform a controlled intervention on `train2017` by replacing each caption, with probability $p$, by a caption from another image sharing at least one COCO category, thereby preserving coarse semantic relatedness while weakening instance-level correspondence. In Fig. 6b, increasing $p$ consistently degrades retrieval and enlarges the centroid gap for both RN50 and ViT-B-16. Together, these results support the prediction that modality gap is not reducible to retrieval quality alone, and that weakening conditional compatibility systematically pushes the image and text marginals further apart. Further details are deferred to App. E.4.

## 6. Related Work

**Contrastive objectives and population geometry.** InfoNCE is an instance of noise-contrastive density-ratio estimation, contrasting positives with negatives from a background marginal through softmax normalization (Gutmann & Hyvärinen, 2010; Mnih & Teh, 2012; Oord et al., 2018). It is often used as a scalable surrogate for mutual-information criteria (Poole et al., 2019; Tschannen et al., 2020), though recent work argues it is better viewed as learning structured density ratios than as an accurate MI estimator (Ryu et al., 2026). A related lens studies population landscapes in-

duced by contrastive objectives, notably through alignment–uniformity on hyperspherical embeddings (Wang & Isola, 2020; Wang & Liu, 2021; Chen et al., 2021; Betser et al., 2026); other analyses show such objectives can admit multiple, sometimes trivial, minimizers, with optimization implicitly selecting structured solutions (Calder & Lee, 2025). We instead prove large-batch gradient consistency and derive deterministic energy landscapes for representations on compact manifolds.

**Training dynamics, learning theory, and loss design.** A distinct line of theory studies contrastive learning under stylized data or generative assumptions. Some analyses track feature or weight dynamics under imperfections such as misalignment, imbalance, or augmentation-induced variation, explaining suppression, robustness, or the effects of filtering and pruning (Sun et al., 2025; Liao et al., 2026). Others develop learning-theoretic guarantees under latent-class or multi-view models, and analyze how negative sampling and objective variants affect transfer (Saunshi et al., 2019; 2022; Chuang et al., 2020; Tian et al., 2020; HaoChen & Ma, 2023; Ji et al., 2023); recent CLIP-style analyses further characterize optimal similarity structure and motivate refined objectives (Uesaka et al., 2025; Yoshida et al., 2025). We do not track weight dynamics under a specific latent model, but characterize InfoNCE's population geometry via large-batch energy limits and variational functionals.

**Multimodal representation alignment and modality gap.** Empirical work shows that CLIP-style alignment can coex-

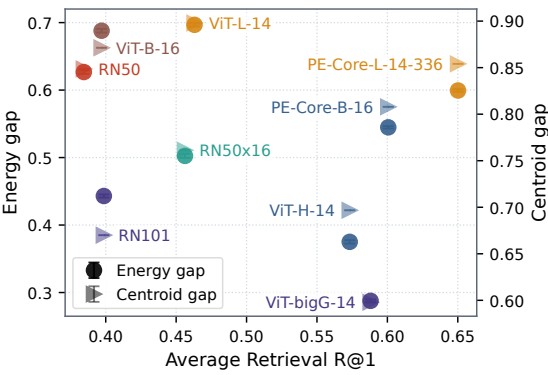

(a) Observational study on pretrained OpenCLIP models.

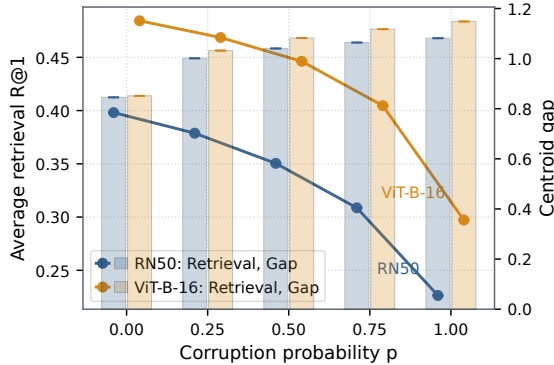

(b) Interventional study with caption corruption.

*Figure 6.* MS-COCO validation of the modality-gap mechanism. (**a**) On frozen OpenCLIP checkpoints, retrieval quality and cross-modal geometric agreement are related but not equivalent. (**b**) Controlled same-category corruption: Even a semantically plausible weakening of instance-level correspondence systematically enlarges the modality gap while degrading retrieval.

ist with a persistent modality gap (Liang et al., 2022; Shi et al., 2023; Levi & Gilboa, 2025; Betser et al., 2025), and recent methods seek to control it through adapters, alternative losses, or explicit regularizers. Yi et al. (2025) shows that unconstrained or cone-constrained optima close the gap, while dimension collapse into distinct hyperplanes leaves a gap equal to their smallest angle; Yin et al. (2026) augment InfoNCE with a kernel-estimated Cauchy–Schwarz divergence to align modality distributions, including unpaired settings; and pairwise sigmoid objectives likewise indicate that cross-modal coupling is sensitive to objective design (Zhai et al., 2023). We provide a geometric mechanism: symmetric InfoNCE induces cross-modal coupling in which each modality's density shapes the other's effective field via a persistent negative symmetric divergence term, making the modality gap a structural consequence of the objective.

**Identifiability in contrastive learning.** A complementary line of work asks when contrastive objectives recover latent generative factors under data-generating assumptions. In nonlinear ICA, auxiliary variables such as temporal nonstationarity enable source recovery up to component-wise transformations (Hyvarinen & Morioka, 2016). For invertible-mixing models, global optima can invert the generator up to linear ambiguities, often orthogonal transformations on hyperspherical latents (Zimmermann et al., 2021; Liu et al., 2026). Multi-view models separating invariant content from view-varying style admit recovery of the invariant block up to an invertible mapping (Von Kügelgen et al., 2021; Cai et al., 2024); in multimodal settings with distinct generative mechanisms, contrastive learning further block-identifies shared factors under nontrivial dependencies (Daunhawer et al., 2023). Broader causal analyses with partial observability and data-misalignment formulations also establish identifiability of shared semantics up to smooth bijections (Yao et al., 2024; Cai et al., 2025). We instead character-

ize objective-induced optimization geometry that standard identifiability analyses do not explain.

# 7. Conclusion

We developed a population-energy view of InfoNCE that exposes a geometric bifurcation in unimodal and symmetric multimodal contrastive learning. In the unimodal regime, intrinsic energy is convex and Gibbs-like, with entropy selecting an internally dispersed equilibrium within the aligned basin. In the multimodal regime, conditional heterogeneity gives rise to a negative symmetric divergence coupling, sustaining a persistent modality gap as a structural consequence of the objective rather than merely an optimization artifact. These results show that pairwise alignment alone is not sufficient to control cross-modal marginal structure and motivate objectives that explicitly regulate cross-modal divergence. More broadly, our analysis suggests that multimodal contrastive learning should be understood not only through matched pairs, but through the coupled population geometry of representation marginals. This shift from pointwise matching to population geometry provides a framework for analyzing and designing future representation-learning objectives. See App. F for further discussion.

# Acknowledgements

This project was partially supported by the Responsible AI Research (RAIR) Centre, Australia (Z. Zhang, Y. Liu, and J. Q. Shi). We sincerely thank the anonymous reviewers for their constructive feedback.

# Impact Statement

This paper presents work whose goal is to advance the field of Machine Learning. While there are broad societal conse-

quences associated with the general advancement of AI, our specific theoretical contributions do not present immediate societal impacts that require dedicated discussion.

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

# Appendix

## Contents

*Table 1.* Core notations used throughout the paper.

**Spaces, Measures, Encoders and Laws**

| | |
|---|---|
| $\mathcal{X}, \mathcal{Y}$ | Observation spaces; in the unimodal case both views lie in $\mathcal{X}$. |
| $\mathcal{Z} \subseteq \mathbb{R}^n$ | Common representation space, with volume measure $\mu$. |
| $\mathcal{P}(\mathcal{S})$ | Set of probability measures on a measurable space $\mathcal{S}$. |
| $\mathcal{P}_\mu(\mathcal{Z})$ | Set of probability densities on $\mathcal{Z}$ w.r.t. $\mu$. |
| $r_{\mathrm{um}} \in \mathcal{P}(\mathcal{X} \times \mathcal{X})$ | Unimodal positive-pair law governing $(\mathbf{x}, \tilde{\mathbf{x}})$. |
| $r_{\mathrm{mm}} \in \mathcal{P}(\mathcal{X} \times \mathcal{Y})$ | Multimodal positive-pair law governing $(\mathbf{x}, \mathbf{y})$. |
| $p_{\mathbf{x}}, p_{\mathbf{y}}$ | Marginal data laws on $\mathcal{X}$ and $\mathcal{Y}$, respectively. |
| $\Theta, \Phi$ | Compact parameter spaces of the encoder families. |
| $f_\theta : \mathcal{X} \to \mathcal{Z}$ | Encoder for the first modality. |
| $g_\phi : \mathcal{Y} \to \mathcal{Z}$ | Encoder for the second modality. |
| $q_\theta := (f_\theta)_\# p_{\mathbf{x}}$ | Encoded marginal law induced by $f_\theta$. |
| $q_\phi := (g_\phi)_\# p_{\mathbf{y}}$ | Encoded marginal law induced by $g_\phi$. |
| $\pi_{\theta\theta} := (f_\theta \times f_\theta)_\# r_{\mathrm{um}}$ | Encoded unimodal positive-pair law. |
| $\pi_{\theta\phi} := (f_\theta \times g_\phi)_\# r_{\mathrm{mm}}$ | Encoded multimodal positive-pair law. |

**InfoNCE Objectives and Kernel Quantities**

| | |
|---|---|
| $N \in \mathbb{N}_+$ | Number of negatives in the InfoNCE denominator. |
| $s(\mathbf{z}, \mathbf{w})$ | Similarity critic on $\mathcal{Z} \times \mathcal{Z}$. |
| $\kappa_\tau(\mathbf{z}, \mathbf{w}) = \exp(s(\mathbf{z}, \mathbf{w})/\tau)$ | Exponential similarity kernel at temperature $\tau > 0$. |
| $V_\kappa(\tau)$ | Kernel volume constant under Asm. 2.1. |
| $\mathcal{L}_{\mathrm{NCE}}(\theta)$ | Empirical unimodal InfoNCE loss. |
| $\mathcal{L}^{\mathrm{Sym}}(\theta, \phi)$ | Symmetric multimodal InfoNCE loss. |
| $\mathcal{L}_{\mathrm{NCE}}^{x \to y}(\theta, \phi), \mathcal{L}_{\mathrm{NCE}}^{y \to x}(\phi, \theta)$ | Directional multimodal InfoNCE losses for $\mathbf{x} \to \mathbf{y}$ and $\mathbf{y} \to \mathbf{x}$. |
| $\Gamma_{\theta,\tau}, \Gamma_{\phi,\tau}$ | Population partition fields at $\tau$, induced by $q_\theta$ and $q_\phi$, respectively. |
| $\tilde{\rho}_{\theta,\tau}, \tilde{\rho}_{\phi,\tau}$ | Kernel-smoothed densities at $\tau$, induced by $q_\theta$ and $q_\phi$, respectively. |

**Unimodal Geometry**

| | |
|---|---|
| $\nu_{\theta,\mathbf{z}}$ | Conditional law of the positive partner given encoded query $\mathbf{z}$ under $\pi_{\theta\theta}$. |
| $U_\theta(\mathbf{z}) : \mathcal{Z} \to \mathbb{R}$ | Unimodal alignment potential field: the negative conditional expected similarity at $\mathbf{z}$. |
| $H_\times(q, \tilde{\rho})$ | Cross-entropy of a probability measure $q$ against a positive density $\tilde{\rho}$. |
| $H(\rho)$ | Differential entropy of density $\rho$. |
| $\mathcal{J}_\tau(\theta)$ | Unimodal parametric energy optimized in the large-batch limit. |
| $\mathcal{F}_{\tau,U}(\rho)$ | Intrinsic unimodal free-energy functional over densities $\rho$. |

**Multimodal Geometry**

| | |
|---|---|
| $\nu_{\theta\phi,\mathbf{z}}$ | Conditional law of the $\phi$-encoded partner at point $\mathbf{z}$ under $\pi_{\theta\phi}$. |
| $\nu_{\phi\theta,\mathbf{w}}$ | Conditional law of the $\theta$-encoded partner at point $\mathbf{w}$ under $\pi_{\theta\phi}$. |
| $U_{\theta \to \phi}(\mathbf{z}), U_{\phi \to \theta}(\mathbf{w})$ | Directional alignment potential seen by $\theta$ against $\phi$, and the reverse version. |
| $\mathcal{J}_\tau^{x \to y}(\theta, \phi)$ | Directional multimodal parametric energy for $x \to y$. |
| $\mathcal{J}_\tau^{y \to x}(\theta, \phi)$ | Directional multimodal parametric energy for $y \to x$. |
| $\mathcal{J}_\tau^{\mathrm{Sym}}(\theta, \phi)$ | Symmetric multimodal parametric energy. |
| $\mathbf{U}_{1,2} := (U_{1 \to 2}, U_{2 \to 1})$ | Pair of directional potentials in the intrinsic multimodal functional. |
| $\mathcal{F}_{\tau,\mathbf{U}_{1,2}}^{\mathrm{Sym}}(\rho_1, \rho_2)$ | Intrinsic symmetric multimodal energy. |
| $D_{\mathrm{KL}}(\rho_1 \| \rho_2)$ | Kullback–Leibler divergence. |
| $D_{\mathrm{KL}}^{\mathrm{Sym}}(\rho_1, \rho_2)$ | Symmetric KL divergence, $\frac{1}{2}(D_{\mathrm{KL}}(\rho_1 \| \rho_2) + D_{\mathrm{KL}}(\rho_2 \| \rho_1))$. |
| $\underline{\rho}_i > 0$ | Density floor(s) used to restrict the feasible probability density sets $\mathcal{P}_{\mu,\underline{\rho}_i}(\mathcal{Z})$. |
| $\mathcal{P}_{\mu,\underline{\rho}}(\mathcal{Z})$ | Strictly positive density class satisfying $\rho \geq \underline{\rho}$ a.e. and $\int_{\mathcal{Z}} \rho \, d\mu = 1$. |
| $V_{1|2}(\mathbf{z})$ | Effective barrier potential for modality 1 given modality 2, $\frac{1}{\tau} U_{1 \to 2} + \log \rho_2$. |
| $V_{2|1}(\mathbf{z})$ | Effective barrier potential for modality 2 given modality 1. |

# A. Interpretation of Assumptions

## A.1. On the Optimization Regularity Conditions

We argue that the regularity conditions in Asm. 3.1 are compatible with standard practice in representation learning. We instantiate these conditions in two dominant geometric regimes: the spherical regime used in modern contrastive learning (e.g., SimCLR (Chen et al., 2020), CLIP (Radford et al., 2021)) and the Euclidean regime common in classical kernel methods. Both admit the generalized exponential form $\kappa_\tau(\mathbf{z}, \mathbf{w}) := \exp(s(\mathbf{z}, \mathbf{w})/\tau)$.

- *Spherical regime (von Mises–Fisher kernel).* Representations are constrained to the hypersphere $\mathbb{S}^{d-1}$ via $\ell_2$ normalization, and the critic is the cosine similarity $s(\mathbf{z}, \mathbf{w}) = \langle \mathbf{z}, \mathbf{w} \rangle$:

$$\kappa_\tau^{(\text{vmf})}(\mathbf{z}, \mathbf{w}) := \exp\left(\frac{\langle \mathbf{z}, \mathbf{w} \rangle}{\tau}\right), \qquad \forall \mathbf{z}, \mathbf{w} \in \mathbb{S}^{n-1}. \tag{11}$$

- *Euclidean regime (RBF kernel).* The representations lie in a compact subset of $\mathbb{R}^n$, and the similarity critic is the negative squared Euclidean distance $s(\mathbf{z}, \mathbf{w}) = -\|\mathbf{z} - \mathbf{w}\|^2$:

$$\kappa_\tau^{(\text{rbf})}(\mathbf{z}, \mathbf{w}) := \exp\left(-\frac{\|\mathbf{z} - \mathbf{w}\|^2}{\tau}\right). \tag{12}$$

On a generic compact subset of $\mathbb{R}^n$, this example satisfies the boundedness and smoothness requirements used in Asm. 3.1, but it does not, in general, satisfy the constant kernel-volume condition in Asm. 2.1 because boundary effects make $V_\tau(\mathbf{z}) := \int_{\mathcal{Z}} \kappa_\tau(z, w) \, d\mu(w)$ depend on $\mathbf{z}$. The condition holds exactly on homogeneous boundaryless domains, such as a flat torus with periodic distance/measure, or after an appropriate boundary-corrected normalization. If Asm. 2.1 is relaxed, the large-batch denominator still converges to the population partition field $\Gamma_{q,\tau}$; however, the clean identity $\Gamma_{q,\tau} = V_\kappa(\tau)\tilde{\rho}_{q,\tau}$ is replaced by a position-dependent normalization. This introduces additional boundary-volume correction terms, e.g., an effective potential contribution $\tau \log V_\tau(z)$, but does not change the large-batch consistency mechanism or the central population-field interpretation.

**Encoder regularity.** Condition Asm. 3.1 (i) requires that for each $\mathbf{x}$, the map $\theta \mapsto f_\theta(\mathbf{x})$ is $C^1$ and that its parameter Jacobian is uniformly bounded: $\sup_{\theta \in \Theta} \sup_{\mathbf{x} \in \mathcal{X}} \|J_\theta f_\theta(\mathbf{x})\| < \infty$. For standard neural encoders built from differentiable primitives (e.g., linear maps, convolutions, normalization layers) and smooth activations (e.g., GELU (Hendrycks, 2016), SiLU (Elfwing et al., 2018)), the map $\theta \mapsto f_\theta(\mathbf{x})$ is $C^1$ (or at least piecewise $C^1$, which suffices for almost-everywhere analysis). In practice, uniform Jacobian control is promoted by explicit Lipschitz constraints (e.g., spectral normalization, which bounds layer operator norms and hence the network Jacobian via composition (Miyato et al., 2018)) together with standard norm-regularizing training choices such as weight decay (Loshchilov & Hutter, 2019; Zhang et al., 2019). Accordingly, we interpret compactness of $\Theta$ as a mild *localization* assumption: although the ambient parameter space is $\mathbb{R}^p$, successful training typically keeps iterates within a bounded region (often formalized as "SGD iterates belong to a ball" in stability analyses (Lei & Ying, 2020)). One may therefore restrict the analysis to a sufficiently large compact subset $\Theta_R$ containing the training trajectory, or equivalently consider projected updates onto $\Theta_R$ as standard in constrained optimization (Boyd & Vandenberghe, 2004).

**Critic regularity.** Condition Asm. 3.1 (ii) requires that $s$ is $C^1$ on an open neighborhood of $\mathcal{Z} \times \mathcal{Z}$ and that $\sup_{(\mathbf{z}, \mathbf{w}) \in \mathcal{Z} \times \mathcal{Z}} \|\nabla s(\mathbf{z}, \mathbf{w})\| < \infty$. In the spherical regime, $s(\mathbf{z}, \mathbf{w}) = \langle \mathbf{z}, \mathbf{w} \rangle$ is $C^\infty$ on $\mathbb{R}^d \times \mathbb{R}^d$ with gradient norms $\|\nabla_{\mathbf{z}} s\| = \|\mathbf{w}\|$ and $\|\nabla_{\mathbf{w}} s\| = \|\mathbf{z}\|$. Since $\|\mathbf{z}\| = \|\mathbf{w}\| = 1$, these are uniformly bounded by 1. In the Euclidean regime, $s(\mathbf{z}, \mathbf{w}) = -\|\mathbf{z} - \mathbf{w}\|^2$ is also $C^\infty$ with gradients scaling linearly with distance. Thus, provided $\mathcal{Z}$ is compact (e.g., via bounded activations like $\tanh$ or explicit clipping), $\|\nabla s\|$ remains uniformly bounded on $\mathcal{Z} \times \mathcal{Z}$.

**Boundedness of the kernel.** For any fixed $\tau > 0$, the exponential kernel $\kappa_\tau = \exp(s/\tau)$ is strictly positive and bounded. For the spherical regime, $\langle \mathbf{z}, \mathbf{w} \rangle \in [-1, 1]$ implies $\kappa_\tau^{(\text{vmf})}(\mathbf{z}, \mathbf{w}) \in [\exp(-1/\tau), \exp(1/\tau)]$. For the RBF regime on a compact domain with diameter $D = \text{diam}(\mathcal{Z})$, we have $\kappa_\tau^{(\text{rbf})}(\mathbf{z}, \mathbf{w}) \in [\exp(-D^2/\tau), 1]$. Crucially, in both cases, the kernel is bounded away from zero ($\kappa \geq c > 0$), ensuring that the logarithmic loss terms are well-defined and Lipschitz continuous.

## A.2. Interpretation of the Sharp Diagonal Peak Regime

**Geometric Interpretation.** Asm. 3.2 is a uniform non-degeneracy condition on the similarity critic $s(\mathbf{z}, \mathbf{w})$ near the diagonal $\mathbf{w} = \mathbf{z}$. The inequalities in Eq. (8) state that for each anchor $\mathbf{z}$, the function $\mathbf{w} \mapsto s(\mathbf{z}, \mathbf{w})$ has an isolated maximizer at $\mathbf{w} = \mathbf{z}$ and admits a locally quadratic upper and lower envelope in geodesic distance. For the exponential kernel $\kappa_\tau(\mathbf{z}, \mathbf{w}) = \exp(s(\mathbf{z}, \mathbf{w})/\tau)$, this condition implies that the normalized kernel

$$\bar{\kappa}_\tau := \frac{\kappa_\tau(\mathbf{z}, \mathbf{w})}{\int_{\mathcal{Z}} \kappa_\tau(\mathbf{z}, \mathbf{u}) \mathrm{d}\mu(\mathbf{u})}$$

concentrates its mass in a geodesic ball $\mathbb{B}_{c\sqrt{\tau}}(\mathbf{z}) := \{\mathbf{w} \in \mathcal{Z} \mid d_{\mathcal{Z}}(\mathbf{z}, \mathbf{w}) < c\sqrt{\tau}\}$ for some constant $c > 0$ uniformly in $\mathbf{z}$. Intuitively, $\bar{\kappa}(\mathbf{z}, \cdot)$ behaves like a Gaussian bump of bandwidth $\sqrt{\tau}$ centered at $\mathbf{z}$. As a result, the partition field $\Gamma_{\theta,\tau}(\mathbf{z}) = \int_{\mathcal{Z}} \kappa_\tau(\mathbf{z}, \mathbf{w}) \mathrm{d}q_\theta(\mathbf{w})$ probes only a local neighborhood of $\mathbf{z}$ when $\tau$ is small, and the smoothed density $\tilde{\rho}_{\theta,\tau} = \Gamma_{\theta,\tau}/V_\kappa(\tau)$ becomes a controlled local smoothing $\rho_\theta$.

**Examples.** For the RBF critic $s(z, w) = -\|z - w\|^2$, Eq. (8) holds with equality, so the sharp-diagonal condition in Asm. 3.2 is satisfied locally. This statement concerns the local quadratic peak condition only. The separate constant-volume condition in Asm. 2.1 holds exactly for the RBF kernel on homogeneous boundaryless geometries, such as a flat torus with periodic distance/measure, but not on a generic compact Euclidean subset without boundary correction. On the hypersphere with cosine similarity, $1 - \langle z, w \rangle \asymp d_{\mathbb{S}^{n-1}}(z, w)^2$ for nearby points, and the vMF kernel is isotropic on a compact homogeneous manifold, so both the local sharp-peak behavior and the constant-volume condition hold exactly.

## A.3. On Density Floors and Positivity of the Effective Laws

Several parts of our analysis involve logarithms of representation densities (e.g., cross-entropies and KL-type terms). In the unimodal intrinsic problem this is benign—the entropy term $-H(\rho)$ already acts as an interior regularizer and the Gibbs minimizer is well-defined without imposing a pointwise lower bound. In the *multimodal* intrinsic problem, however, the objective *subtracts* a symmetric KL-type divergence, creating an explicit "barrier" mechanism; without positivity constraints the variational problem becomes ill-posed. This section explains why (i) the *parametric* large-batch objective is automatically interior due to smoothing, and (ii) the intrinsic density floor is a faithful theoretical proxy for that interior regime.

**Smoothed densities.** In the large-batch expansion, the quantities that appear are the kernel-smoothed $\mu$-densities $\tilde{\rho}_{\theta,\tau}, \tilde{\rho}_{\phi,\tau}$ from Def. 2.2, not the raw pushforwards $q_\theta, q_\phi$. Because the exponential kernel satisfies $\kappa_\tau(\mathbf{z}, \mathbf{w}) = \exp(s(\mathbf{z}, \mathbf{w})/\tau) > 0$ pointwise, we have for every $\tau > 0$ and every probability measure $q_\theta$,

$$\tilde{\rho}_{\theta,\tau}(\mathbf{z}) = \frac{1}{V_\kappa(\tau)} \int_{\mathcal{Z}} \kappa_\tau(\mathbf{z}, \mathbf{w}) \, \mathrm{d}q_\theta(\mathbf{w}) > 0, \qquad \forall \mathbf{z} \in \mathcal{Z},$$

and analogously for $\tilde{\rho}_{\phi,\tau}$. Moreover, when $\kappa_\tau$ is continuous (as in our standing setting), $\tilde{\rho}_{\theta,\tau}$ is continuous on compact $\mathcal{Z}$ and therefore attains a strictly positive minimum

$$m_{\theta,\tau} := \min_{\mathbf{z} \in \mathcal{Z}} \tilde{\rho}_{\theta,\tau}(\mathbf{z}) > 0,$$

and similarly $m_{\phi,\tau} > 0$. Thus all log-terms appearing in the parametric energy are well-defined at finite temperature. Importantly, $m_{\theta,\tau}$ depends on $(\theta, \tau)$ and we do *not* assume a uniform lower bound over $\Theta$ (or $\Theta \times \Phi$ in the multimodal case).

**From smoothed densities to raw parametric densities.** When we compare $\mathcal{J}_\tau(\theta)$ to intrinsic functionals expressed in terms of an unsmoothed density $\rho_\theta := \mathrm{d}q_\theta/\mathrm{d}\mu$ (when $q_\theta \ll \mu$), the only discrepancy is the KDE bias between $\tilde{\rho}_{\theta,\tau}$ and $\rho_\theta$. Concretely, whenever

$$\|\tilde{\rho}_{\theta,\tau} - \rho_\theta\|_\infty \leq \tfrac{1}{2} m_{\theta,\tau}, \tag{13}$$

we immediately obtain $\rho_\theta(\mathbf{z}) \geq \tfrac{1}{2} m_{\theta,\tau}$ for all $\mathbf{z}$, i.e., the raw density inherits a (parameter-dependent) floor. The sharp-peak smoothing lemma (Lem. B.3) is precisely the tool that makes Eq. (13) achievable in the low-temperature regime: under Asm. 3.2 and mild regularity, the KDE mismatch vanishes as $\tau \downarrow 0^+$ for each fixed $\theta$ satisfying the lemma's hypotheses. Accordingly, the "density floor" used in the intrinsic best-response analysis should be viewed as a technical proxy for the fact that, at any finite $\tau$, the *effective* densities that govern the parametric objective are strictly positive and (in regimes of small KDE bias) the same interior behavior transfers to $\rho_\theta, \rho_\phi$.

**Density floor in multimodal analysis.** In the multimodal formulation (Def. 4.3), we restrict $(\rho_1, \rho_2)$ to

$$\mathcal{P}_{\mu,\underline{\rho}_i}(\mathcal{Z}) := \left\{ \rho \in L^1(\mathcal{Z}) : \rho \geq \underline{\rho}_i \mu\text{-a.e.}, \int_{\mathcal{Z}} \rho \, d\mu = 1 \right\},$$

for some fixed $\underline{\rho}_1, \underline{\rho}_2 > 0$ respectively. This floor serves two roles.

*(i) Well-posedness under negative symmetric divergence.* Because the multimodal functional contains a *negative* symmetric divergence coupling, allowing densities to vanish can make the objective unbounded below. Indeed, if $\rho_1 = 0$ on a set $\mathcal{A}$ with $\mu(\mathcal{A}) > 0$ while $\rho_2 > 0$ on $\mathcal{A}$, then $D_{\mathrm{KL}}(\rho_2 \| \rho_1) = +\infty$ and hence the negative symmetric divergence term drives $\mathcal{F}_{\tau,\mathbf{U}}^{\mathrm{mm}}(\rho_1, \rho_2) = -\infty$. The floor $\rho_i \geq \underline{\rho}_i$ prevents this degeneracy and makes the best-response problem meaningful. Equivalently, one may view the constraint as a hard version of the standard "interior" condition used to keep log-barrier objectives finite.

*(ii) Transparent best-response geometry.* With the floor, we can decompose

$$\rho_1(\mathbf{z}) = \underline{\rho}_1 + \eta_1(\mathbf{z}), \qquad \eta_1(\mathbf{z}) \geq 0, \qquad \int_{\mathcal{Z}} \eta_1(\mathbf{z}) \, d\mu =: M_{\mathrm{ex}}^{(1)} = 1 - \underline{\rho}_1 \, \mu(\mathcal{Z}),$$

and analogously for $\rho_2$. This isolates a fixed "background" mass from an "excess" budget $M_{\mathrm{ex}}^{(1)}$, making the extremal best-response behavior in Prop. 4.1 quantitatively controllable. In this sense, the density floor is not an arbitrary restriction: it is the simplest device that (a) mirrors the intrinsic interiority of the parametric smoothed quantities at finite $\tau$, and (b) exposes the barrier-driven coordinate-collapse mechanism in closed form.

# B. Lemmas and Extended Formal Statements

## B.1. Uniform Gradient Bounds for InfoNCE Objectives

In this section, we prove uniform gradient bounds for unimodal and multimodal InfoNCE objectives, which are used to justify limit–gradient interchange in Thms. 3.1 and 4.1.

### B.1.1. UNIMODAL OPTIMIZATION STABILITY

**Lemma B.1** (Stable unimodal optimization). *Assume Asm. 3.1. Fix any temperature $\tau > 0$ and any batch size $N \in \mathbb{N}_+$. Then the unimodal InfoNCE objective defined in Eq. (1) has a uniformly bounded gradient over the parameter space $\Theta$. Formally, there exists a constant $C(\tau) < \infty$ such that $\|\nabla_\theta \mathcal{L}_{\mathrm{NCE}}(\theta)\| \leq C(\tau)$, uniformly for all $\theta \in \Theta$.*

> ☞ **Proof intuition.** This proof establishes that the InfoNCE gradients cannot explode, providing a uniform upper bound that is strictly independent of the number of negative samples $N$. The intuition rests on two core mechanisms: **(i)** Because the representation manifold is compact and both the encoder and similarity functions are sufficiently smooth with bounded Jacobians, the chain rule guarantees that the gradient of any single pairwise similarity score is inherently bounded. **(ii)** The gradient of the log-partition denominator mathematically simplifies to a softmax-weighted sum of the negative-pair gradients. Because this is a convex combination (i.e., a weighted average where coefficients sum to one), the aggregated gradient vector cannot stretch beyond the maximum length of the individual bounded vectors. Consequently, the contrastive push from an arbitrarily large sea of negatives is structurally prevented from overwhelming the gradient, ensuring global stability for the entire learning objective.

*Proof.* **Step 1: Bounding components in the loss.** Since $\mathcal{Z}$ is compact, the product $\mathcal{Z} \times \mathcal{Z}$ is compact. By Asm. 3.1 (ii), $s$ is $C^1$ on an open neighborhood of $\mathcal{Z} \times \mathcal{Z}$, so $\nabla s$ is continuous on $\mathcal{Z} \times \mathcal{Z}$ and hence attains its maximum norm on this compact set. For a fixed $\tau > 0$, define the temperature-dependent critic sensitivity bound

$$C_s(\tau) := \sup_{(\mathbf{z},\mathbf{w}) \in \mathcal{Z} \times \mathcal{Z}} \left\| \nabla_{(\mathbf{z},\mathbf{w})} \left( \tfrac{1}{\tau} s(\mathbf{z}, \mathbf{w}) \right) \right\| = \tfrac{1}{\tau} \sup_{(\mathbf{z},\mathbf{w}) \in \mathcal{Z} \times \mathcal{Z}} \|\nabla s(\mathbf{z}, \mathbf{w})\| < \infty. \tag{14}$$

Next, by Asm. 3.1 (i), the encoder Jacobian is uniformly bounded:

$$C_\Theta := \sup_{\theta \in \Theta} \sup_{\mathbf{x} \in \mathcal{X}} \|J_\theta f_\theta(\mathbf{x})\| < \infty. \tag{15}$$

For any $\mathbf{x}, \mathbf{x}' \in \mathcal{X}$ with representations $\mathbf{z} = f_\theta(\mathbf{x})$ and $\mathbf{w} = f_\theta(\mathbf{x}')$, define

$$\alpha_\theta(\mathbf{z}, \mathbf{w}) := \nabla_\theta\left(\tfrac{1}{\tau} s(\mathbf{z}, \mathbf{w})\right).$$

By the chain rule,

$$\alpha_\theta(\mathbf{z}, \mathbf{w}) = \nabla_\mathbf{z}\left(\tfrac{1}{\tau} s(\mathbf{z}, \mathbf{w})\right)^\top J_\theta f_\theta(\mathbf{x}) + \nabla_\mathbf{w}\left(\tfrac{1}{\tau} s(\mathbf{z}, \mathbf{w})\right)^\top J_\theta f_\theta(\mathbf{x}').$$

Therefore, by the triangle inequality and sub-multiplicativity,

$$\|\alpha_\theta(\mathbf{z}, \mathbf{w})\| \leq \left\|\nabla_\mathbf{z} \tfrac{1}{\tau} s(\mathbf{z}, \mathbf{w})\right\| \cdot \|J_\theta f_\theta(\mathbf{x})\| + \left\|\nabla_\mathbf{w} \tfrac{1}{\tau} s(\mathbf{z}, \mathbf{w})\right\| \cdot \|J_\theta f_\theta(\mathbf{x}')\| \leq 2\, C_s(\tau)\, C_\Theta. \tag{16}$$

Thus, the parameter gradient of any pairwise similarity term is uniformly bounded.

**Step 2: Gradient of the empirical partition function.** Fix an anchor $\mathbf{z} = f_\theta(\mathbf{x})$, a positive $\mathbf{v} = f_\theta(\tilde{\mathbf{x}})$, and negatives $\{\mathbf{w}_j\}_{j=1}^N$. Define the contrastive set $\mathcal{C}(\mathbf{z}) := \{\mathbf{v}\} \cup \{\mathbf{w}_j\}_{j=1}^N$. The gradient of the log-partition function $\log \mathsf{Z}_\mathcal{B}(\mathbf{z}) = \log \sum_{\mathbf{w} \in \mathcal{C}(\mathbf{z})} \exp(s(\mathbf{z}, \mathbf{w})/\tau)$ is given by the softmax-weighted sum of the inner gradients (see e.g. Boyd & Vandenberghe, 2004, p. 74). Since $\kappa_\tau = \exp(s/\tau) > 0$, we may differentiate $\log \mathsf{Z}_\mathcal{B}$ and write

$$\nabla_\theta \log \mathsf{Z}_\mathcal{B}(\mathbf{z}) = \sum_{\mathbf{w} \in \mathcal{C}(\mathbf{z})} \sigma_\mathcal{B}(\mathbf{w})\, \nabla_\theta \log \kappa_\tau(\mathbf{z}, \mathbf{w}) = \sum_{\mathbf{w} \in \mathcal{C}(\mathbf{z})} \sigma_\mathcal{B}(\mathbf{w})\, \alpha_\theta(\mathbf{z}, \mathbf{w}), \tag{17}$$

where

$$\sigma_\mathcal{B}(\mathbf{w}) := \frac{\kappa(\mathbf{z}, \mathbf{w})}{\mathsf{Z}_\mathcal{B}(\mathbf{z})} \quad \text{satisfies} \quad \sigma_\mathcal{B}(\mathbf{w}) \geq 0, \quad \sum_{\mathbf{w} \in \mathcal{C}(\mathbf{z})} \sigma_\mathcal{B}(\mathbf{w}) = 1.$$

Therefore, using Eq. (16),

$$\|\nabla_\theta \log \mathsf{Z}_\mathcal{B}(\mathbf{z})\| \leq \sum_{\mathbf{w} \in \mathcal{C}(\mathbf{z})} \sigma_\mathcal{B}(\mathbf{w})\, \|\alpha_\theta(\mathbf{z}, \mathbf{w})\| \leq 2\, C_s(\tau)\, C_\Theta \sum_{\mathbf{w} \in \mathcal{C}(\mathbf{z})} \sigma_\mathcal{B}(\mathbf{w}) = 2\, C_s(\tau)\, C_\Theta. \tag{18}$$

**Step 3: Bounding the loss gradient.** For a batch $\mathcal{B}$ and anchor $\mathbf{z} = f_\theta(\mathbf{x})$, define the per-batch loss

$$\ell_\mathcal{B}(\theta) := -\log \kappa_\tau(\mathbf{z}, \mathbf{v}) + \log \mathsf{Z}_\mathcal{B}(\mathbf{z}),$$

where $\mathbf{v} = f_\theta(\tilde{\mathbf{x}})$ is the positive match. Since $\nabla_\theta \log \kappa_\tau(\mathbf{z}, \mathbf{v}) = \alpha_\theta(\mathbf{z}, \mathbf{v})$,

$$\nabla_\theta \ell_\mathcal{B}(\theta) = -\alpha_\theta(\mathbf{z}, \mathbf{v}) + \nabla_\theta \log \mathsf{Z}_\mathcal{B}(\mathbf{z}).$$

By the triangle inequality and Eqs. (16) and (18),

$$\|\nabla_\theta \ell_\mathcal{B}(\theta)\| \leq \|\alpha_\theta(\mathbf{z}, \mathbf{v})\| + \|\nabla_\theta \log \mathsf{Z}_\mathcal{B}(\mathbf{z})\| \leq 2C_s(\tau)C_\Theta + 2C_s(\tau)C_\Theta = 4C_s(\tau)C_\Theta. \tag{19}$$

Finally, the population objective is $\mathcal{L}_{\mathrm{NCE}}(\theta) = \mathbb{E}_\mathcal{B}[\ell_\mathcal{B}(\theta)]$. Since $\ell_\mathcal{B}$ is differentiable in $\theta$ by Asm. 3.1 for each $\mathcal{B}$ and $\|\nabla_\theta \ell_\mathcal{B}(\theta)\| \leq 4C_s(\tau)C_\Theta$ provides an integrable dominating function, we may differentiate under the expectation by dominated convergence, obtaining

$$\nabla_\theta \mathcal{L}_{\mathrm{NCE}}(\theta) = \mathbb{E}_\mathcal{B}\big[\nabla_\theta \ell_\mathcal{B}(\theta)\big].$$

Taking norms and applying Jensen's inequality yields

$$\|\nabla_\theta \mathcal{L}_{\mathrm{NCE}}(\theta)\| \leq \mathbb{E}_\mathcal{B}\big[\|\nabla_\theta \ell_\mathcal{B}(\theta)\|\big] \leq 4C_s(\tau)C_\Theta.$$

Thus, for each fixed $\tau > 0$ and any $N \in \mathbb{N}_+$, the gradient of unimodal InfoNCE defined in Eq. (1) is uniformly bounded over $\Theta$ by $C(\tau) := 4C_s(\tau)C_\Theta$. $\qquad\square$

### B.1.2. MULTIMODAL OPTIMIZATION STABILITY

**Lemma B.2** (Stable multimodal optimization). *Assume Asm. 4.1. Fix any $\tau > 0$ and any $N \in \mathbb{N}_+$. Then the symmetric multimodal loss $\mathcal{L}_{\mathrm{Sym}}(\theta, \phi)$ defined in Eq. (2) has uniformly bounded joint gradient over $\Theta \times \Phi$. Formally, there exists a constant $C(\tau) < \infty$ such that*

$$\|\nabla_{(\theta, \phi)} \mathcal{L}_{\mathrm{Sym}}(\theta, \phi)\| \leq C(\tau), \qquad \forall (\theta, \phi) \in \Theta \times \Phi.$$

☞ **Proof intuition.** This proof extends the uniform gradient bound established for the unimodal case to the dual-encoder, cross-modal setting. The core logic inherits the properties of *atomic smoothness* and the *convex bottleneck* detailed previously, applying them to the joint parameter space $(\Theta, \Phi)$. The extension relies on two structural observations: **(i)** When evaluating a single contrastive direction (e.g., matching a vision anchor to language keys), the partial gradients with respect to either the anchor network ($\theta$) or the key network ($\phi$) behave exactly like the unimodal case. The softmax-weighted denominator ensures the gradient for each individual modality remains strictly bounded. **(ii)** The total joint gradient across both modalities is the vector concatenation of the individual $\theta$ and $\phi$ updates. By treating these parameter spaces orthogonally, we can apply the Pythagorean theorem to fuse their separate bounds. Because the symmetric loss is simply the average of two bounded directional losses, the triangle inequality yields one absolute, uniform upper bound for the entire cross-modal objective.

*Proof.* **Step 1: Uniform bounds on components.** We retain the bound on the similarity critic sensitivity $C_s(\tau)$ from Eq. (14) in unimodal derivations. For the encoders, we define the uniform bound on the Jacobians over the joint space, which is guaranteed by Asm. 4.1. Let $C_\Theta$ and $C_\Phi$ be the maximal sensitivities:

$$C_\Theta := \sup_{\theta \in \Theta} \sup_{\mathbf{x} \in \mathcal{X}} \|J_\theta f_\theta(\mathbf{x})\| < \infty, \quad C_\Phi := \sup_{\phi \in \Phi} \sup_{\mathbf{y} \in \mathcal{Y}} \|J_\phi g_\phi(\mathbf{y})\| < \infty.$$

We define the global encoder constant $C_{\text{enc}} := \max(C_\Theta, C_\Phi)$. Consider a generic cross-modal representation pair $(\mathbf{z}, \mathbf{w})$ where $\mathbf{z}$ depends on $\theta$ and $\mathbf{w}$ depends on $\phi$. The gradient of the kernel term $\kappa_\tau(\mathbf{z}, \mathbf{w})$ with respect to the joint parameter vector $(\theta, \phi)$ splits into block components. Using Eq. (16), the partial gradients satisfy:

$$\|\nabla_\theta \log \kappa_\tau(\mathbf{z}, \mathbf{w})\| \le C_s(\tau) C_\Theta \le C_s(\tau) C_{\text{enc}}, \quad \|\nabla_\phi \log \kappa_\tau(\mathbf{z}, \mathbf{w})\| \le C_s(\tau) C_\Phi \le C_s(\tau) C_{\text{enc}}. \quad (20)$$

**Step 2: Bounding directional partial derivatives.** Define a training batch $\mathcal{B}$ as a collection of $N + 1$ independent pairs: a primary positive pair $(\mathbf{x}, \mathbf{y})$ and $N$ negative keys $\{\mathbf{y}'_j\}_{j=1}^N$ (and similarly in reverse). The symmetric loss is the average of two directional losses computed on this batch:

$$\ell_{\text{Sym}}(\mathcal{B}) = \tfrac{1}{2}\big(\ell_{\theta \to \phi}(\mathcal{B}) + \ell_{\phi \to \theta}(\mathcal{B})\big).$$

By the triangle inequality, the norm of the joint gradient satisfies:

$$\|\nabla_{(\theta, \phi)} \ell_{\text{Sym}}\| \le \tfrac{1}{2} \|\nabla_{(\theta, \phi)} \ell_{\theta \to \phi}\| + \tfrac{1}{2} \|\nabla_{(\theta, \phi)} \ell_{\phi \to \theta}\|.$$

We first analyze the forward component $\ell_{\theta \to \phi}$. Here, $\mathbf{x}$ serves as the anchor, $\mathbf{y}$ as the positive key, and the set $\{\mathbf{y}'_j\}_{j=1}^N$ as the negative keys. The loss is:

$$\ell_{\theta \to \phi} := -\log \kappa_\tau(f_\theta(\mathbf{x}), g_\phi(\mathbf{y})) + \log\Big(\kappa_\tau(f_\theta(\mathbf{x}), g_\phi(\mathbf{y})) + \sum_{j=1}^N \kappa_\tau(f_\theta(\mathbf{x}), g_\phi(\mathbf{y}'_j))\Big).$$

Crucially, the joint gradient vector is the concatenation $\nabla_{(\theta, \phi)} \ell_{\theta \to \phi} = (\nabla_\theta \ell_{\theta \to \phi}^\top, \nabla_\phi \ell_{\theta \to \phi}^\top)^\top$. We bound the partial gradients of $\ell_{\theta \to \phi}$ with respect to each parameter block separately. Note that the set of key representations in the batch is $\mathcal{C}_\phi = \{g_\phi(\mathbf{y})\} \cup \{g_\phi(\mathbf{y}'_j)\}_{j=1}^N$, and we denote the softmax probability assigned to any key $\mathbf{v} \in \mathcal{C}_\phi$ as $\sigma(\mathbf{v})$.

(i) Gradient with respect to $\theta$ (anchor sensitivity): In this direction, $\theta$ governs only the anchor $f_\theta(\mathbf{x})$; the keys $\mathcal{C}_\phi$ are constant. We expand the gradient of the loss as the alignment term plus the gradient of the log-partition function:

$$\nabla_\theta \ell_{\theta \to \phi} = -\nabla_\theta \log \kappa_\tau\big(f_\theta(\mathbf{x}), g_\phi(\mathbf{y})\big) + \sum_{\mathbf{v} \in \mathcal{C}_\phi} \sigma(\mathbf{v}) \nabla_\theta \log \kappa_\tau\big(f_\theta(\mathbf{x}), \mathbf{v}\big).$$

Applying the triangle inequality and the uniform sensitivity bound $C_s(\tau) C_\Theta$ derived in Eq. (20):

$$\|\nabla_\theta \ell_{\theta \to \phi}\| \le \|\nabla_\theta \log \kappa_\tau\big(f_\theta(\mathbf{x}), g_\phi(\mathbf{y})\big)\| + \sum_{\mathbf{v} \in \mathcal{C}_\phi} \sigma(\mathbf{v}) \|\nabla_\theta \log \kappa_\tau\big(f_\theta(\mathbf{x}), \mathbf{v}\big)\|$$

$$\le C_s(\tau) C_\Theta + C_s(\tau) C_\Theta \sum_{\mathbf{v} \in \mathcal{C}_\phi} \sigma(\mathbf{v}) = 2C_s(\tau) C_\Theta. \qquad \Big(\text{Since} \sum_{\mathbf{v} \in \mathcal{C}_\phi} \sigma(\mathbf{v}) = 1\Big)$$

(ii) Gradient with respect to $\phi$ (key sensitivity): In this direction, $\phi$ parametrizes every key in the batch (both positive and negative), while the anchor $f_\theta(\mathbf{x})$ acts as a constant. The gradient is:

$$\nabla_\phi \ell_{\theta \to \phi} = -\nabla_\phi \log \kappa_\tau\big(f_\theta(\mathbf{x}), g_\phi(\mathbf{y})\big) + \sum_{\mathbf{v} \in \mathcal{C}_\phi} \sigma(\mathbf{v}) \nabla_\phi \log \kappa_\tau\big(f_\theta(\mathbf{x}), \mathbf{v}\big).$$

Similarly, applying the triangle inequality and the sensitivity bound $C_s(\tau)C_\Phi$:

$$\|\nabla_\phi \ell_{\theta \to \phi}\| \leq C_s(\tau)C_\Phi + C_s(\tau)C_\Phi \sum_{\mathbf{v} \in \mathcal{C}_\phi} \sigma(\mathbf{v}) = 2C_s(\tau)C_\Phi.$$

**Step 3: Global uniform bound.** Based on the above analysis, the total gradient magnitude for the forward loss is bounded by the vector norm:

$$\|\nabla_{(\theta,\phi)} \ell_{\theta \to \phi}\| = \sqrt{\|\nabla_\theta \ell_{\theta \to \phi}\|^2 + \|\nabla_\phi \ell_{\theta \to \phi}\|^2} \leq \sqrt{(2C_s(\tau)C_\Theta)^2 + (2C_s(\tau)C_\Phi)^2} \leq 2\sqrt{2}C_s(\tau)C_{\text{enc}}.$$

By symmetry, the exact same bound applies to the backward loss $\ell_{\phi \to \theta}$. Therefore, the batch loss gradient is bounded by:

$$\|\nabla_{(\theta,\phi)} \ell_{\text{Sym}}\| = \tfrac{1}{2}\|\nabla_{(\theta,\phi)} \ell_{\theta \to \phi} + \nabla_{(\theta,\phi)} \ell_{\phi \to \theta}\| \leq \tfrac{1}{2}(\|\nabla_{(\theta,\phi)} \ell_{\theta \to \phi}\| + \|\nabla_{(\theta,\phi)} \ell_{\phi \to \theta}\|) = 2\sqrt{2}C_s(\tau)C_{\text{enc}}.$$

Since $\|\nabla_{(\theta,\phi)} \ell_{\text{Sym}}(\mathcal{B})\|$ is uniformly bounded and $\ell_{\text{Sym}}$ is differentiable a.e. in $(\theta, \phi)$, we may differentiate under the expectation by dominated convergence, yielding

$$\|\nabla_{(\theta,\phi)} \mathcal{L}_{\text{Sym}}\| = \|\nabla_{\theta,\phi} \mathbb{E}_{\mathcal{B}}[\ell_{\text{Sym}}]\| \leq \mathbb{E}_{\mathcal{B}}[\|\nabla \ell_{\text{Sym}}\|] \leq 2\sqrt{2}C_s(\tau)C_{\text{enc}}.$$

Defining $C := 2\sqrt{2}C_s(\tau)C_{\text{enc}} < \infty$, the gradient is uniformly bounded over $\Theta \times \Phi$. $\qquad\square$

### B.2. Sharp-peak Smoothing Implies Vanishing KDE Mismatch

The large-batch parametric energy $\mathcal{J}_\tau(\theta)$ depends on the *kernel-smoothed* density $\tilde{\rho}_{\theta,\tau}$ through the cross-entropy term $H_\times(q_\theta, \tilde{\rho}_{\theta,\tau})$, whereas the intrinsic functional $\mathcal{F}_{\tau,U_\theta}$ is expressed in terms of the *true* density $\rho_\theta$ and its entropy $H(\rho_\theta)$. To compare $\mathcal{J}_\tau(\theta)$ with $\mathcal{F}_{\tau,U_\theta}(\rho_\theta)$ in Thm. 3.2, we therefore require a quantitative control of the bias introduced by smoothing, i.e., that $\tilde{\rho}_{\theta,\tau}$ approximates $\rho_\theta$ when the kernel is sufficiently sharp.

**Lemma B.3** (Vanishing KDE error under sharp peak). *Assume Asms. 2.1, 3.1 and 3.2. Fix $\theta \in \Theta$ and assume $q_\theta \ll \mu$ with a continuous density $\rho_\theta$ bounded away from zero, i.e. $\inf_{\mathbf{z} \in \mathcal{Z}} \rho_\theta(\mathbf{z}) > \underline{\rho}_\theta > 0$. Define the KDE mismatch $\varepsilon_{\text{kde}}^{(\theta)}(\tau) := \|\tilde{\rho}_{\theta,\tau} - \rho_\theta\|_\infty$, where $\tilde{\rho}_{\theta,\tau}$ is the kernel-smoothed density (Def. 2.2). Then, as $\tau \to 0^+$, $\varepsilon_{\text{kde}}^{(\theta)}(\tau) \to 0$.*

☞ **Proof intuition.** The core objective is to establish that as the temperature $\tau$ approaches zero, the kernel-smoothed density $\tilde{\rho}_{\theta,\tau}$ converges to the true continuous density $\rho_\theta$, uniformly across the entire manifold $\mathcal{Z}$. The intuition relies on two geometric properties: **(i)** First, consider how the temperature parameter scales the kernel. The similarity critic is defined to have a sharp, unique peak when a representation is compared to itself. As $\tau \to 0^+$, the exponential nature of the kernel drastically exaggerates this peak, effectively morphing the kernel into a Dirac delta function. The probability mass assigned to any representation even slightly separated from the target decays exponentially fast. Simultaneously, the compact geometry of the manifold guarantees there is always a stable, non-vanishing volume of space immediately surrounding the target, ensuring the local density does not collapse to zero. **(ii)** Second, because the true density $\rho_\theta$ is continuous on a compact manifold, it is uniformly continuous—it cannot exhibit infinitely steep cliffs or discontinuous jumps. At near-zero temperatures, the smoothed density $\tilde{\rho}_{\theta,\tau}(\mathbf{z})$ evaluates a weighted average using almost exclusively the points infinitesimally close to $\mathbf{z}$. Within this microscopic neighborhood, the true density is essentially flat. The distant regions of the manifold, where the true density might differ wildly, are completely muted by the vanishing tail of the kernel. Consequently, the local weighted average approximates the true local density, driving the maximum global estimation error to zero.

*Proof.* Fix any $\theta \in \Theta$. Let $d_\mathcal{Z}(\cdot, \cdot)$ denote the geodesic distance on the compact Riemannian manifold $\mathcal{Z}$ as in Asm. 3.2. Since $\mathcal{Z}$ is compact and $\rho_\theta$ is continuous, $\rho_\theta$ is bounded. Define the normalized kernel

$$\bar{\kappa}_\tau(\mathbf{z}, \mathbf{w}) := \frac{\kappa_\tau(\mathbf{z}, \mathbf{w})}{V_\kappa(\tau)}.$$

We first show that $\bar{\kappa}_\tau$ concentrates near the diagonal $\{\mathbf{z} = \mathbf{w}\}$ as $\tau \to 0^+$, and then prove Lem. B.3.

**Step 1: Uniform concentration of $\bar{\kappa}_\tau$ near the diagonal.** Fix any $\delta \in (0, r)$, where $r$ is from Asm. 3.2, and define

$$\mathcal{A}_\delta := \{(\mathbf{z}, \mathbf{w}) \in \mathcal{Z} \times \mathcal{Z} : d_{\mathcal{Z}}(\mathbf{z}, \mathbf{w}) \geq \delta\}.$$

By compactness of $\mathcal{Z}$, the set $\mathcal{A}_\delta$ is compact. Since $s$ is continuous by Asm. 3.1, consider the continuous function

$$g(\mathbf{z}, \mathbf{w}) := s(\mathbf{z}, \mathbf{z}) - s(\mathbf{z}, \mathbf{w}).$$

By Asm. 3.2, for each fixed $\mathbf{z} \in \mathcal{Z}$ the map $\mathbf{w} \mapsto s(\mathbf{z}, \mathbf{w})$ has a unique maximizer at $\mathbf{w} = \mathbf{z}$; hence $g(\mathbf{z}, \mathbf{w}) > 0$ whenever $d_{\mathcal{Z}}(\mathbf{z}, \mathbf{w}) \geq \delta$. Therefore $g$ is strictly positive on $\mathcal{A}_\delta$, and by compactness it attains a strictly positive minimum:

$$m_\delta := \min_{(\mathbf{z}, \mathbf{w}) \in \mathcal{A}_\delta} g(\mathbf{z}, \mathbf{w}) > 0.$$

Equivalently, for all $\mathbf{z} \in \mathcal{Z}$ and all $\mathbf{w} \in \mathcal{Z}$ with $d_{\mathcal{Z}}(\mathbf{z}, \mathbf{w}) \geq \delta$,

$$s(\mathbf{z}, \mathbf{w}) \leq s(\mathbf{z}, \mathbf{z}) - m_\delta. \tag{21}$$

Fix $\eta \in (0, \delta)$. Let $\mathbb{B}_l(\mathbf{z}) := \{\mathbf{w} \in \mathcal{Z} \mid d_{\mathcal{Z}}(\mathbf{z}, \mathbf{w}) < l\}$ denote an open ball with radius $l$. By the compactness of $\mathcal{Z}$, any open cover of $\mathcal{Z}$ has a finite subcover. Consider the open cover given by $\{\mathbb{B}_{\eta/2}(\mathbf{z}) \mid \mathbf{z} \in \mathcal{Z}\}$. We can extract a finite subcover, meaning there exist points $\{\mathbf{z}_1, \ldots, \mathbf{z}_M\} \subset \mathcal{Z}$ such that

$$\mathcal{Z} = \bigcup_{j=1}^{M} \mathbb{B}_{\eta/2}(\mathbf{z}_j).$$

For any arbitrary point $\mathbf{z} \in \mathcal{Z}$, the properties of the cover guarantee there exists at least one index $i \in \{1, \ldots, M\}$ such that $\mathbf{z} \in \mathbb{B}_{\eta/2}(\mathbf{z}_i)$. Now, choose any point $\mathbf{w} \in \mathbb{B}_{\eta/2}(\mathbf{z}_i)$. By the triangle inequality

$$d_{\mathcal{Z}}(\mathbf{z}, \mathbf{w}) \leq d_{\mathcal{Z}}(\mathbf{z}_i, \mathbf{w}) + d_{\mathcal{Z}}(\mathbf{z}_i, \mathbf{z}).$$

Since both $\mathbf{z}$ and $\mathbf{w}$ are points of the open ball $\mathbb{B}_{\eta/2}(\mathbf{z}_i)$, we have strict inequalities $d_{\mathcal{Z}}(\mathbf{z}_i, \mathbf{w}) < \eta/2$ and $d_{\mathcal{Z}}(\mathbf{z}_i, \mathbf{z}) < \eta/2$. Combining this with the triangle inequality, we have:

$$d_{\mathcal{Z}}(\mathbf{z}, \mathbf{w}) < \frac{\eta}{2} + \frac{\eta}{2} = \eta \implies \mathbf{w} \in \mathbb{B}_\eta(\mathbf{z}).$$

This establishes $\mathbb{B}_{\eta/2}(\mathbf{z}_i) \subseteq \mathbb{B}_\eta(\mathbf{z})$. By the monotonicity of measures, applying $\mu$ yields

$$\mu\big(\mathbb{B}_\eta(\mathbf{z})\big) \geq \mu\big(\mathbb{B}_{\eta/2}(\mathbf{z}_i)\big).$$

Since this holds for the specific $i$ associated with the arbitrary point $\mathbf{z}$, the measure is bounded below by the minimum across all $M$ patches:

$$\mu\big(\mathbb{B}_\eta(\mathbf{z})\big) \geq \min_{1 \leq j \leq M} \mu\big(\mathbb{B}_{\eta/2}(\mathbf{z}_j)\big) =: b_\eta > 0.$$

where positivity $b_\eta > 0$ holds because $\mu$ has full support on the manifold.

Now, let $\delta \in (0, r)$ be as above and choose $\eta \in (0, \delta)$ so that

$$m_2 \eta^2 < m_\delta, \tag{22}$$

where $m_2$ is the constant in Asm. 3.2. This is possible since $m_\delta > 0$. Then, for $\mathbf{w} \in \mathbb{B}_\eta(\mathbf{z}) \subset \mathbb{B}_r(\mathbf{z})$, the lower quadratic bound in Asm. 3.2 gives

$$s(\mathbf{z}, \mathbf{w}) \geq s(\mathbf{z}, \mathbf{z}) - m_2 d_{\mathcal{Z}}(\mathbf{z}, \mathbf{w})^2 \geq s(\mathbf{z}, \mathbf{z}) - m_2 \eta^2,$$

so

$$\kappa_\tau(\mathbf{z}, \mathbf{w}) \geq \exp\big(s(\mathbf{z}, \mathbf{z})/\tau\big) \exp\big(-m_2 \eta^2/\tau\big).$$

Integrating over $\mathbb{B}_\eta(\mathbf{z})$ and using $\mu(\mathbb{B}_\eta(\mathbf{z})) \geq b_\eta$ yields the uniform lower bound

$$V_\kappa(\tau) = \int_{\mathcal{Z}} \kappa_\tau(\mathbf{z}, \mathbf{w}) \, \mathrm{d}\mu(\mathbf{w}) \ \geq \ b_\eta \, \exp\big(s(\mathbf{z}, \mathbf{z})/\tau\big) \exp\big(- m_2 \, \eta^2/\tau\big). \tag{23}$$

Here, by Asm. 2.1, $V_\kappa(\tau)$ is independent of $\mathbf{z}$. (The factor $\exp(s(\mathbf{z}, \mathbf{z})/\tau)$ cancels in the ratio below.)

Next, for $\mathbf{w} \notin \mathbb{B}_\delta(\mathbf{z})$ we have $d_{\mathcal{Z}}(\mathbf{z}, \mathbf{w}) \geq \delta$, so by Eq. (21),

$$\kappa_\tau(\mathbf{z}, \mathbf{w}) = \exp\big(s(\mathbf{z}, \mathbf{w})/\tau\big) \ \leq \ \exp\big(s(\mathbf{z}, \mathbf{z})/\tau\big) \exp\big(- m_\delta/\tau\big).$$

Therefore, with $V_\mu := \mu(\mathcal{Z}) < \infty$,

$$\int_{\mathcal{Z} \backslash \mathbb{B}_\delta(\mathbf{z})} \kappa_\tau(\mathbf{z}, \mathbf{w}) \, \mathrm{d}\mu(\mathbf{w}) \ \leq \ V_\mu \, \exp\big(s(\mathbf{z}, \mathbf{z})/\tau\big) \exp\big(- m_\delta/\tau\big). \tag{24}$$

Dividing Eq. (24) by Eq. (23) and using Eq. (22) yields, uniformly in $\mathbf{z}$,

$$\int_{\mathcal{Z} \backslash \mathbb{B}_\delta(\mathbf{z})} \bar{\kappa}_\tau(\mathbf{z}, \mathbf{w}) \, \mathrm{d}\mu(\mathbf{w}) \ \leq \ \frac{V_\mu}{b_\eta} \exp\Big(- \frac{m_\delta - m_2 \, \eta^2}{\tau}\Big) \ \xrightarrow{\tau \to 0^+} \ 0. \tag{25}$$

**Step 2: Sup-norm convergence of the smoothing.** Fix $\varepsilon > 0$. Since $\rho_\theta$ is continuous on compact $\mathcal{Z}$, it is uniformly continuous. By uniform continuity in $\mathbf{z}$ of $\rho_\theta$ for each fixed $\theta \in \Theta$, choose $\delta \in (0, r)$ such that

$$|\rho_\theta(\mathbf{w}) - \rho_\theta(\mathbf{z})| \leq \varepsilon \qquad \text{whenever} \qquad d_{\mathcal{Z}}(\mathbf{z}, \mathbf{w}) < \delta.$$

For any $\mathbf{z} \in \mathcal{Z}$, write

$$\tilde{\rho}_{\theta,\tau}(\mathbf{z}) - \rho_\theta(\mathbf{z}) = \int_{\mathcal{Z}} \bar{\kappa}_\tau(\mathbf{z}, \mathbf{w}) \big(\rho_\theta(\mathbf{w}) - \rho_\theta(\mathbf{z})\big) \, \mathrm{d}\mu(\mathbf{w}),$$

and split the integral over $\mathbb{B}_\delta(\mathbf{z})$ and its complement. Using $\int_{\mathcal{Z}} \bar{\kappa}_\tau(\mathbf{z}, \mathbf{w}) \, \mathrm{d}\mu(\mathbf{w}) = 1$, the bound on $\mathbb{B}_\delta(\mathbf{z})$, and $|\rho_\theta(\mathbf{w}) - \rho_\theta(\mathbf{z})| \leq 2\|\rho_\theta\|_\infty$ everywhere, we obtain

$$|\tilde{\rho}_{\theta,\tau}(\mathbf{z}) - \rho_\theta(\mathbf{z})| \leq \varepsilon + 2\|\rho_\theta\|_\infty \int_{\mathcal{Z} \backslash \mathbb{B}_\delta(\mathbf{z})} \bar{\kappa}_\tau(\mathbf{z}, \mathbf{w}) \, \mathrm{d}\mu(\mathbf{w}).$$

By the boundedness of $\rho_\theta$, taking the supremum over $\mathbf{z} \in \mathcal{Z}$ and invoking Eq. (25) gives

$$\limsup_{\tau \to 0^+} \ \|\tilde{\rho}_{\theta,\tau} - \rho_\theta\|_\infty \leq \varepsilon.$$

Since $\varepsilon > 0$ is arbitrary, and by the non-negativity of norm, $\liminf_{\tau \to 0^+} \ \|\tilde{\rho}_{\theta,\tau} - \rho_\theta\|_\infty \geq 0$. Therefore, for each fixed $\theta \in \Theta$,

$$\varepsilon_{\mathrm{kde}}^{(\theta)}(\tau) := \|\tilde{\rho}_{\theta,\tau} - \rho_\theta\|_\infty \xrightarrow{\tau \to 0^+} 0$$

$\square$

### B.3. Unimodal Parametric Inheritance and Emergence of Uniformity

This subsection formalizes the intuition used in § 3.2: unimodal InfoNCE inherits the single-well intrinsic geometry when (i) the kernel smoothing bias is small and (ii) the encoder family is expressive enough to (approximately) realize the intrinsic Gibbs equilibrium at the best attainable potential level.

**Recap of objects.** Fix $\tau > 0$. For each $\theta \in \Theta$, recall the induced representation law $q_\theta \in \mathcal{P}(\mathcal{Z})$ and encoded positive-pair law $\pi_{\theta\theta} \in \mathcal{P}(\mathcal{Z} \times \mathcal{Z})$. Under the same setting of Thm. 3.2, assume $q_\theta \ll \mu$ and write $\rho_\theta := \mathrm{d}q_\theta/\mathrm{d}\mu$. Let $U_\theta$ be the alignment potential field induced by $\pi_{\theta\theta}$ (Def. 3.1). The intrinsic functional is $\mathcal{F}_{\tau, U_\theta}(\rho)$ (Eq. (7)), and the parametric energy is $\mathcal{J}_\tau(\theta)$ (Def. 3.2).

**Free-energy gap equals KL divergence.** From Prop. 3.1, for any fixed potential $U$ and temperature $\tau$, define the partition function

$$Z_\tau := \int_{\mathcal{Z}} \exp\big(-U(\mathbf{z})/\tau\big)\, \mathrm{d}\mu(\mathbf{z}), \tag{26}$$

and the associated Gibbs density

$$\rho^*(\mathbf{z}) := \frac{\exp\big(-U(\mathbf{z})/\tau\big)}{Z_\tau}. \tag{27}$$

Then for any $\rho \in \mathcal{P}_\mu(\mathcal{Z})$ with $\rho \log(\rho/\rho^*) \in L^1(\mu)$,

$$\mathcal{F}_{\tau,U}(\rho) + \log Z_\tau = D_{\mathrm{KL}}\big(\rho \,\|\, \rho^*\big) \ \geq\ 0. \tag{28}$$

In particular, $\rho^*$ is the unique minimizer of $\mathcal{F}_{\tau,U}$ and the free-energy suboptimality controls proximity to the Gibbs equilibrium via KL divergence.

*Remark* B.1 (Proof of Eq. (28)). Expanding $D_{\mathrm{KL}}(\rho\|\rho^*) = \int_{\mathcal{Z}} \rho \log(\rho/\rho^*)\, \mathrm{d}\mu$ and substituting $\log \rho^* = -(U/\tau) - \log Z_\tau$ yields Eq. (28).

**Expressivity at the best attainable potential level.** The intrinsic minimum of $\mathcal{F}_{\tau,U_\theta}$ equals $-\log Z_{\theta,\tau}$ by Eq. (28), where $Z_{\theta,\tau} = \int_{\mathcal{Z}} \exp(-U_\theta(\mathbf{z})/\tau)\mathrm{d}\mu(\mathbf{z})$ is the normalization constant of the Gibbs density. We therefore define the *best attainable potential level* at temperature $\tau$ as any

$$\hat{\theta} \in \arg\min_{\theta \in \Theta} \Big(-\log Z_{\theta,\tau}\Big). \tag{29}$$

**Assumption B.1** (Expressivity at the best level). There exists a function $\varepsilon_{\mathrm{real}}(\tau) \to 0$ as $\tau \to 0^+$ such that for $\hat{\theta}$ in Eq. (29),

$$\mathcal{F}_{\tau,U_{\hat{\theta}}}(\rho_{\hat{\theta}}) + \log Z_{\hat{\theta},\tau} \leq \varepsilon_{\mathrm{real}}(\tau). \tag{30}$$

Equivalently, at the best attainable potential level, the encoder-induced density $\rho_{\hat{\theta}}$ attains the intrinsic minimum of $\mathcal{F}_{\tau,U_{\hat{\theta}}}$ up to error $\varepsilon_{\mathrm{real}}(\tau)$.

**Ground-state inheritance under encoder expressivity.** Under Asm. 3.2 and mild regularity conditions, Thm. 3.2 bounds the gap between $\mathcal{J}_\tau(\theta)$ and $\mathcal{F}_{\tau,U_\theta}(\rho_\theta)$ by the KDE mismatch $\varepsilon_{\mathrm{kde}}^{(\theta)}(\tau)$. By Lem. B.3, this mismatch vanishes for each fixed $\theta$. Since the relevant optimizers may depend on $\tau$, we state the following inheritance result along optimizing sequences for which the corresponding KDE mismatches vanish at the evaluated temperatures:

**Corollary B.1** (Unimodal parametric inheritance along optimizing sequences). *For each $\tau > 0$, let $\theta_\tau^* \in \arg\min_{\theta \in \Theta} \mathcal{J}_\tau(\theta)$ and $\hat{\theta}_\tau \in \arg\min_{\theta \in \Theta}\big(-\log Z_{\theta,\tau}\big)$. Assume the conditions of Thm. 3.2 hold for $\theta_\tau^*$ and $\hat{\theta}_\tau$ for all sufficiently small $\tau > 0$. Assume additionally that the expressivity condition in Asm. B.1 holds along the sequence $\hat{\theta}_\tau$, and that the KDE mismatch of energies satisfies*

$$\eta_\tau(\theta_\tau^*) + \eta_\tau(\hat{\theta}_\tau) \to 0 \quad as\ \tau \to 0^+, \qquad where\ \eta_\tau(\theta) := \big|\mathcal{J}_\tau(\theta) - \mathcal{F}_{\tau,U_\theta}(\rho_\theta)\big|.$$

*Further, let $\rho_{\theta_\tau^*}^{*;\tau}$ denote the Gibbs density associated with $U_{\theta_\tau^*}$ at temperature $\tau$:*

$$\rho_{\theta_\tau^*}^{*,\tau}(\mathbf{z}) := \frac{\exp\big(-U_{\theta_\tau^*}(\mathbf{z})/\tau\big)}{Z_{\theta_\tau^*,\tau}}, \qquad Z_{\theta_\tau^*,\tau} := \int_{\mathcal{Z}} \exp\big(-U_{\theta_\tau^*}(\mathbf{z}')/\tau\big)\, \mathrm{d}\mu(\mathbf{z}').$$

*Then, $D_{\mathrm{KL}}\big(\rho_{\theta_\tau^*} \,\|\, \rho_{\theta_\tau^*}^{*;\tau}\big) \to 0$ and $\big\|\rho_{\theta_\tau^*} - \rho_{\theta_\tau^*}^{*;\tau}\big\|_{\mathrm{TV}} \to 0$. Consequently, the learned representation measures inherit any low-temperature concentration property of the associated Gibbs sequence. In particular, if for some fixed $\sigma > 0$ and some family of open sets $\mathcal{O}_\tau \subset \mathcal{Z}$ satisfying*

$$\mathcal{O}_\tau \supset \mathcal{W}_{\theta_\tau^*}^\sigma := \big\{\mathbf{z} \in \mathcal{Z} : U_{\theta_\tau^*}(\mathbf{z}) \leq \operatorname{ess\,inf}_\mu U_{\theta_\tau^*} + \sigma\big\},$$

*the Gibbs sequence satisfies $(\rho_{\theta_\tau^*}^{*;\tau}\mu)(\mathcal{Z} \setminus \mathcal{O}_\tau) \to 0$, then $q_{\theta_\tau^*}(\mathcal{Z} \setminus \mathcal{O}_\tau) \to 0$.*

☞ **Proof intuition.** The core objective is to show that a fully optimized encoder sequence $\theta_\tau^*$ tracks the Gibbs equilibrium induced by its own alignment potential as $\tau \to 0^+$. The proof has three steps. **(i)** For any fixed induced potential field $U_\theta$, the intrinsic free-energy gap equals the KL divergence from the corresponding Gibbs density. **(ii)** By the expressivity condition, the comparator $\hat{\theta}_\tau$ can approach the intrinsic optimum at the best attainable potential level. Since $\theta_\tau^*$ minimizes the parametric energy $\mathcal{J}_\tau$, and since the sequence-level energy approximation errors vanish for both $\theta_\tau^*$ and $\hat{\theta}_\tau$, the intrinsic free-energy gap of $\rho_{\theta_\tau^*}$ also vanishes. Hence $D_{\mathrm{KL}}(\rho_{\theta_\tau^*} \| \rho_{\theta_\tau^*}^{*,\tau}) \to 0$. **(iii)** Pinsker's inequality gives convergence in total variation, so any low-temperature concentration property of the Gibbs sequence transfers to the learned representation measures.

*Proof.* **Step 1: Well-defined "best achievable" equilibrium.** We first isolate the intrinsic quantity that governs the "best achievable" equilibrium value at temperature $\tau$. Fix any $\theta \in \Theta$ and $\tau > 0$. By Asm. 2.1, $\mathcal{Z}$ is compact with $\mu(\mathcal{Z}) < \infty$; and by Asm. 3.1, $s : \mathcal{Z} \times \mathcal{Z} \to \mathbb{R}$ is continuous. Then, by the extreme value theorem, $s$ is bounded on $\mathcal{Z} \times \mathcal{Z}$:

$$S_{\max} := \max_{\mathbf{z}, \mathbf{w} \in \mathcal{Z}} s(\mathbf{z}, \mathbf{w}) < \infty.$$

The induced alignment potential field (Def. 3.1) is $U_\theta(\mathbf{z}) = - \int_{\mathcal{Z}} s(\mathbf{z}, \mathbf{w}) \, d\nu_{\theta, \mathbf{z}}(\mathbf{w})$, hence $U_\theta(\mathbf{z}) \geq -S_{\max}$ for all $\mathbf{z}$. Therefore, for any fixed $\tau > 0$,

$$0 < \exp\left( -\frac{U_\theta(\mathbf{z})}{\tau} \right) \leq \exp\left( \frac{S_{\max}}{\tau} \right) \quad \text{for all } \mathbf{z} \in \mathcal{Z},$$

and thus

$$0 < Z_{\theta, \tau} := \int_{\mathcal{Z}} \exp\left( -\frac{U_\theta(\mathbf{z})}{\tau} \right) d\mu(\mathbf{z}) \leq \mu(\mathcal{Z}) \exp\left( \frac{S_{\max}}{\tau} \right) < \infty.$$

Thus, the representational energy $\mathcal{F}_{\tau, U_\theta}(\rho) = \frac{1}{\tau} \int_{\mathcal{Z}} U_\theta \, \rho \, d\mu - H(\rho)$ admits the unique Gibbs minimizer $\rho_\theta^* \propto e^{-U_\theta/\tau}$ (Prop. 3.1). The corresponding minimum is

$$\inf_{\rho \in \mathcal{P}_\mu(\mathcal{Z})} \mathcal{F}_{\tau, U_\theta}(\rho) = \mathcal{F}_{\tau, U_\theta}(\rho_\theta^*) = -\log Z_{\theta, \tau}, \qquad Z_{\theta, \tau} := \int_{\mathcal{Z}} \exp\left( -\frac{U_\theta(\mathbf{z})}{\tau} \right) d\mu(\mathbf{z}). \tag{31}$$

We interpret $-\log Z_{\theta, \tau}$ as the lowest representation energy level induced by the potential field $U_\theta$, which is the best intrinsic value attainable by *any* density in $\mathcal{P}_\mu(\mathcal{Z})$ once $U_\theta$ is fixed.

**Step 2: Representation energy gap equals a KL-divergence.** Fix $\theta$ and $\tau$. Recall that, by Prop. 3.1, the unique Gibbs equilibrium is

$$\rho_\theta^*(\mathbf{z}) = \frac{\exp\left( -U_\theta(\mathbf{z})/\tau \right)}{Z_{\theta, \tau}},$$

we have the pointwise identity

$$\log \rho_\theta^*(\mathbf{z}) = -\frac{U_\theta(\mathbf{z})}{\tau} - \log Z_{\theta, \tau}.$$

Define the KL divergence $D_{\mathrm{KL}}(\rho_\theta \| \rho_\theta^*)$. Applying Eq. (28) with $\rho = \rho_\theta$ and $\rho^* = \rho_\theta^*$ yields

$$D_{\mathrm{KL}}(\rho_\theta \| \rho_\theta^*) = \mathcal{F}_{\tau, U_\theta}(\rho_\theta) + \log Z_{\theta, \tau}.$$

Finally, by Eq. (31), $\inf_{\rho \in \mathcal{P}_\mu(\mathcal{Z})} \mathcal{F}_{\tau, U_\theta}(\rho) = \mathcal{F}_{\tau, U_\theta}(\rho_\theta^*) = -\log Z_{\theta, \tau}$, hence

$$\mathcal{F}_{\tau, U_\theta}(\rho_\theta) - \inf_{\rho \in \mathcal{P}_\mu(\mathcal{Z})} \mathcal{F}_{\tau, U_\theta}(\rho) = \mathcal{F}_{\tau, U_\theta}(\rho_\theta) + \log Z_{\theta, \tau} = D_{\mathrm{KL}}(\rho_\theta \| \rho_\theta^*). \tag{32}$$

**Step 3: Transfer intrinsic optimality to the optimizing sequence.** For each $\tau > 0$, let

$$\theta_\tau^* \in \arg\min_{\theta \in \Theta} \mathcal{J}_\tau(\theta), \qquad \hat{\theta}_\tau \in \arg\min_{\theta \in \Theta} \left( -\log Z_{\theta, \tau} \right).$$

Define

$$\eta_\tau(\theta) := \left| \mathcal{J}_\tau(\theta) - \mathcal{F}_{\tau, U_\theta}(\rho_\theta) \right|.$$

By the sequence condition in the statement,

$$\eta_\tau(\theta_\tau^*) + \eta_\tau(\hat{\theta}_\tau) \to 0.$$

By optimality, $\mathcal{J}_\tau(\theta_\tau^*) \le \mathcal{J}_\tau(\hat{\theta}_\tau)$, hence

$$
\begin{aligned}
\mathcal{F}_{\tau, U_{\theta_\tau^*}}(\rho_{\theta_\tau^*}) &\le \mathcal{J}_\tau(\theta_\tau^*) + \eta_\tau(\theta_\tau^*) \\
&\le \mathcal{J}_\tau(\hat{\theta}_\tau) + \eta_\tau(\theta_\tau^*) \\
&\le \mathcal{F}_{\tau, U_{\hat{\theta}_\tau}}(\rho_{\hat{\theta}_\tau}) + \eta_\tau(\theta_\tau^*) + \eta_\tau(\hat{\theta}_\tau).
\end{aligned}
$$

Adding $\log Z_{\theta_\tau^*, \tau}$ and using $\log Z_{\theta_\tau^*, \tau} \le \log Z_{\hat{\theta}_\tau, \tau}$ by the definition of $\hat{\theta}_\tau$, we obtain

$$\mathcal{F}_{\tau, U_{\theta_\tau^*}}(\rho_{\theta_\tau^*}) + \log Z_{\theta_\tau^*, \tau} \le \mathcal{F}_{\tau, U_{\hat{\theta}_\tau}}(\rho_{\hat{\theta}_\tau}) + \log Z_{\hat{\theta}_\tau, \tau} + \eta_\tau(\theta_\tau^*) + \eta_\tau(\hat{\theta}_\tau).$$

Using Eq. (30) along the sequence $\hat{\theta}_\tau$ gives

$$\mathcal{F}_{\tau, U_{\theta_\tau^*}}(\rho_{\theta_\tau^*}) + \log Z_{\theta_\tau^*, \tau} \le \varepsilon_{\mathrm{real}}(\tau) + \eta_\tau(\theta_\tau^*) + \eta_\tau(\hat{\theta}_\tau).$$

By Eq. (32), applied at $\theta = \theta_\tau^*$,

$$D_{\mathrm{KL}}\big(\rho_{\theta_\tau^*} \,\|\, \rho_{\theta_\tau^*}^{*,\tau}\big) \le \varepsilon_{\mathrm{real}}(\tau) + \eta_\tau(\theta_\tau^*) + \eta_\tau(\hat{\theta}_\tau) \xrightarrow{\tau \to 0^+} 0.$$

Therefore, by Pinsker's inequality,

$$\big\| \rho_{\theta_\tau^*} - \rho_{\theta_\tau^*}^{*,\tau} \big\|_{\mathrm{TV}} \le \sqrt{\tfrac{1}{2} D_{\mathrm{KL}}\big(\rho_{\theta_\tau^*} \,\|\, \rho_{\theta_\tau^*}^{*,\tau}\big)} \xrightarrow{\tau \to 0^+} 0. \tag{33}$$

Now let $\mathcal{O}_\tau$ be any family of open sets satisfying $\mathcal{O}_\tau \supset \mathcal{W}_{\theta_\tau^*}^\sigma$ and

$$(\rho_{\theta_\tau^*}^{*,\tau} \mu)(\mathcal{Z} \setminus \mathcal{O}_\tau) \to 0.$$

By the variational characterization of total variation, for any measurable $\mathcal{A} \subseteq \mathcal{Z}$,

$$\big| q_{\theta_\tau^*}(\mathcal{A}) - (\rho_{\theta_\tau^*}^{*,\tau} \mu)(\mathcal{A}) \big| \le \big\| \rho_{\theta_\tau^*} - \rho_{\theta_\tau^*}^{*,\tau} \big\|_{\mathrm{TV}}.$$

Taking $\mathcal{A} = \mathcal{Z} \setminus \mathcal{O}_\tau$ gives

$$q_{\theta_\tau^*}(\mathcal{Z} \setminus \mathcal{O}_\tau) \le (\rho_{\theta_\tau^*}^{*,\tau} \mu)(\mathcal{Z} \setminus \mathcal{O}_\tau) + \big\| \rho_{\theta_\tau^*} - \rho_{\theta_\tau^*}^{*,\tau} \big\|_{\mathrm{TV}} \xrightarrow{\tau \to 0^+} 0.$$

This proves the claimed inherited concentration. $\qquad\square$

*Remark* B.2 (Why this implies uniformity is a tie-breaker). Inside the aligned basin $\{\mathbf{z} : U_\theta(\mathbf{z}) \approx \mathrm{ess\,inf}_\mu U_\theta\}$, the potential term in $\mathcal{F}_{\tau, U_\theta}$ is nearly constant. In that regime, minimizing $\mathcal{F}_{\tau, U_\theta}$ reduces (approximately) to maximizing entropy subject to remaining constraints imposed by the encoder family, so the selected configuration is the most internally dispersed distribution compatible with being well-aligned. This is the precise sense in which "uniformity" in unimodal contrastive learning is not a competing global force but an entropic tie-breaker within the alignment basin.

### B.4. Multimodal Parametric Inheritance

This subsection formalizes the sense in which the practical multimodal energy $\mathcal{J}_\tau^{\mathrm{Sym}}(\theta, \phi)$ inherits the intrinsic barrier-driven geometry of $\mathcal{F}_{\tau, \mathbf{U}}^{\mathrm{Sym}}$ when the encoder family is sufficiently expressive. Concretely, fixing one modality (say $\phi$), we show that if the image encoder family can realize (up to a vanishing gap) the intrinsic best response against the peer density, then any parametric minimizer concentrates its *excess* mass near minimizers of the effective barrier potential $V_{\theta|\phi} = \frac{1}{\tau} U_{\theta \to \phi} + \log \rho_\phi$.

**Background-plus-excess decomposition.** Given a density floor $\underline{\rho}_i > 0$ with $\underline{\rho}_i \, \mu(\mathcal{Z}) < 1$, any feasible density can be decomposed as

$$\rho_i(\mathbf{z}) = \underline{\rho}_i + M_{\mathrm{ex}}^{(i)} \bar{\rho}_i(\mathbf{z}), \qquad \text{with } M_{\mathrm{ex}}^{(i)} := 1 - \underline{\rho}_i \, \mu(\mathcal{Z}), \qquad \text{satisfying } \bar{\rho}_i \geq 0, \int_{\mathcal{Z}} \bar{\rho}_i \, \mathrm{d}\mu = 1.$$

Here $\underline{\rho}_i$ plays the role of a fixed background mass and $\bar{\rho}_i = (\rho_i - \underline{\rho}_i)/M_{\mathrm{ex}}^{(i)}$ is the normalized distribution of the remaining excess budget $M_{\mathrm{ex}}^{(i)}$.

**Corollary B.2** (Multimodal parametric inheritance under model expressivity). *Assume the setting of Thm. 4.2. Fix a peer encoder $\phi \in \Phi$ and, for each $\tau > 0$, let $\theta^* \in \arg\min_{\theta \in \Theta} \mathcal{J}_\tau^{\mathrm{Sym}}(\theta, \phi)$. Write the density floors parameterized by $\theta^*$ as $\underline{\rho}_{\theta^*}$ with $\underline{\rho}_{\theta^*} \mu(\mathcal{Z}) < 1$, and define the set of strictly positive $\mu$-densities bounded below by $\underline{\rho}_{\theta^*}$ as*

$$\mathcal{P}_{\mu, \underline{\rho}_{\theta^*}}(\mathcal{Z}) := \{\rho \in L^1(\mathcal{Z}) : \rho \geq \underline{\rho}_{\theta^*} \ \mu\text{-a.e.}, \int_{\mathcal{Z}} \rho \, \mathrm{d}\mu = 1\},$$

*and the excess mass $M_{\mathrm{ex}}^{(\theta^*)} := 1 - \underline{\rho}_{\theta^*} \mu(\mathcal{Z})$. Define the normalized excess density as $\bar{\rho}_{\theta^*}(\mathbf{z}) := (\rho_{\theta^*}(\mathbf{z}) - \underline{\rho}_{\theta^*})/M_{\mathrm{ex}}^{(\theta^*)}$. Further, assume the encoder family $\{f_\theta\}_\Theta$ is sufficiently expressive: $\exists \varepsilon_{\mathrm{real}}(\tau) \to 0$ as $\tau \downarrow 0^+$ such that*

$$\mathcal{F}_{\tau, \mathbf{U}_{\theta^*, \phi}}^{\mathrm{Sym}}(\rho_{\theta^*}, \rho_\phi) \leq \inf_{\rho \in \mathcal{P}_{\mu, \underline{\rho}_{\theta^*}}(\mathcal{Z})} \mathcal{F}_{\tau, \mathbf{U}_{\theta^*, \phi}}^{\mathrm{Sym}}(\rho, \rho_\phi) + \varepsilon_{\mathrm{real}}(\tau). \tag{34}$$

*Let $V_{\theta^*|\phi}(\mathbf{z}) := \frac{1}{\tau} U_{\theta^* \to \phi}(\mathbf{z}) + \log \rho_\phi(\mathbf{z})$ be the effective potential field parameterized by $(\theta^*, \phi)$. For $\sigma > 0$, define $\mathcal{W}_{\theta^*}^\sigma := \{\mathbf{z} \in \mathcal{Z} : V_{\theta^*|\phi}(\mathbf{z}) \leq \operatorname{ess\,inf}_\mu V_{\theta^*|\phi} + \sigma\}$. Then, for every $\sigma > 0$,*

$$\bar{\rho}_{\theta^*} \mu\big(\mathcal{Z} \setminus \mathcal{W}_{\theta^*}^\sigma\big) \leq 2\,\varepsilon_{\mathrm{real}}(\tau)/(\sigma M_{\mathrm{ex}}^{(\theta^*)}). \tag{35}$$

*Moreover, as $\tau \downarrow 0^+$, if the expressivity condition satisfies $\varepsilon_{\mathrm{real}}(\tau)/\sigma \to 0$ for any $\sigma \downarrow 0$, we have $\bar{\rho}_{\theta^*} \mu(\mathcal{Z} \setminus \mathcal{W}_{\theta^*}^\sigma) \to 0$. A symmetric statement holds for optimizing in $\phi$ with $\theta$ fixed.*

> ☞ **Proof intuition.** The proof proceeds by decoupling the induced density $\rho_{\theta^*}$ into a fixed background floor $\underline{\rho}_{\theta^*}$ and a free excess mass $M_{\mathrm{ex}}^{(\theta^*)}$. By the expressivity assumption, the parameterized model can closely approach the true coordinate infimum of the intrinsic functional. We construct a sequence of boundary competitors that concentrate all their excess mass strictly inside the sublevel sets of the effective barrier potential $V_{\theta^*|\phi}$, driving the cross-entropy term toward its absolute lower bound $\log \underline{\rho}_{\theta^*}$. By comparing the energy of the parameterized density against these optimal competitors, we show that any excess mass placed outside the sublevel set $\mathcal{W}_{\theta^*}^\sigma$ incurs a strict linear energy penalty. The expressivity gap $\varepsilon_{\mathrm{real}}(\tau)$ bounds this penalty, directly restricting the fraction of excess mass that can "leak" outside the ground state.

*Proof.* We provide the derivation for the modality $\mathcal{X}$ with parameter $\theta \in \Theta$. The corresponding result for $\phi \in \Phi$ follows by interchanging modalities and notation.

Fix $\phi \in \Phi$ and $\tau > 0$, and let $\theta^* \in \arg\min_{\theta \in \Theta} \mathcal{J}_\tau^{\mathrm{Sym}}(\theta, \phi)$. Since $\rho_{\theta^*} \geq \underline{\rho}_{\theta^*} \ \mu\text{-a.e.}$, $M_{\mathrm{ex}}^{(\theta^*)} = 1 - \underline{\rho}_{\theta^*} \mu(\mathcal{Z}) > 0$, and $\int_{\mathcal{Z}} \rho_{\theta^*} \, \mathrm{d}\mu = 1$, we have $\bar{\rho}_{\theta^*} \geq 0 \ \mu\text{-a.e.}$ and

$$\int_{\mathcal{Z}} \bar{\rho}_{\theta^*} \mathrm{d}\mu = (1 - \underline{\rho}_{\theta^*} \mu(\mathcal{Z}))/M_{\mathrm{ex}}^{(\theta^*)} = 1,$$

hence $\bar{\rho}_{\theta^*} \mu$ is a valid probability measure.

Define the effective potential field and its essential infimum

$$V_{\theta^*|\phi}(\mathbf{z}) = \frac{1}{\tau} U_{\theta^* \to \phi}(\mathbf{z}) + \log \rho_\phi(\mathbf{z}), \qquad v_* := \operatorname{ess\,inf}_\mu V_{\theta^*|\phi},$$

and for $\sigma > 0$ the sublevel set is $\mathcal{W}_{\theta^*}^\sigma = \{\mathbf{z} \in \mathcal{Z} : V_{\theta^*|\phi}(\mathbf{z}) \leq v_* + \sigma\}$.

**Step 1: A quantitative implication of near-optimality.** Consider the coordinate functional with the peer encoder $\phi$ fixed

$$\mathcal{F}_\tau(\rho) := \mathcal{F}_{\tau, \mathbf{U}_{\theta^*, \phi}}^{\mathrm{Sym}}(\rho, \rho_\phi), \qquad \rho \in \mathcal{P}_{\mu, \underline{\rho}_{\theta^*}}(\mathcal{Z}).$$

As made explicit in the proof of Prop. 4.1 by expanding $\mathcal{F}_{\tau,\mathbf{U}_{\theta^*,\phi}}^{\text{Sym}}$ and collecting $\rho$-dependent terms, $\mathcal{F}_\tau$ can be written in the form

$$\mathcal{F}_\tau(\rho) = \tfrac{1}{2} \int_{\mathcal{Z}} V_{\theta^*|\phi}(\mathbf{z})\, \rho(\mathbf{z})\, \mathrm{d}\mu(\mathbf{z}) + \tfrac{1}{2} \int_{\mathcal{Z}} \rho_\phi(\mathbf{z})\, \log \rho(\mathbf{z})\, \mathrm{d}\mu(\mathbf{z}) + \mathrm{const}_\tau(\phi), \tag{36}$$

where $\mathrm{const}_\tau(\phi)$ is independent of $\rho$. By the floor constraint $\rho \geq \underline{\rho}_{\theta^*}$ $\mu$-a.e., the monotonicity of the logarithm, and the fact that $\rho_\phi$ is a probability density ($\int_{\mathcal{Z}} \rho_\phi\, \mathrm{d}\mu = 1$), we have

$$\int_{\mathcal{Z}} \rho_\phi \log \rho\, \mathrm{d}\mu \;\geq\; \int_{\mathcal{Z}} \rho_\phi \log \underline{\rho}_{\theta^*}\, \mathrm{d}\mu \;=\; \log \underline{\rho}_{\theta^*}.$$

Hence, for any $\rho \in \mathcal{P}_{\mu,\underline{\rho}_{\theta^*}}(\mathcal{Z})$,

$$\mathcal{F}_\tau(\rho) \;\geq\; \tfrac{1}{2} \int_{\mathcal{Z}} V_{\theta^*|\phi}\rho\, \mathrm{d}\mu + \tfrac{1}{2} \log \underline{\rho}_{\theta^*} + \mathrm{const}_\tau(\phi). \tag{37}$$

Let

$$\mathcal{F}_\tau^* := \inf_{\rho \in \mathcal{P}_{\mu,\underline{\rho}_{\theta^*}}(\mathcal{Z})} \mathcal{F}_\tau(\rho).$$

Applying Prop. 4.1 (with density floor $\underline{\rho} = \underline{\rho}_{\theta^*}$) to $\mathcal{F}_\tau$ yields, for each $\delta > 0$, a boundary competitor $\rho^{(\delta)} \in \mathcal{P}_{\mu,\underline{\rho}_{\theta^*}}(\mathcal{Z})$ whose excess mass is supported on a shrinking set $\mathcal{S}^\delta \subseteq \mathcal{W}_{\theta^*}^\delta$ with $\mu(\mathcal{S}^\delta) \xrightarrow{\delta \to 0} 0$, such that $\mathcal{F}_\tau(\rho^{(\delta)}) \to \mathcal{F}_\tau^*$ as $\delta \to 0$. Moreover, since $\rho^{(\delta)} = \underline{\rho}_{\theta^*}$ on $\mathcal{Z} \setminus \mathcal{S}^\delta$ and $\rho_\phi$ is continuous on compact $\mathcal{Z}$, we have

$$\begin{aligned}
\int_{\mathcal{Z}} \rho_\phi \log \rho^{(\delta)}\, \mathrm{d}\mu &= \log \underline{\rho}_{\theta^*} + \int_{\mathcal{S}^\delta} \rho_\phi \log\Big(1 + \frac{M_{\mathrm{ex}}^{(\theta^*)}}{\underline{\rho}_{\theta^*}\mu(\mathcal{S}^\delta)}\Big)\, \mathrm{d}\mu \\
&\leq \log \underline{\rho}_{\theta^*} + \|\rho_\phi\|_\infty\, \mu(\mathcal{S}^\delta) \log\Big(1 + \frac{c}{\mu(\mathcal{S}^\delta)}\Big) \\
&= \log \underline{\rho}_{\theta^*} + o_\delta(1),
\end{aligned}$$

where $o_\delta(1) \to 0$ as $\delta \to 0$ because $\mu(\mathcal{S}^\delta) \log(1/\mu(\mathcal{S}^\delta)) \to 0$. In addition, since $\rho^{(\delta)} - \underline{\rho}_{\theta^*}$ places all excess mass inside $\{\mathbf{z} \in \mathcal{Z} : V_{\theta^*|\phi}(\mathbf{z}) \leq v_* + \delta\}$,

$$\begin{aligned}
\int_{\mathcal{Z}} V_{\theta^*|\phi}\, \rho^{(\delta)}\, \mathrm{d}\mu &= \underline{\rho}_{\theta^*} \int_{\mathcal{Z}} V_{\theta^*|\phi}\, \mathrm{d}\mu + M_{\mathrm{ex}}^{(\theta^*)} \int_{\mathcal{Z}} V_{\theta^*|\phi}\, \bar{\rho}^{(\delta)}\, \mathrm{d}\mu \\
&\leq \underline{\rho}_{\theta^*} \int_{\mathcal{Z}} V_{\theta^*|\phi}\, \mathrm{d}\mu + M_{\mathrm{ex}}^{(\theta^*)}(v_* + \delta),
\end{aligned}$$

where $\bar{\rho}^{(\delta)} := (\rho^{(\delta)} - \underline{\rho}_{\theta^*})/M_{\mathrm{ex}}^{(\theta^*)}$.

Plugging these bounds into Eq. (36) and letting $\delta \to 0$ yields the upper bound

$$\mathcal{F}_\tau^* \;\leq\; \tfrac{1}{2}\Big(\underline{\rho}_{\theta^*} \int_{\mathcal{Z}} V_{\theta^*|\phi}\, \mathrm{d}\mu + M_{\mathrm{ex}}^{(\theta^*)}\, v_*\Big) + \tfrac{1}{2} \log \underline{\rho}_{\theta^*} + \mathrm{const}_\tau(\phi). \tag{38}$$

Now apply Eq. (37) to $\rho = \rho_{\theta^*}$ and subtract Eq. (38). Using

$$\int_{\mathcal{Z}} V_{\theta^*|\phi}\rho_{\theta^*}\, \mathrm{d}\mu = \underline{\rho}_{\theta^*} \int_{\mathcal{Z}} V_{\theta^*|\phi}\, \mathrm{d}\mu + M_{\mathrm{ex}}^{(\theta^*)} \int_{\mathcal{Z}} V_{\theta^*|\phi}\, \bar{\rho}_{\theta^*}\, \mathrm{d}\mu.$$

we obtain

$$\mathcal{F}_\tau(\rho_{\theta^*}) - \mathcal{F}_\tau^* \;\geq\; \frac{M_{\mathrm{ex}}^{(\theta^*)}}{2} \int_{\mathcal{Z}} \big(V_{\theta^*|\phi}(\mathbf{z}) - v_*\big)\, \bar{\rho}_{\theta^*}(\mathbf{z})\, \mathrm{d}\mu(\mathbf{z}). \tag{39}$$

**Step 2: Use expressivity and convert to leakage.** By the expressivity assumption in Eq. (34),

$$\mathcal{F}_\tau(\rho_{\theta^*}) \;\leq\; \mathcal{F}_\tau^* + \varepsilon_{\mathrm{real}}(\tau).$$

Combining with Eq. (39) gives

$$\int_{\mathcal{Z}} \left( V_{\theta^*|\phi} - v_* \right) \bar{\rho}_{\theta^*} \, \mathrm{d}\mu \; \leq \; \frac{2\,\varepsilon_{\mathrm{real}}(\tau)}{M_{\mathrm{ex}}^{(\theta^*)}}. \tag{40}$$

Finally, since $V_{\theta^*|\phi}(\mathbf{z}) - v_* \geq \sigma$ $\mu$-a.e. on $\mathcal{Z} \setminus \mathcal{W}_{\theta^*}^{\sigma}$, we have

$$\int_{\mathcal{Z}} \left( V_{\theta^*|\phi} - v_* \right) \bar{\rho}_{\theta^*} \, \mathrm{d}\mu \geq \int_{\mathcal{Z}\setminus\mathcal{W}_{\theta^*}^{\sigma}} \left( V_{\theta^*|\phi} - v_* \right) \bar{\rho}_{\theta^*} \, \mathrm{d}\mu \geq \sigma \int_{\mathcal{Z}\setminus\mathcal{W}_{\theta^*}^{\sigma}} \bar{\rho}_{\theta^*} \, \mathrm{d}\mu = \sigma\,\bar{\rho}_{\theta^*}\mu(\mathcal{Z} \setminus \mathcal{W}_{\theta^*}^{\sigma}),$$

and therefore, by Eq. (40),

$$\bar{\rho}_{\theta^*}\mu(\mathcal{Z} \setminus \mathcal{W}_{\theta^*}^{\sigma}) \; \leq \; \frac{2\,\varepsilon_{\mathrm{real}}(\tau)}{\sigma M_{\mathrm{ex}}^{(\theta^*)}},$$

which is exactly Eq. (35). The concluding asymptotic statement follows immediately: if $\sigma \to 0$ with $\varepsilon_{\mathrm{real}}(\tau)/\sigma \to 0$, then the right-hand side vanishes and $\bar{\rho}_{\theta^*}\mu(\mathcal{Z} \setminus \mathcal{W}_{\theta^*}^{\sigma}) \longrightarrow 0$ as $\tau \to 0^+$. $\qquad\square$

## C. Proofs of Unimodal Formal Statements

This section provides rigorous proofs for all formal statements concerning our unimodal analysis. Each statement from the main text is restated for self-containment, accompanied by a proof intuition for providing high-level insights.

### C.1. Proof of Thm. 3.1

**Theorem 3.1** (Large-batch unimodal dynamics). *Consider the unimodal InfoNCE objective $\mathcal{L}_{\mathrm{NCE}}(\theta)$ in Eq. (1) and the parametric energy $\mathcal{J}_{\tau}(\theta)$ in Def. 3.2. Assume Asms. 2.1 and 3.1. Then for any fixed $\tau > 0$ and $\theta \in \Theta$, as $N \to \infty$,*

$$\left| \mathcal{L}_{\mathrm{NCE}}(\theta) - \mathcal{J}_{\tau}(\theta) - \log\left( N V_{\kappa}(\tau) \right) \right| \to 0, \tag{5}$$

$$\text{and} \;\; \left\| \nabla_{\theta}\mathcal{L}_{\mathrm{NCE}}(\theta) - \nabla_{\theta}\mathcal{J}_{\tau}(\theta) \right\| \to 0. \tag{6}$$

---

☞ **Proof intuition.** This proof establishes that as the number of negative samples grows infinitely large ($N \to \infty$), the empirical InfoNCE loss and its gradients converge to the deterministic, population-level energy functional $\mathcal{J}_{\tau}(\theta)$. The intuition rests on how the Law of Large Numbers transforms the partition function (i.e., the denominator) of the contrastive loss: **(i)** In a finite batch, the positive and negative samples compete inside the denominator. However, as $N \to \infty$, the single positive sample's contribution becomes mathematically negligible. Meanwhile, the discrete sum of the $N$ independent negative samples approximates the continuous, smoothed density of the entire representation manifold. Consequently, the loss cleanly splits into two decoupled forces: the expected alignment potential pulling positive pairs together, and the population-level cross-entropy pushing representations away from the global background density. **(ii)** To guarantee the optimization trajectories match, we analyze the gradients. The derivative of the log-denominator acts as a softmax-weighted average. As the crowd of negatives becomes infinite, the softmax weight assigned to the single positive pair naturally shrinks to zero. Thus, the repulsive "push" from the denominator is dictated entirely by the continuous distribution of negative samples. Because the underlying gradients are strictly bounded, this empirical crowd tracks the theoretical cross-entropy gradient, making large-batch InfoNCE mathematically equivalent to optimizing the continuous energy landscape.

---

*Proof.* Consider a batch $\mathcal{B}$ consisting of a positive pair and $N$ independent negative samples. For a specific anchor $\mathbf{x} \sim p_{\mathbf{x}}$, the batch is constructed as:

$$\mathcal{B} := \{(\mathbf{x}, \tilde{\mathbf{x}}), \mathbf{x}_1', \ldots, \mathbf{x}_N'\},$$

where the positive partner $\tilde{\mathbf{x}}$ is sampled from the positive conditional $r_{\mathrm{um}}(\tilde{\mathbf{x}}|\mathbf{x})$, and the negative samples $\{\mathbf{x}_j'\}_{j=1}^N$ are drawn i.i.d. from the marginal observation distribution $p_{\mathbf{x}}$. Fix any encoder parameter $\theta \in \Theta$, we denote the representations as $\mathbf{z} = f_{\theta}(\mathbf{x})$, $\mathbf{v} = f_{\theta}(\tilde{\mathbf{x}})$, and $\mathbf{w}_j = f_{\theta}(\mathbf{x}_j')$. The per-batch unimodal InfoNCE loss $\ell_{\mathcal{B}}(\theta; \mathbf{x})$ is defined as:

$$\ell_{\mathcal{B}}(\theta; \mathbf{x}) := -\tfrac{1}{\tau}s(\mathbf{z}, \mathbf{v}) + \log \underbrace{\left( \kappa_{\tau}(\mathbf{z}, \mathbf{v}) + \sum_{j=1}^{N} \kappa_{\tau}(\mathbf{z}, \mathbf{w}_j) \right)}_{=:Z_{\mathcal{B}}(\mathbf{z})}, \tag{41}$$

where the kernel $\kappa_{\tau}(\cdot, \cdot) = \exp(s(\cdot, \cdot)/\tau)$ induced by the loss formulation.

**Part I. Value consistency.** We analyze the asymptotic behavior of the loss as $N \to \infty$.

**Step 1: Isolate the positive bias.** First, we decompose the partition sum $Z_{\mathcal{B}}$ inside the logarithm into the positive term and the sum over negatives:

$$Z_{\mathcal{B}} = P + S_N, \quad \text{where } P := \kappa_\tau(\mathbf{z}, \mathbf{v}), \quad S_N := \sum_{j=1}^{N} \kappa_\tau(\mathbf{z}, \mathbf{w}_j).$$

The logarithmic term $\log Z_{\mathcal{B}}$ becomes:

$$\log Z_{\mathcal{B}} = \log(S_N + P) = \log S_N + \underbrace{\log\left(1 + \frac{P}{S_N}\right)}_{=: \epsilon_N}.$$

Fix $\tau > 0$. Since $\mathcal{Z}$ is compact and $s$ is continuous on $\mathcal{Z} \times \mathcal{Z}$ by Asm. 3.1 (ii), there exist finite constants $s_{\min}$ and $s_{\max}$ such that $s_{\min} \le s(\mathbf{z}, \mathbf{w}) \le s_{\max}$ for all $(\mathbf{z}, \mathbf{w}) \in \mathcal{Z} \times \mathcal{Z}$. Consequently, the exponential kernel satisfies the bounds

$$0 < m_\tau := \exp(s_{\min}/\tau) \le \kappa_\tau(\mathbf{z}, \mathbf{w}) \le \exp(s_{\max}/\tau) =: M_\tau < \infty.$$

Thus, $m_\tau \le P \le M_\tau$ and $N m_\tau \le S_N \le N M_\tau$. Using $\log(1 + x) \le x$ for $x \ge 0$,

$$0 < \epsilon_N = \log\left(1 + \frac{P}{S_N}\right) \le \frac{P}{S_N} \le \frac{M_\tau}{N m_\tau} = O_\tau(N^{-1}).$$

Therefore, for each fixed $\tau > 0$, $\log Z_{\mathcal{B}} = \log S_N + O_\tau(N^{-1})$.

**Step 2: Concentration of the negative partition sum.** Fix $\theta$ and an anchor representation $\mathbf{z} = f_\theta(\mathbf{x})$. For negatives $\mathbf{x}'_j \overset{\text{i.i.d.}}{\sim} p_{\mathbf{x}}$, define $\mathbf{w}_j := f_\theta(\mathbf{x}'_j)$ and

$$Y_j(\mathbf{z}) := \kappa_\tau(\mathbf{z}, \mathbf{w}_j), \qquad S_N(\mathbf{z}) := \sum_{j=1}^{N} Y_j(\mathbf{z}).$$

For fixed $\mathbf{z}$, the variables $\{Y_j(\mathbf{z})\}_{j=1}^{N}$ are i.i.d. with mean

$$\mu_\kappa(\mathbf{z}) := \mathbb{E}[Y_1(\mathbf{z}) \mid \mathbf{z}] = \int_{\mathcal{Z}} \kappa_\tau(\mathbf{z}, \mathbf{w}) \, dq_\theta(\mathbf{w}) \equiv \Gamma_{\theta, \tau}(\mathbf{z}) \ge m_\tau,$$

and variance $\sigma_\kappa^2(\mathbf{z}) := \text{Var}(Y_1(\mathbf{z}) \mid \mathbf{z}) < \infty$. Moreover, for each fixed $\tau > 0$, $\kappa_\tau$ is bounded on $\mathcal{Z} \times \mathcal{Z}$, so $Y_j(\mathbf{z})$ is bounded and hence sub-Gaussian. By Hoeffding's inequality, the sample mean concentrates:

$$\frac{S_N(\mathbf{z})}{N} - \mu_\kappa(\mathbf{z}) = O_p(N^{-1/2}).$$

Equivalently, writing

$$\frac{S_N(\mathbf{z})}{N \mu_\kappa(\mathbf{z})} = 1 + \Delta_N(\mathbf{z}), \qquad \Delta_N(\mathbf{z}) = O_p(N^{-1/2}),$$

and noting $\mu_\kappa(\mathbf{z}) > 0$ since $\kappa_\tau > 0$, we can expand the logarithm with the Taylor expansion $\log(1 + x) = x + O(x^2)$ and $\Delta_N(\mathbf{z}) = o_p(1)$:

$$\log S_N(\mathbf{z}) = \log\big(N \mu_\kappa(\mathbf{z})\big) + \log\big(1 + \Delta_N(\mathbf{z})\big) = \log N + \log \Gamma_{\theta, \tau}(\mathbf{z}) + O_p(N^{-1/2}),$$

Therefore,

$$\log S_N(\mathbf{z}) = \log N + \log \Gamma_{\theta, \tau}(\mathbf{z}) + O_p(N^{-1/2}).$$

**Step 3: Identification of the induced parametric energy.** Combining Steps 1–2, for an anchor $\mathbf{z} = f_\theta(\mathbf{x})$ we have

$$\log Z_{\mathcal{B}}(\mathbf{z}) = \log N + \log \Gamma_{\theta, \tau}(\mathbf{z}) + o_p(1),$$

where $o_p(1)$ denotes a term vanishing in probability as $N \to \infty$ (for fixed $\tau > 0$). Substituting into Eq. (41) yields the per-batch expansion

$$\ell_{\mathcal{B}}(\theta; \mathbf{x}) = -\tfrac{1}{\tau} s(\mathbf{z}, \mathbf{v}) + \log N + \log \Gamma_{\theta,\tau}(\mathbf{z}) + o_p(1). \tag{42}$$

Taking expectations over the batch construction $(\mathbf{x}, \tilde{\mathbf{x}}) \sim r_{\text{um}}$ and i.i.d. negatives, and using $\mathcal{L}_{\text{NCE}}(\theta) = \mathbb{E}[\ell_{\mathcal{B}}(\theta; \mathbf{x})]$, we obtain

$$\mathcal{L}_{\text{NCE}}(\theta) = \mathbb{E}_{(\mathbf{x}, \tilde{\mathbf{x}})}\left[ -\tfrac{1}{\tau} s(f_\theta(\mathbf{x}), f_\theta(\tilde{\mathbf{x}})) \right] + \mathbb{E}_{\mathbf{x}}[\log \Gamma_{\theta,\tau}(f_\theta(\mathbf{x}))] + \log N + o(1), \tag{43}$$

where $o(1) \to 0$ as $N \to \infty$.

By disintegration of $\pi_{\theta\theta} = (f_\theta \times f_\theta)_{\#} r_{\text{um}}$, define $U_\theta(\mathbf{z}) = -\int_{\mathcal{Z}} s(\mathbf{z}, \mathbf{w}) \, d\nu_{\mathbf{z}}(\mathbf{w})$. Then the law of iterated expectations gives

$$\mathbb{E}_{(\mathbf{x}, \tilde{\mathbf{x}})}\left[ -\tfrac{1}{\tau} s(\mathbf{z}, \mathbf{v}) \right] = \tfrac{1}{\tau} \mathbb{E}_{\mathbf{z} \sim q_\theta}[U_\theta(\mathbf{z})] = \tfrac{1}{\tau} \int_{\mathcal{Z}} U_\theta(\mathbf{z}) dq_\theta(\mathbf{z}) \tag{44}$$

Since $\mathbf{z} = f_\theta(\mathbf{x}) \sim q_\theta$, we can rewrite

$$\mathbb{E}_{\mathbf{x}}[\log \Gamma_{\theta,\tau}(f_\theta(\mathbf{x}))] = \mathbb{E}_{\mathbf{z} \sim q_\theta}[\log \Gamma_{\theta,\tau}(\mathbf{z})].$$

Under the constant kernel-volume condition Asm. 2.1, $\tilde{\rho}_{\theta,\tau}(\mathbf{z}) = \Gamma_{\theta,\tau}(\mathbf{z}) / V_\kappa(\tau)$ (Def. 2.2), hence

$$\mathbb{E}_{\mathbf{z} \sim q_\theta}[\log \Gamma_{\theta,\tau}(\mathbf{z})] = \log V_\kappa(\tau) + \mathbb{E}_{\mathbf{z} \sim q_\theta}[\log \tilde{\rho}_{\theta,\tau}(\mathbf{z})] = \log V_\kappa(\tau) - H_\times(q_\theta, \tilde{\rho}_{\theta,\tau}), \tag{45}$$

where $H_\times(q_\theta, \tilde{\rho}_{\theta,\tau}) := -\mathbb{E}_{q_\theta}[\log \tilde{\rho}_{\theta,\tau}]$ is the cross-entropy of $q_\theta$ against $\tilde{\rho}_{\theta,\tau}$.

Substituting Eqs. (44) and (45) into Eq. (43) yields

$$\begin{aligned} \mathcal{L}_{\text{NCE}}(\theta) &= \tfrac{1}{\tau} \int_{\mathcal{Z}} U_\theta(\mathbf{z}) dq_\theta(\mathbf{z}) - H_\times(q_\theta, \tilde{\rho}_{\theta,\tau}) + \log\left(N V_\kappa(\tau)\right) + o(1) \\ &= \mathcal{J}_\tau(\theta) + \log\left(N V_\kappa(\tau)\right) + o(1). \end{aligned}$$

Since $\theta$ is arbitrarily chosen from $\Theta$ and $\Theta$ is compact by Asm. 3.1, this proves the value-consistency $|\mathcal{L}_{\text{NCE}}(\theta) - \mathcal{J}_\tau(\theta) - \log(N V_\kappa(\tau))| \to 0$ of Thm. 3.1 for any fixed $\theta \in \Theta$.

**Part II. Gradient consistency.** We now analyze the gradient dynamics of the batch loss $\ell_{\mathcal{B}}(\theta; \mathbf{x})$. Fix $\tau > 0$ in the limit of $N \to \infty$. Differentiating Eq. (42) with respect to $\theta$ yields

$$\nabla_\theta \ell_{\mathcal{B}}(\theta; \mathbf{x}) = -\tfrac{1}{\tau} \nabla_\theta s(\mathbf{z}, \mathbf{v}) + \nabla_\theta \log \mathsf{Z}_{\mathcal{B}}(\mathbf{z}), \qquad \mathsf{Z}_{\mathcal{B}}(\mathbf{z}) = \mathsf{P} + \mathsf{S}_N, \tag{46}$$

where $\mathsf{P} := \kappa_\tau(\mathbf{z}, \mathbf{v})$ and $\mathsf{S}_N := \sum_{j=1}^N \kappa_\tau(\mathbf{z}, \mathbf{w}_j)$.

**Step 1: Reducing the positive term in the log-partition gradient.** Let

$$\mathsf{g}_N(\mathbf{z}) := \nabla_\theta \log(\mathsf{P} + \mathsf{S}_N), \qquad \mathsf{r}_N(\mathbf{z}) := \nabla_\theta \log \mathsf{S}_N.$$

Then

$$\begin{aligned} \mathsf{g}_N - \mathsf{r}_N &= \nabla_\theta \log(\mathsf{P} + \mathsf{S}_N) - \nabla_\theta \log \mathsf{S}_N \\ &= \frac{\nabla_\theta \mathsf{P} + \nabla_\theta \mathsf{S}_N}{\mathsf{P} + \mathsf{S}_N} - \frac{\nabla_\theta \mathsf{S}_N}{\mathsf{S}_N} \\ &= \frac{\nabla_\theta \mathsf{P}}{\mathsf{P} + \mathsf{S}_N} + \nabla_\theta \mathsf{S}_N \left( \frac{\mathsf{S}_N - (\mathsf{P} + \mathsf{S}_N)}{\mathsf{S}_N(\mathsf{P} + \mathsf{S}_N)} \right) \\ &= \frac{\nabla_\theta \mathsf{P}}{\mathsf{P} + \mathsf{S}_N} - \frac{\mathsf{P}}{\mathsf{S}_N(\mathsf{P} + \mathsf{S}_N)} \nabla_\theta \mathsf{S}_N. \end{aligned} \tag{47}$$

Under Asm. 3.1, the continuity of $s$ on the compact domain $\mathcal{Z} \times \mathcal{Z}$ ensures the kernel $\kappa_\tau$ is uniformly bounded. Furthermore, expanding $\nabla_\theta \kappa_\tau$ via the chain rule yields a product of terms—specifically the kernel, the similarity critic gradient, and the encoder Jacobian—that are all uniformly bounded by assumption. Thus, for any fixed $\tau > 0$, there exist finite constants

$M_\tau, G_\tau$ such that $0 < \kappa_\tau \leq M_\tau$ and $\|\nabla_\theta \kappa_\tau\| \leq G_\tau$. Moreover, since $\kappa_\tau$ is continuous and strictly positive on the compact $\mathcal{Z} \times \mathcal{Z}$, it attains a positive minimum $m_\tau > 0$. Hence

$$\mathsf{P} \leq M_\tau, \quad \|\nabla_\theta \mathsf{P}\| \leq G_\tau, \quad \mathsf{S}_N \geq N m_\tau, \quad \text{and} \quad \|\nabla_\theta \mathsf{S}_N\| \leq N G_\tau.$$

Taking norms in Eq. (47) and using these bounds gives

$$\|\mathsf{g}_N - \mathsf{r}_N\| \leq \frac{G_\tau}{N m_\tau} + \frac{M_\tau}{(N m_\tau)^2} \cdot N G_\tau = O(N^{-1}).$$

Therefore,

$$\nabla_\theta \log \mathsf{Z}_\mathcal{B}(\mathbf{z}) = \nabla_\theta \log \mathsf{S}_N(\mathbf{z}) + O(N^{-1}). \tag{48}$$

**Step 2: Consistency of the negative-sum ratio.** Write

$$\nabla_\theta \log \mathsf{S}_N(\mathbf{z}) = \frac{\nabla_\theta \mathsf{S}_N(\mathbf{z})}{\mathsf{S}_N(\mathbf{z})} = \frac{\frac{1}{N} \sum_{j=1}^N \nabla_\theta \kappa_\tau(\mathbf{z}, \mathbf{w}_j)}{\frac{1}{N} \sum_{j=1}^N \kappa_\tau(\mathbf{z}, \mathbf{w}_j)}.$$

Conditioning on the anchor $\mathbf{z}$, the negatives $\mathbf{w}_j$ are i.i.d. with law $q_\theta$, and both $\kappa_\tau(\mathbf{z}, \mathbf{w})$ and $\nabla_\theta \kappa_\tau(\mathbf{z}, \mathbf{w})$ are bounded (by Lem. B.1). Thus by the law of large numbers,

$$\frac{1}{N} \sum_{j=1}^N \kappa_\tau(\mathbf{z}, \mathbf{w}_j) \xrightarrow{p} \Gamma_{\theta,\tau}(\mathbf{z}), \qquad \frac{1}{N} \sum_{j=1}^N \nabla_\theta \kappa_\tau(\mathbf{z}, \mathbf{w}_j) \xrightarrow{p} \mathbb{E}_{\mathbf{w} \sim q_\theta}\big[\nabla_\theta \kappa_\tau(\mathbf{z}, \mathbf{w})\big].$$

As $q_\theta = (f_\theta)_\# p_\mathbf{x}$ and $\sup_{\theta \in \Theta, \mathbf{x} \in \mathcal{X}} \|\nabla_\theta \kappa_\tau(\mathbf{z}, f_\theta(\mathbf{x}))\| < \infty$ by Asm. 3.1, differentiation under the integral sign is justified (e.g., by dominated convergence). Thus, for each fixed $\mathbf{z}$,

$$\mathbb{E}_{\mathbf{w} \sim q_\theta}\big[\nabla_\theta \kappa_\tau(\mathbf{z}, \mathbf{w})\big] = \mathbb{E}_{\mathbf{x} \sim p_\mathbf{x}}\big[\nabla_\theta \kappa_\tau(\mathbf{z}, f_\theta(\mathbf{x}))\big] = \nabla_\theta \mathbb{E}_{\mathbf{x} \sim p_\mathbf{x}}\big[\kappa_\tau(\mathbf{z}, f_\theta(\mathbf{x}))\big] = \nabla_\theta \Gamma_{\theta,\tau}(\mathbf{z}).$$

Since $\Gamma_{\theta,\tau}(\mathbf{z}) \geq m_\tau > 0$, the continuous mapping theorem yields

$$\nabla_\theta \log \mathsf{S}_N(\mathbf{z}) \xrightarrow{p} \frac{\nabla_\theta \Gamma_{\theta,\tau}(\mathbf{z})}{\Gamma_{\theta,\tau}(\mathbf{z})} = \nabla_\theta \log \Gamma_{\theta,\tau}(\mathbf{z}). \tag{49}$$

Combining Eqs. (48) and (49) gives

$$\nabla_\theta \log \mathsf{Z}_\mathcal{B}(\mathbf{z}) \xrightarrow{p} \nabla_\theta \log \Gamma_{\theta,\tau}(\mathbf{z}). \tag{50}$$

**Step 3: Passing to the population objective and identifying $\nabla_\theta \mathcal{J}_\tau$.** Recall $\mathcal{L}_{\mathrm{NCE}}(\theta) = \mathbb{E}_\mathcal{B}[\ell_\mathcal{B}(\theta; \mathbf{x})]$ and $\nabla_\theta \mathcal{L}_{\mathrm{NCE}}(\theta) = \mathbb{E}_\mathcal{B}[\nabla_\theta \ell_\mathcal{B}(\theta; \mathbf{x})]$ by Lem. B.1. From Eq. (46) and Eq. (50), we have the pointwise (in the batch randomness) convergence

$$\nabla_\theta \ell_\mathcal{B}(\theta; \mathbf{x}) = -\tfrac{1}{\tau} \nabla_\theta s(\mathbf{z}, \mathbf{v}) + \nabla_\theta \log \mathsf{Z}_\mathcal{B}(\mathbf{z}) \xrightarrow{p} -\tfrac{1}{\tau} \nabla_\theta s(\mathbf{z}, \mathbf{v}) + \nabla_\theta \log \Gamma_{\theta,\tau}(\mathbf{z}).$$

Moreover, by Lem. B.1, $\|\nabla_\theta \ell_\mathcal{B}\|$ is uniformly bounded, so we may use dominated convergence together with Eq. (50) to interchange $\lim_{N \to \infty}$ and expectation:

$$\nabla_\theta \mathcal{L}_{\mathrm{NCE}}(\theta) = \mathbb{E}_\mathcal{B}\big[\nabla_\theta \ell_\mathcal{B}(\theta; \mathbf{x})\big] \to -\tfrac{1}{\tau} \mathbb{E}_{r_{\mathrm{um}}}\big[\nabla_\theta s(f_\theta(\mathbf{x}), f_\theta(\tilde{\mathbf{x}}))\big] + \mathbb{E}_{\mathbf{x} \sim p_\mathbf{x}}\big[\nabla_\theta \log \Gamma_{\theta,\tau}(f_\theta(\mathbf{x}))\big]. \tag{$\star$}$$

We now identify the two terms in $(\star)$ as the gradient of the parametric energy $\mathcal{J}_\tau(\theta)$. Recall that the alignment potential field as a function of $\theta$ by

$$\int_\mathcal{Z} U_\theta(\mathbf{z}) \mathrm{d}q_\theta(\mathbf{z}) := \mathbb{E}_{(\mathbf{x}, \tilde{\mathbf{x}}) \sim r_{\mathrm{um}}}\big[-s(f_\theta(\mathbf{x}), f_\theta(\tilde{\mathbf{x}}))\big].$$

Under Asm. 3.1, $\nabla_\theta s$ is bounded on the relevant compact set, so we may differentiate under the expectation to obtain

$$\nabla_\theta \int_\mathcal{Z} U_\theta(\mathbf{z}) \mathrm{d}q_\theta(\mathbf{z}) = -\mathbb{E}_{r_{\mathrm{um}}}\big[\nabla_\theta s(f_\theta(\mathbf{x}), f_\theta(\tilde{\mathbf{x}}))\big].$$

Therefore the first term in $(\star)$ equals $\frac{1}{\tau} \nabla_\theta \int_\mathcal{Z} U_\theta(\mathbf{z}) \mathrm{d}q_\theta(\mathbf{z})$.

By definition,
$$H_\times(q_\theta, \tilde{\rho}_{\theta,\tau}) := -\mathbb{E}_{\mathbf{z} \sim q_\theta}\big[\log \tilde{\rho}_{\theta,\tau}(\mathbf{z})\big] = -\mathbb{E}_{\mathbf{x} \sim p_\mathbf{x}}\big[\log \tilde{\rho}_{\theta,\tau}(f_\theta(\mathbf{x}))\big].$$

Under Asm. 2.1, $\tilde{\rho}_{\theta,\tau}(\mathbf{z}) = \Gamma_{\theta,\tau}(\mathbf{z})/V_\kappa(\tau)$ with $V_\kappa(\tau)$ constant in $\theta$ (Def. 2.2). Hence

$$\nabla_\theta H_\times(q_\theta, \tilde{\rho}_{\theta,\tau}) = -\mathbb{E}_\mathbf{x}\big[\nabla_\theta \log \tilde{\rho}_{\theta,\tau}(f_\theta(\mathbf{x}))\big] = -\mathbb{E}_\mathbf{x}\big[\nabla_\theta \log \Gamma_{\theta,\tau}(f_\theta(\mathbf{x}))\big],$$

where differentiation under the expectation is justified by boundedness of $\nabla_\theta \log \Gamma_{\theta,\tau}(f_\theta(\mathbf{x}))$. Thus the second term in $(\star)$ equals $-\nabla_\theta H_\times(q_\theta, \tilde{\rho}_{\theta,\tau})$. This gives

$$\nabla_\theta \mathcal{L}_{\mathrm{NCE}}(\theta) \longrightarrow \tfrac{1}{\tau}\nabla_\theta \int_\mathcal{Z} U_\theta(\mathbf{z})\mathrm{d}q_\theta(\mathbf{z}) - \nabla_\theta H_\times(q_\theta, \tilde{\rho}_{\theta,\tau}) = \nabla_\theta \mathcal{J}_\tau(\theta),$$

which proves gradient consistency with the cross-entropy-induced energy $\mathcal{J}_\tau$ for each fixed $\tau > 0$. Since $\theta$ is arbitrarily chosen from $\Theta$, this proves the gradient-consistency $\|\nabla_\theta \mathcal{L}_{\mathrm{NCE}}(\theta) - \nabla_\theta \mathcal{J}_\tau(\theta)\| \to 0$ of Thm. 3.1. $\qquad\square$

## C.2. Proof of Prop. 3.1

**Proposition 3.1** (Unique Gibbs equilibrium). *Assume $\mathcal{Z}$ is compact with $\mu(\mathcal{Z}) < \infty$. Fix $\tau > 0$ and let $U : \mathcal{Z} \to \mathbb{R}$ be Borel measurable and bounded below $\mu$-a.e. Then, $\mathcal{F}_{\tau,U}$ is strictly convex on $\mathcal{P}_\mu(\mathcal{Z})$ with a unique minimizer*

$$\rho^*(\mathbf{z}) = \frac{\exp(-U(\mathbf{z})/\tau)}{Z_\tau}, \; Z_\tau := \int_\mathcal{Z} \exp\left(\frac{-U(\mathbf{z})}{\tau}\right) \mathrm{d}\mu(\mathbf{z}).$$

> ☞ **Proof intuition.** The objective of this proof is to demonstrate that the representation energy functional always settles into a single, predictable equilibrium state—the Gibbs distribution. The logic unfolds in three geometric phases: First, we confirm that the space of all valid probability distributions is convex. If you mathematically blend any two valid distributions, the result remains a valid distribution. This establishes a well-behaved, continuous foundation for optimization. Next, we examine the shape of the energy landscape sitting on top of this foundation. The functional consists of two competing forces: a linear binding energy term (which tilts the landscape) and a negative entropy term. Because negative entropy is strictly convex, it forces the entire landscape to curve upward. Combining these creates a strictly convex bowl. A perfect bowl over a convex domain is mathematically guaranteed to have exactly one absolute lowest point. Finally, as we are guaranteed a unique global minimum, finding it is just a matter of locating where the functional's derivative is zero. By applying the calculus of variations—constrained by the requirement that the total probability must sum to 1—we derive the Gibbs distribution ($\rho^*$). This proves that the system's absolute ground state balances the thermodynamic pull toward low-energy regions with the entropic push to remain spread out.

*Proof.* We analyze the properties of the functional $\mathcal{F}_{\tau,U} : \mathcal{P}_\mu(\mathcal{Z}) \to \mathbb{R}$ defined on the space of probability densities $\mathcal{P}_\mu(\mathcal{Z}) = \{\rho \in L^1(\mathcal{Z}) : \rho \geq 0 \; \mu\text{-a.e.}, \int_\mathcal{Z} \rho \, \mathrm{d}\mu = 1\}$. The proof proceeds in three steps: establishing the geometry of the domain, proving the convexity of the energy functional, and deriving the unique minimizer.

**Step 1: Convexity of the domain $\mathcal{P}(\mathcal{Z})$.** First, we verify that the optimization domain $\mathcal{P}_\mu(\mathcal{Z})$ is a convex set. Consider any two densities $\rho_1, \rho_2 \in \mathcal{P}_\mu(\mathcal{Z})$ and a mixing coefficient $\lambda \in (0,1)$. Define the interpolation $\rho_\lambda := \lambda \rho_1 + (1 - \lambda)\rho_2$. We observe that $\rho_\lambda$ satisfies the following two conditions. (i) Non-negativity: Since $\rho_1, \rho_2 \geq 0$ and $\lambda \in (0,1)$ by construction, $\rho_\lambda \geq 0$ $\mu$-a.e. (ii) Normalization: By the linearity of the integral, $\int_\mathcal{Z} \rho_\lambda = \lambda \int_\mathcal{Z} \rho_1 \mathrm{d}\mu + (1 - \lambda) \int_\mathcal{Z} \rho_2 \mathrm{d}\mu = 1$. Therefore, $\rho_\lambda \in \mathcal{P}_\mu(\mathcal{Z})$ for any $\lambda \in (0,1)$, confirming that the space of valid densities $\mathcal{P}_\mu(\mathcal{Z})$ is a convex set.

**Step 2: Strict functional convexity.** Let $\rho_1, \rho_2 \in \mathcal{P}_\mu(\mathcal{Z})$ be two distinct densities (i.e., $\rho_1 \not\equiv \rho_2$ almost everywhere) and let $\lambda \in (0,1)$ be a mixing coefficient. Define the convex combination $\rho_\lambda := \lambda \rho_1 + (1 - \lambda)\rho_2$. We now analyze the two components of $\mathcal{F}_{\tau,U}(\rho_\lambda)$ separately.

In the binding energy term, since $U(\mathbf{z})$ acts as a fixed coefficient relative to $\rho$ for fixed $\tau > 0$, the total binding energy $\frac{1}{\tau}\int_\mathcal{Z} U(\mathbf{z})\rho(\mathbf{z}) \, \mathrm{d}\mu(\mathbf{z})$ is therefore a linear functional. By the linearity of the Lebesgue integral:

$$\begin{aligned}
\tfrac{1}{\tau} \int_\mathcal{Z} U(\mathbf{z})\rho(\mathbf{z}) \, \mathrm{d}\mu(\mathbf{z}) &= \tfrac{1}{\tau} \int_\mathcal{Z} (\lambda \rho_1(\mathbf{z}) + (1 - \lambda)\rho_2(\mathbf{z}))U(\mathbf{z}) \, \mathrm{d}\mu(\mathbf{z}) \\
&= \tfrac{\lambda}{\tau} \int_\mathcal{Z} \rho_1(\mathbf{z})U(\mathbf{z}) \, \mathrm{d}\mu(\mathbf{z}) + \tfrac{1-\lambda}{\tau} \int_\mathcal{Z} \rho_2(\mathbf{z})U(\mathbf{z}) \, \mathrm{d}\mu(\mathbf{z}).
\end{aligned}$$

The entropy term is $-H(\rho) = \int_{\mathcal{Z}} \rho \log \rho \, d\mu$, with the convention $0 \log 0 = 0$. Since $\phi(t) = t \log t$ is strictly convex on $(0, \infty)$ and convex on $[0, \infty)$, we have

$$\int_{\mathcal{Z}} \phi(\rho_\lambda) \, d\mu < \lambda \int_{\mathcal{Z}} \phi(\rho_1) \, d\mu + (1 - \lambda) \int_{\mathcal{Z}} \phi(\rho_2) \, d\mu$$

whenever $\rho_1 \not\equiv \rho_2$ (i.e., they differ on a set of positive $\mu$-measure). Hence $-H$ is strictly convex on $\mathcal{P}_\mu(\mathcal{Z})$. Combining the two parts:

$$
\begin{aligned}
\mathcal{F}_{\tau,U}(\rho_\lambda) &= \tfrac{1}{\tau} \int_{\mathcal{Z}} U(\mathbf{z}) \rho(\mathbf{z}) \, d\mu(\mathbf{z}) - H(\rho_\lambda) \\
&< \tfrac{\lambda}{\tau} \int_{\mathcal{Z}} U(\mathbf{z}) \rho_1(\mathbf{z}) \, d\mu(\mathbf{z}) + \tfrac{1-\lambda}{\tau} \int_{\mathcal{Z}} U(\mathbf{z}) \rho_2(\mathbf{z}) \, d\mu(\mathbf{z}) - \lambda H(\rho_1) - (1 - \lambda) H(\rho_2) \\
&= \lambda \mathcal{F}_{\tau,U}(\rho_1) + (1 - \lambda) \mathcal{F}_{\tau,U}(\rho_2).
\end{aligned}
$$

Therefore, the functional $\mathcal{F}_{\tau,U}(\rho)$ is strictly convex over the space of probability densities $\mathcal{P}_\mu(\mathcal{Z})$.

**Step 3: Existence and uniqueness of global minimizer.** Since $\mathcal{F}_{\tau,U}(\rho)$ is strictly convex on the convex domain $\mathcal{P}_\mu(\mathcal{Z})$, any local minimizer is the unique global minimizer. For fixed $\tau > 0$, we solve for the optimal density $\rho^* \in \mathcal{P}_\mu(\mathcal{Z})$ using the method of Lagrange multipliers. Define the Lagrangian $\Lambda(\rho, \gamma)$ to enforce the normalization constraint $\int_{\mathcal{Z}} \rho(\mathbf{z}) d\mu(\mathbf{z}) = 1$:

$$\Lambda(\rho, \gamma) := \mathcal{F}_{\tau,U}(\rho) + \gamma \left( \int_{\mathcal{Z}} \rho(\mathbf{z}) d\mu(\mathbf{z}) - 1 \right).$$

Taking the functional derivative with respect to $\rho(\mathbf{z})$ and setting it to zero:

$$
\begin{aligned}
\frac{\delta \Lambda}{\delta \rho(\mathbf{z})} &= \int_{\mathcal{Z}} \frac{\delta}{\delta \rho(\mathbf{z})} \left[ \rho(\mathbf{w}) \left( \tfrac{1}{\tau} U(\mathbf{w}) + \log \rho(\mathbf{w}) + \gamma \right) \right] d\mu(\mathbf{w}) \\
&= = \tfrac{1}{\tau} U(\mathbf{z}) + \log \rho(\mathbf{z}) + 1 + \gamma. \\
&= 0.
\end{aligned}
$$

Solving for $\rho^*(\mathbf{z})$:

$$\log \rho^*(\mathbf{z}) = -\frac{U(\mathbf{z})}{\tau} - \gamma - 1 \quad \Longrightarrow \quad \rho^*(\mathbf{z}) = \frac{1}{\exp(\gamma + 1)} \exp\left( -\frac{U(\mathbf{z})}{\tau} \right),$$

where the first term of the right-hand side is a constant determined by the normalization constraint. Therefore, the unique minimizer is the Gibbs distribution:

$$\rho^*(\mathbf{z}) = \frac{1}{Z_\tau} \cdot \exp\left( -\frac{U(\mathbf{z})}{\tau} \right), \quad \text{where} \ \ Z_\tau = \int_{\mathcal{Z}} \exp\left( -\frac{U(\mathbf{z})}{\tau} \right) d\mu(\mathbf{w}).$$

Since $\operatorname{ess\,inf}_\mu U > -\infty$ by assumption, we have $U(\mathbf{z}) \geq m$ for $\mu$-a.e. $\mathbf{z}$. Hence

$$0 < Z_\tau = \int_{\mathcal{Z}} e^{-U(\mathbf{z})/\tau} \, d\mu(\mathbf{z}) \leq e^{-m/\tau} \mu(\mathcal{Z}) < \infty,$$

so $\rho^*$ is well-defined. This confirms that the system admits a unique Gibbs equilibrium characterized by the alignment potential field $U$. $\qquad\square$

### C.3. Proof of Prop. 3.2

**Proposition 3.2** (Low-temperature concentration). *In the setting of Prop. 3.1, fix $\sigma > 0$ and define*

$$\mathcal{W}^\sigma := \left\{ \mathbf{z} \in \mathcal{Z} : \ U(\mathbf{z}) \leq \operatorname{ess\,inf}_\mu U + \sigma \right\}.$$

*Then as $\tau \to 0^+$, the Gibbs measure $\rho^*(\mathbf{z}) \, d\mu(\mathbf{z})$ concentrates on $\mathcal{W}^\sigma$ in the sense that, for every open $\mathcal{O} \supset \mathcal{W}^\sigma$, $\rho^* \mu(\mathcal{Z} \setminus \mathcal{O}) \to 0$.*

☞ **Proof intuition.** This proof establishes that as the temperature $\tau \to 0^+$, the Gibbs distribution concentrates near the absolute lowest valleys of the potential landscape (the $\sigma$-ground state). The argument relies on a direct comparison of exponential decay rates relative to the minimum energy level $m$. We evaluate the ratio of probability mass between two distinct regions: **(i)** In any region strictly outside the ground state, the potential energy is guaranteed to be at least $\sigma$ higher than the absolute minimum. Consequently, the total probability mass in this outer region decays exponentially fast, strictly bounded above by a factor of $\exp(-\sigma/\tau)$. **(ii)** Conversely, because the true minimum $m$ is defined by the essential infimum, there must exist a pocket of volume near the bottom of the valley where the energy is strictly within $\sigma/2$ of the minimum. The probability mass in this core pocket shrinks much more slowly, strictly bounded below by $\exp(-\sigma/(2\tau))$. By taking the ratio of these two bounds, we mathematically squeeze the outer probability mass against the inner mass. Because $\exp(-\sigma/\tau)$ decays exponentially faster than $\exp(-\sigma/(2\tau))$, their ratio gets crushed to exactly zero as $\tau \to 0^+$. This guarantees that no statistical mass can survive outside the lowest-energy valleys, resulting in ground-state concentration.

*Proof.* We analyze the Gibbs-equilibrium measure $\rho^* \mu$ as $\tau \to 0^+$. Let

$$m := \operatorname{ess\,inf}_\mu U \qquad \text{and} \qquad \mathcal{W}^\sigma := \Big\{ \mathbf{z} \in \mathcal{Z} : U(\mathbf{z}) \le m + \sigma \Big\},$$

for an arbitrarily small but fixed $\sigma > 0$.

Consider any open set $\mathcal{O} \subset \mathcal{Z}$ such that $\mathcal{O} \supset \mathcal{W}^\sigma$, and denote $\mathcal{O}^c := \mathcal{Z} \setminus \mathcal{O}$. Since $\mathcal{O}$ contains $\mathcal{W}^\sigma$, its complement satisfies

$$\mathcal{O}^c \subset \{ \mathbf{z} \in \mathcal{Z} : U(\mathbf{z}) > m + \sigma \}.$$

Therefore, using the Gibbs distribution $\rho^*(\mathbf{z}) = \frac{1}{Z_\tau} \exp\big( - U(\mathbf{z})/\tau \big)$,

$$(\rho^* \mu)(\mathcal{O}^c) = \frac{\int_{\mathcal{O}^c} \exp\big( - U(\mathbf{z})/\tau \big)\, d\mu(\mathbf{z})}{\int_{\mathcal{Z}} \exp\big( - U(\mathbf{w})/\tau \big)\, d\mu(\mathbf{w})}.$$

Multiplying numerator and denominator by $\exp(m/\tau)$ yields

$$(\rho^* \mu)(\mathcal{O}^c) = \frac{\int_{\mathcal{O}^c} \exp\big( - (U(\mathbf{z}) - m)/\tau \big)\, d\mu(\mathbf{z})}{\int_{\mathcal{Z}} \exp\big( - (U(\mathbf{w}) - m)/\tau \big)\, d\mu(\mathbf{w})}.$$

(i) Numerator bound. For all $\mathbf{z} \in \mathcal{O}^c$ we have $U(\mathbf{z}) - m \ge \sigma$, hence

$$\int_{\mathcal{O}^c} \exp\Big( - \frac{U(\mathbf{z}) - m}{\tau} \Big)\, d\mu(\mathbf{z}) \le \int_{\mathcal{O}^c} \exp\Big( - \frac{\sigma}{\tau} \Big)\, d\mu(\mathbf{z}) = \mu(\mathcal{O}^c) \exp\Big( - \frac{\sigma}{\tau} \Big).$$

(ii) Denominator lower bound. By the definition of $m = \operatorname{ess\,inf}_\mu U$, for any $\eta > 0$ the set $\{ \mathbf{z} \in \mathcal{Z} : U(\mathbf{z}) < m + \eta \}$ has strictly positive $\mu$-measure. Fix $\eta := \sigma/2$ and define

$$\mathcal{A} := \Big\{ \mathbf{w} \in \mathcal{Z} \,\Big|\, U(\mathbf{w}) < m + \sigma/2 \Big\}, \qquad \text{so that} \qquad \mu(\mathcal{A}) > 0.$$

On $\mathcal{A}$ we have $U(\mathbf{w}) - m < \sigma/2$, hence

$$\int_{\mathcal{Z}} \exp\Big( - \frac{U(\mathbf{w}) - m}{\tau} \Big)\, d\mu(\mathbf{w}) \ge \int_{\mathcal{A}} \exp\Big( - \frac{U(\mathbf{w}) - m}{\tau} \Big)\, d\mu(\mathbf{w}) \ge \mu(\mathcal{A}) \exp\Big( - \frac{\sigma}{2\tau} \Big).$$

Combining the bounds, we obtain

$$(\rho^* \mu)(\mathcal{O}^c) \le \frac{\mu(\mathcal{O}^c) \exp(-\sigma/\tau)}{\mu(\mathcal{A}) \exp(-\sigma/(2\tau))} = \frac{\mu(\mathcal{O}^c)}{\mu(\mathcal{A})} \exp\Big( - \frac{\sigma}{2\tau} \Big).$$

Since $\mu(\mathcal{O}^c) \le \mu(\mathcal{Z}) < \infty$ and $\mu(\mathcal{A}) > 0$ is fixed, the right-hand side converges to 0 as $\tau \to 0^+$. Hence, for any open neighborhood $\mathcal{O} \supset \mathcal{W}^\sigma$, we have $(\rho^* \mu)(\mathcal{O}^c) \to 0$, i.e., the Gibbs measure concentrates in any neighborhood of $\mathcal{W}^\sigma$. □

## C.4. Proof of Thm. 3.2

**Theorem 3.2** (Low-temperature unimodal energy consistency). *Assume Asms. 2.1, 3.1 and 3.2. Fix $\theta \in \Theta$ and assume $q_\theta \ll \mu$ with a strictly positive, continuous density $\rho_\theta$, i.e., $\inf_{\mathbf{z} \in \mathcal{Z}} \rho_\theta(\mathbf{z}) \geq \underline{\rho}_\theta > 0$.[6] Define the KDE mismatch $\varepsilon_{\text{kde}}^{(\theta)}(\tau) := \|\tilde{\rho}_{\theta,\tau} - \rho_\theta\|_\infty$. Then there exists $\tau_0(\theta) > 0$ such that*

$$\left| \mathcal{J}_\tau(\theta) - \mathcal{F}_{\tau, U_\theta}(\rho_\theta) \right| \leq 2\,\varepsilon_{\text{kde}}^{(\theta)}(\tau)/\underline{\rho}_\theta, \quad \forall\, 0 < \tau \leq \tau_0(\theta),$$

*and in particular, as $\tau \to 0^+$, $\left| \mathcal{J}_\tau(\theta) - \mathcal{F}_{\tau, U_\theta}(\rho_\theta) \right| \to 0$.*

---

☞ **Proof intuition.** The objective of this proof is to establish that at low temperatures, the parametric energy functional $\mathcal{J}_\tau(\theta)$ mirrors the intrinsic thermodynamic free energy $\mathcal{F}_{\tau, U_\theta}(\rho_\theta)$. The argument hinges on translating their algebraic difference into an information-theoretic distance, and then squeezing that distance to zero. **(i)** We first algebraically decompose the parametric energy and reveal a fundamental identity: the exact gap between the learned objective and the theoretical free energy is simply the KL divergence between the true representation density ($\rho_\theta$) and its kernel-smoothed counterpart ($\tilde{\rho}_{\theta,\tau}$). **(ii)** To prove this KL divergence vanishes, we must ensure the logarithmic ratio inside the integral never explodes. By our assumption, the true density is strictly bounded away from zero. Because Lem. B.3 guarantees the smoothed density tracks the true density as $\tau \to 0^+$, we mathematically guarantee that the smoothed density also never collapses to zero anywhere on the manifold. **(iii)** Because both densities remain safely away from zero, the logarithm operates in a stable, Lipschitz-continuous regime. This allows us to upper-bound the complex integral of the KL divergence directly by the maximum worst-case smoothing error (the KDE mismatch). Since this KDE mismatch vanishes at $\tau \to 0^+$, the KL divergence is crushed to zero, guaranteeing the parametric and intrinsic energy landscapes become indistinguishable.

---

*Proof.* **Step 1: Well-posedness of the alignment potential term.** Fix any $\theta \in \Theta$. Recall the definition of the alignment potential field (Def. 3.1) by

$$U_\theta(\mathbf{z}) := - \int_{\mathcal{Z}} s(\mathbf{z}, \mathbf{w})\, \mathrm{d}\nu_{\theta, \mathbf{z}}(\mathbf{w}) \qquad \text{for } q_\theta\text{-a.e. } \mathbf{z}.$$

Under Asm. 3.1, the critic $s$ is bounded and Borel measurable, hence $U_\theta$ is Borel measurable and bounded, and therefore integrable with respect to $q_\theta$. Moreover, $q_\theta \ll \mu$ with a continuous density $\rho_\theta$ bounded away from zero by assumption. Thus, the intrinsic representation functional (Eq. (7))

$$\mathcal{F}_{\tau, U_\theta}(\rho_\theta) := \tfrac{1}{\tau} \int_{\mathcal{Z}} U_\theta(\mathbf{z}) \rho_\theta(\mathbf{z}) \mathrm{d}\mu(\mathbf{z}) - H(\rho_\theta)$$

is well-defined for each $\tau > 0$.

**Step 2: Difference reduces to a KL divergence.** Recall

$$\mathcal{J}_\tau(\theta) = \tfrac{1}{\tau} \int_{\mathcal{Z}} U_\theta(\mathbf{z}) \mathrm{d}q_\theta(\mathbf{z}) - H_\times(q_\theta, \tilde{\rho}_{\theta,\tau}).$$

Using $H_\times(q, \tilde{\rho}) = H(\rho) + D_{\text{KL}}(\rho \| \tilde{\rho})$ where the density is defined by $\rho := \mathrm{d}q/\mathrm{d}\mu$, we obtain

$$\mathcal{J}_\tau(\theta) = \tfrac{1}{\tau} \int_{\mathcal{Z}} U_\theta(\mathbf{z}) \rho_\theta \mathrm{d}\mu(\mathbf{z}) - H(\rho_\theta) - D_{\text{KL}}(\rho_\theta \| \tilde{\rho}_{\theta,\tau}),$$

and hence

$$\mathcal{F}_{\tau, U_\theta}(\rho_\theta) - \mathcal{J}_\tau(\theta) = D_{\text{KL}}(\rho_\theta \| \tilde{\rho}_{\theta,\tau}). \tag{51}$$

**Step 3: Vanishing KL divergence of the smoothing.** Fix any $\theta \in \Theta$. By our initial assumption, the true continuous density is uniformly bounded away from zero, meaning $\inf_{\mathbf{z} \in \mathcal{Z}} \rho_\theta(\mathbf{z}) \geq \underline{\rho}_\theta > 0$. Because this lower bound is strictly positive, the quantity $\underline{\rho}_\theta/2$ is also strictly positive and can serve as an arbitrary error tolerance.

---

[6]See App. A.3 for a detailed discussion on the strict positivity assumption and the role of density floors.

Recall from Lem. B.3 that the KDE mismatch vanishes in the low-temperature limit, i.e., $\lim_{\tau \to 0^+} \varepsilon_{\mathrm{kde}}^{(\theta)}(\tau) = 0$. By the formal definition of a right-sided limit, applying our chosen tolerance $\underline{\rho}_\theta/2$ guarantees the existence of a critical temperature threshold $\tau_0(\theta) > 0$ such that for all $\tau \le \tau_0(\theta)$, the smoothing error remains strictly bounded by this tolerance:

$$\varepsilon_{\mathrm{kde}}^{(\theta)}(\tau) := \|\tilde{\rho}_{\theta,\tau} - \rho_\theta\|_\infty \le \frac{\underline{\rho}_\theta}{2}.$$

Then for all $\mathbf{z}$,

$$\tilde{\rho}_{\theta,\tau}(\mathbf{z}) \ge \rho_\theta(\mathbf{z}) - \frac{\underline{\rho}_\theta}{2} \ge \frac{\underline{\rho}_\theta}{2}. \tag{52}$$

On $[\underline{\rho}_\theta/2, \infty)$, the derivative of $\log x$ is $1/x \le 2/\underline{\rho}_\theta$. Hence for all $a, b > \underline{\rho}_\theta/2$,

$$|\log a - \log b| \le \frac{2}{\underline{\rho}_\theta}|a - b|.$$

Apply this with $a = \rho_\theta(\mathbf{z})$ and $b = \tilde{\rho}_{\theta,\tau}(\mathbf{z})$. Therefore,

$$\left|\log \frac{\rho_\theta(\mathbf{z})}{\tilde{\rho}_{\theta,\tau}(\mathbf{z})}\right| = |\log \rho_\theta(\mathbf{z}) - \log \tilde{\rho}_{\theta,\tau}(\mathbf{z})| \le \frac{2}{\underline{\rho}_\theta}|\rho_\theta(\mathbf{z}) - \tilde{\rho}_{\theta,\tau}(\mathbf{z})| \le \frac{2\,\varepsilon_{\mathrm{kde}}^{(\theta)}(\tau)}{\underline{\rho}_\theta}. \tag{53}$$

Now

$$D_{\mathrm{KL}}(\rho_\theta \| \tilde{\rho}_{\theta,\tau}) = \int_\mathcal{Z} \rho_\theta(\mathbf{z}) \log \frac{\rho_\theta(\mathbf{z})}{\tilde{\rho}_{\theta,\tau}(\mathbf{z})} \mathrm{d}\mu(\mathbf{z}).$$

Using $\int_\mathcal{Z} \rho_\theta \mathrm{d}\mu = 1$ and Eq. (53),

$$D_{\mathrm{KL}}(\rho_\theta \| \tilde{\rho}_{\theta,\tau}) \le \int_\mathcal{Z} \rho_\theta(\mathbf{z}) \left|\log \frac{\rho_\theta(\mathbf{z})}{\tilde{\rho}_{\theta,\tau}(\mathbf{z})}\right| \mathrm{d}\mu(\mathbf{z}) \le \frac{2\,\varepsilon_{\mathrm{kde}}^{(\theta)}(\tau)}{\underline{\rho}_\theta}. \tag{54}$$

Substituting this to Eq. (51) yields

$$\left|\mathcal{J}_\tau(\theta) - \mathcal{F}_{\tau,U_\theta}(\rho_\theta)\right| \le \frac{2\,\varepsilon_{\mathrm{kde}}^{(\theta)}(\tau)}{\underline{\rho}_\theta}.$$

Finally, Lem. B.3 guarantees that the smoothing error $\varepsilon_{\mathrm{kde}}^{(\theta)}(\tau)$ vanishes in the low-temperature limit. Consequently, $\left|\mathcal{J}_\tau(\theta) - \mathcal{F}_{\tau,U_\theta}(\rho_\theta)\right| \xrightarrow{\tau \to 0^+} 0$, which concludes the proof. $\qquad\square$

# D. Proofs of Multimodal Formal Statements

This section provides rigorous proofs for all formal statements concerning our multimodal analysis. Each statement from the main text is restated for self-containment, accompanied by a proof intuition for providing high-level insights.

## D.1. Proof of Thm. 4.1

**Theorem 4.1** (Large-batch multimodal dynamics). *Consider the symmetric multimodal loss $\mathcal{L}_{\mathrm{Sym}}(\theta, \phi)$ in Eq. (2) and the energy $\mathcal{J}_\tau^{\mathrm{Sym}}(\theta, \phi)$ in Def. 4.2. Assume Asms. 2.1 and 4.1. Then, for any fixed $\tau > 0$ and $(\theta, \phi) \in \Theta \times \Phi$, as $N \to \infty$,*

$$\left|\mathcal{L}_{\mathrm{Sym}}(\theta, \phi) - \mathcal{J}_\tau^{\mathrm{Sym}}(\theta, \phi) - \log\left(N V_\kappa(\tau)\right)\right| \to 0,$$
$$\text{and } \left\|\nabla_{(\theta,\phi)}\mathcal{L}_{\mathrm{Sym}}(\theta, \phi) - \nabla_{(\theta,\phi)}\mathcal{J}_\tau^{\mathrm{Sym}}(\theta, \phi)\right\| \to 0.$$

☞ **Proof intuition.** This proof extends the asymptotic analysis of unimodal InfoNCE to the dual-encoder, cross-modal setting. The logic directly inherits the statistical mechanics of the unimodal proof (Thm. 3.1) but applies them independently across the two modalities: **(i)** The symmetric loss is the average of two directional losses (e.g., vision-to-language and language-to-vision). By the Law of Large Numbers, the massive empirical sum of negative keys in each direction independently concentrates around the smoothed continuous density of that specific target modality (e.g., the vision anchors push against the continuous background density of the language representations, and vice versa). This

cleanly splits the symmetric objective into a shared attractive alignment potential and two distinct, cross-modal repulsive cross-entropies. **(ii)** When calculating the joint gradient, the derivative of the log-denominators acts as a softmax-weighted average. As $N \to \infty$, the positive pairs' influence on the repulsive forces vanishes. Because the parameter spaces $\Theta$ and $\Phi$ are orthogonal, the empirical gradient of the massive negative crowd in each direction independently and tracks its respective continuous cross-entropy gradient. Fusing these orthogonal bounds via the Pythagorean theorem guarantees that descending the empirical cross-modal loss is mathematically equivalent to minimizing the continuous, dual-density energy landscape.

*Proof.* Consider a batch $\mathcal{B}$ consisting of a positive pair $(\mathbf{x}, \mathbf{y}) \sim r_{\mathrm{mm}}$ together with $N$ i.i.d. negatives $\{\mathbf{y}'_j\}_{j=1}^N \overset{\text{i.i.d.}}{\sim} p_{\mathbf{y}}$ and $N$ i.i.d. negatives $\{\mathbf{x}'_j\}_{j=1}^N \overset{\text{i.i.d.}}{\sim} p_{\mathbf{x}}$, all independent conditional on $(\mathbf{x}, \mathbf{y})$. Write the representations

$$\mathbf{z} = f_\theta(\mathbf{x}), \qquad \mathbf{w} = g_\phi(\mathbf{y}), \qquad \mathbf{w}'_j = g_\phi(\mathbf{y}'_j), \qquad \mathbf{z}'_j = f_\theta(\mathbf{x}'_j),$$

and define the exponential kernel $\kappa_\tau(\cdot, \cdot) = \exp(s(\cdot, \cdot)/\tau)$. Denote the directional partition sums

$$\mathsf{Z}_{\mathcal{B}}^{\theta \to \phi}(\mathbf{z}) := \kappa_\tau(\mathbf{z}, \mathbf{w}) + \sum_{j=1}^N \kappa_\tau(\mathbf{z}, \mathbf{w}'_j), \qquad \mathsf{Z}_{\mathcal{B}}^{\phi \to \theta}(\mathbf{w}) := \kappa_\tau(\mathbf{z}, \mathbf{w}) + \sum_{j=1}^N \kappa_\tau(\mathbf{z}'_j, \mathbf{w}).$$

The symmetric per-batch loss can be written as

$$\ell_{\mathcal{B}}^{\mathrm{Sym}}(\theta, \phi) = \tfrac{1}{2} \underbrace{\left(-\tfrac{1}{\tau} s(\mathbf{z}, \mathbf{w}) + \log \mathsf{Z}_{\mathcal{B}}^{\theta \to \phi}(\mathbf{z})\right)}_{\ell_{\mathcal{B}}^{x \to y}(\theta, \phi)} + \tfrac{1}{2} \underbrace{\left(-\tfrac{1}{\tau} s(\mathbf{z}, \mathbf{w}) + \log \mathsf{Z}_{\mathcal{B}}^{\phi \to \theta}(\mathbf{w})\right)}_{\ell_{\mathcal{B}}^{y \to x}(\theta, \phi)}. \tag{55}$$

**Part I. Value consistency.** Fix any $\tau > 0$ and $(\theta, \phi) \in \Theta \times \Phi$. We analyze the directional loss $\mathcal{L}_{\mathrm{NCE}}^{x \to y}(\theta, \phi)$; the reverse direction follows by symmetry. From Eq. (55), the corresponding per-batch directional loss is

$$\ell_{\mathcal{B}}^{x \to y}(\theta, \phi) := -\tfrac{1}{\tau} s(\mathbf{z}, \mathbf{w}) + \log \mathsf{Z}_{\mathcal{B}}^{\theta \to \phi}(\mathbf{z}). \tag{56}$$

**Step 1: Bias–variance control of the log-partition term.** The structure of $\mathsf{Z}_{\mathcal{B}}^{\theta \to \phi}$ is identical to the unimodal partition function (Eq. (41)), except that negatives are drawn from $q_\phi$ rather than $q_\theta$. By assumption, $\mathcal{Z}$ is compact and $s$ is continuous, hence $\kappa_\tau$ is bounded on $\mathcal{Z} \times \mathcal{Z}$ for each fixed $\tau > 0$. Let

$$\mathsf{P} := \kappa_\tau(\mathbf{z}, \mathbf{w}), \qquad \mathsf{S}_N^{\theta \to \phi}(\mathbf{z}) := \sum_{j=1}^N \kappa_\tau(\mathbf{z}, \mathbf{w}'_j), \qquad \mathsf{Z}_{\mathcal{B}}^{\theta \to \phi} = \mathsf{P} + \mathsf{S}_N^{\theta \to \phi}.$$

As in Step 1 of the unimodal proof (Part I of Thm. 3.1), the positive contribution induces a vanishing bias:

$$\log(\mathsf{P} + \mathsf{S}_N^{\theta \to \phi}) = \log \mathsf{S}_N^{\theta \to \phi} + \log\left(1 + \frac{\mathsf{P}}{\mathsf{S}_N^{\theta \to \phi}}\right), \qquad \log\left(1 + \frac{\mathsf{P}}{\mathsf{S}_N^{\theta \to \phi}}\right) = O_\tau(N^{-1}).$$

Define the population partition function induced by the target encoder $\phi$:

$$\Gamma_{\phi, \tau}(\mathbf{z}) := \mathbb{E}_{\mathbf{w} \sim q_\phi}\left[\kappa_\tau(\mathbf{z}, \mathbf{w})\right] = \int_{\mathcal{Z}} \kappa_\tau(\mathbf{z}, \mathbf{w}) \, \mathrm{d}q_\phi(\mathbf{w}).$$

Conditioned on the anchor $\mathbf{z}$, the terms $\kappa_\tau(\mathbf{z}, \mathbf{w}'_j)$ are i.i.d. and bounded, hence by concentration (cf. Step 2 in Part I of Thm. 3.1)

$$\tfrac{1}{N} \mathsf{S}_N^{\theta \to \phi}(\mathbf{z}) = \Gamma_{\phi, \tau}(\mathbf{z}) + O_p(N^{-1/2}),$$

which implies, by the same Taylor expansion used in the unimodal proof, that

$$\log \mathsf{S}_N^{\theta \to \phi}(\mathbf{z}) = \log N + \log \Gamma_{\phi, \tau}(\mathbf{z}) + O_p(N^{-1/2}).$$

Combining the above displays yields the asymptotic expansion

$$\ell_{\mathcal{B}}^{x \to y}(\theta, \phi) = -\tfrac{1}{\tau} s(\mathbf{z}, \mathbf{w}) + \log N + \log \Gamma_{\phi, \tau}(\mathbf{z}) + O_p(N^{-1/2}). \tag{57}$$

**Step 2: Identification of the directional energy.** Taking expectation of Eq. (57) over the batch construction gives

$$\mathcal{L}_{\mathrm{NCE}}^{x \to y}(\theta, \phi) := \mathbb{E}_{\mathcal{B}}\big[\ell_{\mathcal{B}}^{\theta \to \phi}(\theta, \phi)\big] = \mathbb{E}\big[-\tfrac{1}{\tau} s(\mathbf{z}, \mathbf{w})\big] + \mathbb{E}[\log \Gamma_{\phi, \tau}(\mathbf{z})] + \log N + o(1).$$

Here, since $\kappa_\tau$ is uniformly bounded for fixed $\tau$, the remainder term is uniformly integrable, so taking expectations preserves $o_p(1)$ as $o(1)$. Moreover, $o(1) \to 0$ as $N \to \infty$ for fixed $\tau > 0$ by boundedness and dominated convergence.

Now, treating $\mathbf{x}$ (equivalently $\mathbf{z}$) as the anchor and using the disintegration in Def. 4.1, the law of iterated expectation yields

$$\mathbb{E}_{(\mathbf{x}, \mathbf{y}) \sim r_{\mathrm{mm}}}\big[-\tfrac{1}{\tau} s(f_\theta(\mathbf{x}), g_\phi(\mathbf{y}))\big] = \tfrac{1}{\tau} \mathbb{E}_{\mathbf{z} \sim q_\theta}\big[U_{\theta \to \phi}(\mathbf{z})\big] = \tfrac{1}{\tau} \int_{\mathcal{Z}} U_{\theta \to \phi}(\mathbf{z}) \mathrm{d} q_\theta(\mathbf{z}).$$

Since $\mathbf{z} = f_\theta(\mathbf{x}) \sim q_\theta$, we have

$$\mathbb{E}[\log \Gamma_{\phi, \tau}(\mathbf{z})] = \mathbb{E}_{\mathbf{z} \sim q_\theta}[\log \Gamma_{\phi, \tau}(\mathbf{z})].$$

Under the constant kernel-volume condition Asm. 2.1, $\tilde{\rho}_{\phi, \tau}(\mathbf{z}) = \Gamma_{\phi, \tau}(\mathbf{z}) / V_\kappa(\tau)$ (Def. 2.2), hence

$$\mathbb{E}_{\mathbf{z} \sim q_\theta}[\log \Gamma_{\phi, \tau}(\mathbf{z})] = \log V_\kappa(\tau) + \mathbb{E}_{\mathbf{z} \sim q_\theta}[\log \tilde{\rho}_{\phi, \tau}(\mathbf{z})] = \log V_\kappa(\tau) - H_\times(q_\theta, \tilde{\rho}_{\phi, \tau}).$$

Substituting into the preceding expansion gives

$$\begin{aligned} \mathcal{L}_{\mathrm{NCE}}^{x \to y}(\theta, \phi) &= \tfrac{1}{\tau} \int_{\mathcal{Z}} U_{\theta \to \phi}(\mathbf{z}) \mathrm{d} q_\theta(\mathbf{z}) - H_\times(q_\theta, \tilde{\rho}_{\phi, \tau}) + \log\big(N V_\kappa(\tau)\big) + o(1) \\ &= \mathcal{J}_\tau^{x \to y}(\theta, \phi) + \log\big(N V_\kappa(\tau)\big) + o(1). \end{aligned}$$

The reverse direction $y \to x$ follows analogously, yielding the same form with $(\theta, \phi)$ swapped. Averaging the two directional expansions and using the definition Eq. (9) proves the value-consistency: as $N \to \infty$,

$$\big| \mathcal{L}_{\mathrm{Sym}}(\theta, \phi) - \mathcal{J}_\tau^{\mathrm{Sym}}(\theta, \phi) - \log\big(N V_\kappa(\tau)\big) \big| \to 0, \qquad \forall \tau > 0, (\theta, \phi) \in \Theta \times \Phi.$$

**Part II: Gradient consistency.** Fix any $\tau > 0$ and $(\theta, \phi) \in \Theta \times \Phi$. Differentiating the batch loss in Eq. (55) gives

$$\nabla \ell_{\mathcal{B}}^{\mathrm{Sym}}(\theta, \phi) = -\tfrac{1}{\tau} \nabla s(\mathbf{z}, \mathbf{w}) + \tfrac{1}{2} \nabla \log \mathsf{Z}_{\mathcal{B}}^{\theta \to \phi}(\mathbf{z}) + \tfrac{1}{2} \nabla \log \mathsf{Z}_{\mathcal{B}}^{\phi \to \theta}(\mathbf{w}), \tag{58}$$

where we denote the joint gradient $\nabla_{(\theta, \phi)}$ as $\nabla$ for simplicity.

**Step 1: Removing the positive term in each log-partition gradient.** Define

$$\mathsf{g}_N^{\theta \to \phi} := \nabla \log(\mathsf{P} + \mathsf{S}_N^{\theta \to \phi}), \qquad \mathsf{r}_N^{\theta \to \phi} := \nabla \log \mathsf{S}_N^{\theta \to \phi},$$

and analogously $(\mathsf{g}_N^{\phi \to \theta}, \mathsf{r}_N^{\phi \to \theta})$. As in Eq. (47), a direct algebraic manipulation yields

$$\mathsf{g}_N^{\theta \to \phi} - \mathsf{r}_N^{\theta \to \phi} = \frac{\nabla \mathsf{P}}{\mathsf{P} + \mathsf{S}_N^{\theta \to \phi}} - \frac{\mathsf{P}}{\mathsf{S}_N^{\theta \to \phi}(\mathsf{P} + \mathsf{S}_N^{\theta \to \phi})} \nabla \mathsf{S}_N^{\theta \to \phi},$$

and the same identity holds with $\theta \to \phi$ replaced by $\phi \to \theta$.

By assumption, $\mathcal{Z}$ is compact and $s$ is $C^1$ with bounded input gradient. Together with the $C^1$ encoders with uniformly bounded Jacobians, this implies that for each fixed $\tau > 0$ there exist finite constants

$$0 < m_\tau \le \kappa_\tau(\cdot, \cdot) \le M_\tau < \infty, \qquad \sup_{(\theta, \phi) \in \Theta \times \Phi} \sup_{\mathcal{B}} \|\nabla \kappa_\tau\| \le G_\tau < \infty.$$

Consequently, uniformly over $(\theta, \phi)$ and batch realizations,

$$\mathsf{P} \le M_\tau, \quad \|\nabla \mathsf{P}\| \le G_\tau, \quad \mathsf{S}_N^{\theta \to \phi} \ge N m_\tau, \quad \|\nabla \mathsf{S}_N^{\theta \to \phi}\| \le N G_\tau,$$

and the same bounds hold for $\mathsf{S}_N^{\phi\rightarrow\theta}$. Taking norms in the identity above gives the uniform bound

$$\|\nabla\log\mathsf{Z}_{\mathcal{B}}^{\theta\rightarrow\phi}(\mathbf{z}) - \nabla\log\mathsf{S}_N^{\theta\rightarrow\phi}(\mathbf{z})\| = O(N^{-1}), \quad \|\nabla\log\mathsf{Z}_{\mathcal{B}}^{\phi\rightarrow\theta}(\mathbf{w}) - \nabla\log\mathsf{S}_N^{\phi\rightarrow\theta}(\mathbf{w})\| = O(N^{-1}), \tag{59}$$

where the $O(N^{-1})$ constants depend on $\tau$ but are uniform over $\Theta\times\Phi$.

**Step 2: Consistency of the negative-sum ratios.** We treat the direction $\theta\rightarrow\phi$; the reverse direction is analogous. Rewrite

$$\nabla\log\mathsf{S}_N^{\theta\rightarrow\phi} = \frac{\nabla\mathsf{S}_N^{\theta\rightarrow\phi}}{\mathsf{S}_N^{\theta\rightarrow\phi}} = \frac{\frac{1}{N}\sum_{j=1}^{N}\nabla\kappa_\tau(\mathbf{z},\mathbf{w}_j')}{\frac{1}{N}\sum_{j=1}^{N}\kappa_\tau(\mathbf{z},\mathbf{w}_j')}.$$

Conditioning on $(\mathbf{x},\mathbf{y})$, the anchor representations $\mathbf{z} = f_\theta(\mathbf{x})$ and $\mathbf{w} = g_\phi(\mathbf{y})$ are deterministic for fixed $(\theta,\phi)$. The negatives $\{\mathbf{y}_j'\}_{j=1}^{N}$ are i.i.d. from $p_\mathbf{y}$, hence $\{\mathbf{w}_j'\}_{j=1}^{N}$ are i.i.d. through the map $g_\phi$. Write the kernel-smoothed mass as

$$\Gamma_{\phi,\tau}(\mathbf{z}) = \mathbb{E}_{\mathbf{y}'\sim p_\mathbf{y}}\big[\kappa_\tau(\mathbf{z},g_\phi(\mathbf{y}'))\big],$$

which is strictly positive and uniformly bounded: $\Gamma_{\phi,\tau}(\mathbf{z}) \geq m_\tau$. By the law of large numbers and boundedness of $\kappa_\tau$,

$$\frac{1}{N}\sum_{j=1}^{N}\kappa_\tau(\mathbf{z},\mathbf{w}_j') \xrightarrow{p} \Gamma_{\phi,\tau}(\mathbf{z}).$$

Similarly, since $\|\nabla\kappa_\tau\|$ is uniformly bounded,

$$\frac{1}{N}\sum_{j=1}^{N}\nabla\kappa_\tau(\mathbf{z},\mathbf{w}_j') \xrightarrow{p} \mathbb{E}_{\mathbf{y}'\sim p_\mathbf{y}}\big[\nabla\kappa_\tau(\mathbf{z},g_\phi(\mathbf{y}'))\big].$$

Since the expectation is taken with respect to the parameter-free distribution $p_\mathbf{y}$ and, by Asm. 4.1, $\sup_{(\theta,\phi)}\sup_{\mathbf{y}'\in\mathcal{Y}}\|\nabla\kappa_\tau(\mathbf{z},g_\phi(\mathbf{y}'))\| < \infty$, differentiation under the expectation is justified by dominated convergence. Hence

$$\mathbb{E}_{\mathbf{y}'\sim p_\mathbf{y}}\big[\nabla\kappa_\tau(\mathbf{z},g_\phi(\mathbf{y}'))\big] = \nabla\mathbb{E}_{\mathbf{y}'\sim p_\mathbf{y}}\big[\kappa_\tau(\mathbf{z},g_\phi(\mathbf{y}'))\big] = \nabla\Gamma_{\phi,\tau}(\mathbf{z}).$$

Since the denominator converges to a strictly positive limit, the continuous mapping theorem yields

$$\nabla\log\mathsf{S}_N^{\theta\rightarrow\phi} \xrightarrow{p} \frac{\nabla\Gamma_{\phi,\tau}(\mathbf{z})}{\Gamma_{\phi,\tau}(\mathbf{z})} = \nabla\log\Gamma_{\phi,\tau}(\mathbf{z}). \tag{60}$$

The reverse direction follows identically by defining

$$\Gamma_{\theta,\tau}(\mathbf{w}) := \mathbb{E}_{\mathbf{x}'\sim p_\mathbf{x}}\big[\kappa_\tau(f_\theta(\mathbf{x}'),\mathbf{w})\big],$$

and obtaining

$$\nabla\log\mathsf{S}_N^{\phi\rightarrow\theta} \xrightarrow{p} \nabla\log\Gamma_{\theta,\tau}(\mathbf{w}). \tag{61}$$

Combining Eqs. (59) to (61), as $N\rightarrow\infty$, we obtain the batch-level limit

$$\nabla\log\mathsf{Z}_{\mathcal{B}}^{\theta\rightarrow\phi}(\mathbf{z}) \xrightarrow{p} \nabla\log\Gamma_{\phi,\tau}(\mathbf{z}), \qquad \nabla\log\mathsf{Z}_{\mathcal{B}}^{\phi\rightarrow\theta}(\mathbf{w}) \xrightarrow{p} \nabla\log\Gamma_{\theta,\tau}(\mathbf{w}). \tag{62}$$

**Step 3: Passing to the population objective and identifying $\nabla_{(\theta,\phi)}\mathcal{J}_\tau^{\mathrm{Sym}}$.** By Lem. B.2, $\|\nabla\ell_{\mathcal{B}}^{\mathrm{Sym}}(\theta,\phi)\|$ is uniformly bounded over $\Theta\times\Phi$ and all $N$. Therefore, applying dominated convergence to Eq. (58) together with Eq. (62) yields

$$\begin{aligned}
\nabla\mathcal{L}_{\mathrm{Sym}}(\theta,\phi) = \mathbb{E}_{\mathcal{B}}\big[\nabla\ell_{\mathcal{B}}^{\mathrm{Sym}}(\theta,\phi)\big] \longrightarrow &-\tfrac{1}{\tau}\mathbb{E}_{(\mathbf{z},\mathbf{w})\sim\pi_{\theta\phi}}\big[\nabla s(\mathbf{z},\mathbf{w})\big] \\
&+\tfrac{1}{2}\mathbb{E}_{\mathbf{z}\sim q_\theta}\big[\nabla\log\Gamma_{\phi,\tau}(\mathbf{z})\big] \\
&+\tfrac{1}{2}\mathbb{E}_{\mathbf{w}\sim q_\phi}\big[\nabla\log\Gamma_{\theta,\tau}(\mathbf{w})\big].
\end{aligned} \tag{63}$$

We now identify the right-hand side as $\nabla \mathcal{J}_\tau^{\text{Sym}}(\theta, \phi)$. First, by the definition of $\pi_{\theta\phi} = (f_\theta \times g_\phi)_{\#} r_{\text{mm}}$ and the law of iterated expectation applied to Def. 4.1,

$$\int_{\mathcal{Z}} U_{\theta \to \phi}(\mathbf{z}) \, dq_\theta(\mathbf{z}) = \mathbb{E}_{(\mathbf{x},\mathbf{y}) \sim r_{\text{mm}}} \big[ - s(f_\theta(\mathbf{x}), g_\phi(\mathbf{y})) \big] = \mathbb{E}_{(\mathbf{z},\mathbf{w}) \sim \pi_{\theta\phi}} \big[ \nabla s(\mathbf{z}, \mathbf{w}) \big],$$

and similarly $\int_{\mathcal{Z}} U_{\phi \to \theta}(\mathbf{w}) \, dq_\phi(\mathbf{w})$ equals the same joint expectation.[7] Under Asm. 4.1, we may differentiate under the expectation to obtain

$$\nabla \int_{\mathcal{Z}} U_{\theta \to \phi}(\mathbf{z}) \, dq_\theta(\mathbf{z}) = \nabla \int_{\mathcal{Z}} U_{\phi \to \theta}(\mathbf{w}) \, dq_\phi(\mathbf{w}) = -\mathbb{E}_{(\mathbf{z},\mathbf{w}) \sim \pi_{\theta\phi}} \big[ \nabla s(\mathbf{z}, \mathbf{w}) \big].$$

Second, using $\log \tilde{\rho}_{\phi,\tau} = \log \Gamma_{\phi,\tau} - \log V_\kappa(\tau)$ and $\log \tilde{\rho}_{\theta,\tau} = \log \Gamma_{\theta,\tau} - \log V_\kappa(\tau)$ (Def. 2.2), with $V_\kappa(\tau)$ independent of $(\theta, \phi)$ by Asm. 2.1, we have

$$\nabla H_\times(q_\theta, \tilde{\rho}_{\phi,\tau}) = -\nabla \int_{\mathcal{Z}} \log \tilde{\rho}_{\phi,\tau}(\mathbf{z}) \, dq_\theta(\mathbf{z}) = -\int_{\mathcal{Z}} \nabla \log \Gamma_{\phi,\tau}(\mathbf{z}) \, dq_\theta(\mathbf{z}),$$

$$\nabla H_\times(q_\phi, \tilde{\rho}_{\theta,\tau}) = -\nabla \int_{\mathcal{Z}} \log \tilde{\rho}_{\theta,\tau}(\mathbf{w}) \, dq_\phi(\mathbf{w}) = -\int_{\mathcal{Z}} \nabla \log \Gamma_{\theta,\tau}(\mathbf{w}) \, dq_\phi(\mathbf{w}),$$

where the derivative passes under the integral by dominated convergence, since $\|\nabla \log \Gamma_{\cdot,\tau}\|$ is uniformly bounded by Asm. 4.1 and Lem. B.2.

Combining these identities with the definition of $\mathcal{J}_\tau^{\text{Sym}}$ in Def. 4.2 shows that the limit in Eq. (63) equals $\nabla \mathcal{J}_\tau^{\text{Sym}}(\theta, \phi)$. Therefore, as $N \to \infty$, for any fixed $(\theta, \phi) \in \Theta \times \Phi$, the $\theta$- and $\phi$-components of the above argument yield

$$\|\nabla_\theta \mathcal{L}_{\text{Sym}} - \nabla_\theta \mathcal{J}_\tau^{\text{Sym}}\| \to 0, \qquad \|\nabla_\phi \mathcal{L}_{\text{Sym}} - \nabla_\phi \mathcal{J}_\tau^{\text{Sym}}\| \to 0.$$

Consequently, for the joint gradient $\nabla_{(\theta,\phi)} = (\nabla_\theta, \nabla_\phi)$,

$$\left\| \nabla_{(\theta,\phi)} \mathcal{L}_{\text{Sym}} - \nabla_{(\theta,\phi)} \mathcal{J}_\tau^{\text{Sym}} \right\| = \sqrt{ \left\| \nabla_\theta \mathcal{L}_{\text{Sym}} - \nabla_\theta \mathcal{J}_\tau^{\text{Sym}} \right\|^2 + \left\| \nabla_\phi \mathcal{L}_{\text{Sym}} - \nabla_\phi \mathcal{J}_\tau^{\text{Sym}} \right\|^2 } \longrightarrow 0.$$

$\square$

### D.2. Explicit Extremal Construction and Proof of Prop. 4.1

We now give the extremal-density construction underlying Prop. 4.1 and full proof of the proposition.

**Proposition D.1** (Constructive form of barrier best response and extremal convergence). *Assume $\mathcal{Z}$ is compact with $\mu(\mathcal{Z}) < \infty$, and let $\mathcal{F}_{\tau, \mathbf{U}_{1,2}}^{\text{Sym}}$ be as in Def. 4.3. Fix $\rho_2 \in \mathcal{P}_{\mu, \underline{\rho}_2}(\mathcal{Z})$ and define the* effective potential field *by $V_{1|2}(\mathbf{z}) := \frac{1}{\tau} U_{1 \to 2}(\mathbf{z}) + \log \rho_2(\mathbf{z})$. Then, the coordinate map $\rho_1 \mapsto \mathcal{F}_{\tau, \mathbf{U}_{1,2}}^{\text{Sym}}(\rho_1, \rho_2)$ is concave on $\mathcal{P}_{\mu, \underline{\rho}_1}(\mathcal{Z})$, and its infimum is approached by sequences that concentrate the* excess mass $M_{\text{ex}}^{(1)} := 1 - \underline{\rho}_1 \mu(\mathcal{Z})$ *onto approximate minimizers of $V_{1|2}$, leaving only the floor $\underline{\rho}_1$ elsewhere. Formally, for a sufficiently small $\sigma > 0$, let $\mathcal{W}_1^\sigma := \{\mathbf{z} \in \mathcal{Z} : V_{1|2}(\mathbf{z}) \leq \operatorname{ess\,inf}_\mu V_{1|2} + \sigma\}$ be the sublevel set. Define the $\sigma$-approximate density $\rho_1^{(\sigma)}$ by concentrating the excess mass on a shrinking set $\mathcal{S}_1^\sigma \subseteq \mathcal{W}_1^\sigma$:[8]*

$$\rho_1^{(\sigma)}(\mathbf{z}) = \underline{\rho} + \mathbb{1}_{\mathcal{S}_1^\sigma}(\mathbf{z}) \cdot M_{\text{ex}}^{(1)} / \mu(\mathcal{S}_1^\sigma), \qquad \text{with} \quad \lim_{\sigma \to 0} \mu(\mathcal{S}_1^\sigma) = 0. \tag{64}$$

*As $\sigma \to 0$, the energy of $\rho_1^{(\sigma)}$ converges to the infimum:*

$$\lim_{\sigma \to 0} \mathcal{F}_{\tau, \mathbf{U}_{1,2}}^{\text{Sym}}(\rho_1^{(\sigma)}, \rho_2) = \inf_{\rho_1 \in \mathcal{P}_{\mu, \underline{\rho}_1}(\mathcal{Z})} \mathcal{F}_{\tau, \mathbf{U}_{1,2}}^{\text{Sym}}(\rho_1, \rho_2). \tag{65}$$

*An analogous statement holds when optimizing in $\rho_2$ with $\rho_1$ fixed, with respect to the dual effective potential field $V_{2|1}(\mathbf{z}) := \frac{1}{\tau} U_{2 \to 1}(\mathbf{z}) + \log \rho_1(\mathbf{z})$.*

---

[7]While the binding energy yields the same expectation, the potential fields $U_{\theta \to \phi}$ and $U_{\phi \to \theta}$ differ pointwise.

[8]Typically, $\mathcal{S}_1^\sigma = \mathcal{W}_1^\sigma$ for sharp minima, or any vanishing subset $\mathcal{S}_1^\sigma \subset \mathcal{W}_1^\sigma$ for flat minima.

☞ **Proof intuition.** This proof establishes that when one modality's distribution ($\rho_2$) is held fixed, the optimal distribution for the other modality ($\rho_1$) concentrates near an extreme, "spiky" state. Rather than spreading out, it flattens to the absolute minimum allowed density everywhere, while dumping all its remaining probability mass into the deepest valley of an effective potential landscape. The logic unfolds in three geometric phases: **(i)** Unlike standard convex optimization where distributions seek a balanced equilibrium, the cross-entropy term here makes the functional *concave* with respect to $\rho_1$. Concave minimization pushes solutions to the extreme edges of the feasible space. We calculate a theoretical absolute minimum energy (i.e., the floor) by imagining a hypothetical distribution that sits flat at the strict lower bound $\underline{\rho}_1$ everywhere, with all its "excess" probability mass concentrated at the absolute minimum of the effective potential $V_{1|2}$. **(ii)** To prove this theoretical floor is physically reachable, we construct a sequence of valid distributions. We take the excess mass and squeeze it into a microscopic, highly concentrated spike (a sublevel set) directly over the lowest valleys of the potential. **(iii)** Squeezing mass into an infinitely narrow, tall spike threatens to blow up the logarithmic entropy penalty. However, by exploiting the absolute continuity of the background measure, we can precisely control how fast the spike shrinks. We prove that the log-penalty grows slower than the base shrinks, ultimately crushing the penalty to zero. The sequence touches the theoretical floor, confirming that the optimal coordinate state is a singular concentration atop a flat baseline.

*Proof.* Fix $\tau > 0$ and $\rho_2 \in \mathcal{P}_{\mu,\underline{\rho}_1}(\mathcal{Z})$. Throughout, we treat $\rho_2$ as fixed and analyze $\rho_1$ (while analysis for $\rho_2$ applies symmetrically) and write the coordinate map as

$$\mathcal{F}_1(\rho_1) := \mathcal{F}_{\tau,\mathbf{U}_{1,2}}^{\mathrm{Sym}}(\rho_1, \rho_2), \qquad \rho_1 \in \mathcal{P}_{\mu,\underline{\rho}_1}(\mathcal{Z}).$$

**Step 1: Concavity of $\rho_1 \mapsto \mathcal{F}_\tau$ and its lower bound.** Using $H(\rho_1) = -\int_{\mathcal{Z}} \rho_1 \log \rho_1 \mathrm{d}\mu$, $D_{\mathrm{KL}}(\rho_1 \| \rho_2) = \int_{\mathcal{Z}} \rho_1 \log(\rho_1/\rho_2) \mathrm{d}\mu$, and $D_{\mathrm{KL}}(\rho_2 \| \rho_1) = \int_{\mathcal{Z}} \rho_2 \log(\rho_2/\rho_1) \mathrm{d}\mu$, expand the $\rho_1$-dependent terms in $\mathcal{F}_1(\rho_1)$:

$$\mathcal{F}_1(\rho_1) = \tfrac{1}{2} \int_{\mathcal{Z}} \rho_1(\mathbf{z}) V_{1|2}(\mathbf{z}) \mathrm{d}\mu(\mathbf{z}) + \tfrac{1}{2} \int_{\mathcal{Z}} \rho_2(\mathbf{z}) \log \rho_1(\mathbf{z}) \mathrm{d}\mu(\mathbf{z}) + C(\rho_2), \tag{66}$$

where $V_{1|2}(\mathbf{z}) := \tfrac{1}{\tau} U_{1,2}(\mathbf{z}) + \log \rho_2(\mathbf{z})$, and $C(\rho_2)$ does not depend on $\rho_1$.

We observe $\rho_1 \mapsto \int_{\mathcal{Z}} \rho_1 V_{1|2} \mathrm{d}\mu$ is linear, and $\rho_1 \mapsto \int_{\mathcal{Z}} \rho_2 \log \rho_1 \mathrm{d}\mu$ is concave on the positive cone since $\log$ is concave and $\rho_2 \geq \underline{\rho}_2 > 0$. Thus, $\rho_1 \mapsto \mathcal{F}_1$ is concave on the convex set $\mathcal{P}_{\mu,\underline{\rho}_1}(\mathcal{Z})$. Consequently, the infimum over $\mathcal{P}_{\mu,\underline{\rho}_1}(\mathcal{Z})$ can be approached by a sequence on the boundary.

Now, write any feasible $\rho_1 \in \mathcal{P}_{\mu,\underline{\rho}_1}(\mathcal{Z})$ as $\rho_1(\mathbf{z}) = \underline{\rho}_1 + \eta_1(\mathbf{z})$ so that $\eta_1(\mathbf{z}) \geq 0$. The excess probability mass is strictly positive

$$M_{\mathrm{ex}}^{(1)} = \int_{\mathcal{Z}} \eta_1(\mathbf{z}) \mathrm{d}\mu(\mathbf{z}) > 0.$$

Then, using $V_{1|2}(\mathbf{z}) \geq \operatorname{ess\,inf}_\mu V_{1|2} =: v_*$ and $\log \rho_1(\mathbf{z}) \geq \log \underline{\rho}_1$ almost everywhere,

$$\int_{\mathcal{Z}} \rho_1(\mathbf{z}) V_{1|2}(\mathbf{z}) \mathrm{d}\mu(\mathbf{z}) = \underline{\rho}_1 \int_{\mathcal{Z}} V_{1|2}(\mathbf{z}) \mathrm{d}\mu(\mathbf{z}) + \int_{\mathcal{Z}} \eta_1(\mathbf{z}) V_{1|2}(\mathbf{z}) \mathrm{d}\mu(\mathbf{z})$$

$$\geq \underline{\rho}_1 \int_{\mathcal{Z}} V_{1|2}(\mathbf{z}) \mathrm{d}\mu(\mathbf{z}) + v_* M_{\mathrm{ex}}^{(1)},$$

and

$$\int_{\mathcal{Z}} \rho_2(\mathbf{z}) \log(\rho_1(\mathbf{z})) \mathrm{d}\mu(\mathbf{z}) \geq \int_{\mathcal{Z}} \rho_2(\mathbf{z}) \log \underline{\rho}_1 \mathrm{d}\mu(\mathbf{z}) = \log \underline{\rho}_1.$$

Plugging these into Eq. (66) yields the lower bound

$$\mathcal{F}_1(\rho_1) \geq \tfrac{1}{2} \left( \underline{\rho}_1 \int_{\mathcal{Z}} V_{1|2}(\mathbf{z}) \mathrm{d}\mu(\mathbf{z}) + v_* M_{\mathrm{ex}}^{(1)} \right) + \tfrac{1}{2} \log \underline{\rho}_1 + C(\rho_2) =: \mathcal{F}_*. \tag{67}$$

Therefore,

$$\inf_{\rho_1 \in \mathcal{P}_{\mu,\underline{\rho}_1}(\mathcal{Z})} \mathcal{F}_1(\rho_1) \geq \mathcal{F}_*.$$

**Step 2: Evaluate the coordinate map $\mathcal{F}_1$ at $\rho_1^{(\sigma)}$.** For each $\sigma > 0$, define the sublevel set

$$\mathcal{W}_1^\sigma = \{\mathbf{z} \in \mathcal{Z} : V_{1|2}(\mathbf{z}) \le v_* + \sigma\}, \qquad \text{with} \quad \mu(\mathcal{W}_1^\sigma) > 0.$$

Further, since $\mu$ is the volume measure on the compact manifold $\mathcal{Z}$, $\mu$ is non-atomic. Thus, there exist a shrinking subset $\mathcal{S}_1^\sigma \subseteq \mathcal{W}_1^\sigma$ satisfying $\mu(\mathcal{S}_1^\sigma) \xrightarrow{\sigma \to 0} 0$ (when $\mathcal{W}_1^\sigma$ itself is sharp $\mu(\mathcal{W}_1^\sigma) \xrightarrow{\sigma \to 0} 0$, we simply choose $\mathcal{S}_1^\sigma = \mathcal{W}_1^\sigma$). Define the density as

$$\rho_1^{(\sigma)}(\mathbf{z}) = \underline{\rho}_1 + \mathbb{1}_{\mathcal{S}_1^\sigma}(\mathbf{z}) \frac{M_{\mathrm{ex}}^{(1)}}{\mu(\mathcal{S}_1^\sigma)}.$$

Since $\rho_1^{(\sigma)} \ge \underline{\rho}_1$ and

$$\int_{\mathcal{Z}} \rho_1^{(\sigma)} \mathrm{d}\mu = \underline{\rho}_1 \mu(\mathcal{Z}) + \frac{M_{\mathrm{ex}}^{(1)}}{\mu(\mathcal{S}_1^\sigma)} \mu(\mathcal{S}_1^\sigma) = \underline{\rho}_1 \mu(\mathcal{Z}) + M_{\mathrm{ex}}^{(1)} = 1,$$

we have $\rho_1^{(\sigma)} \in \mathcal{P}_{\mu, \underline{\rho}_1}(\mathcal{Z})$.

Now, evaluate the coordinate map $\mathcal{F}_1$ at $\rho_1^{(\sigma)}$:

*(i) Linear term.* Since $\mathcal{S}_1^\sigma \subseteq \mathcal{W}_1^\sigma$, we have $V_{1|2}(\mathbf{z}) \le v_* + \sigma$ on $\mathcal{S}_1^\sigma$, hence

$$\int_{\mathcal{Z}} \rho_1^{(\sigma)} V_{1|2}(\mathbf{z}) \mathrm{d}\mu = \underline{\rho}_1 \int_{\mathcal{Z}} V_{1|2} \mathrm{d}\mu + \frac{M_{\mathrm{ex}}^{(1)}}{\mu(\mathcal{S}_1^\sigma)} \int_{\mathcal{S}_1^\sigma} V_{1|2} \mathrm{d}\mu \le \underline{\rho}_1 \int_{\mathcal{Z}} V_{1|2} \mathrm{d}\mu + M_{\mathrm{ex}}^{(1)}(v_* + \sigma).$$

*(ii) Log term.* Split the domain $\mathcal{Z} = (\mathcal{Z} \setminus \mathcal{S}_1^\sigma) \cup \mathcal{S}_1^\sigma$:

$$\int_{\mathcal{Z}} \rho_2 \log \rho_1^{(\sigma)} \mathrm{d}\mu = \int_{\mathcal{Z} \setminus \mathcal{S}_1^\sigma} \rho_2 \log \underline{\rho}_1 \, \mathrm{d}\mu + \int_{\mathcal{S}_1^\sigma} \rho_2 \log \left( \underline{\rho}_1 + \frac{M_{\mathrm{ex}}^{(1)}}{\mu(\mathcal{S}_1^\sigma)} \right) \mathrm{d}\mu.$$

Using the mass of the spike $m_\sigma := \int_{\mathcal{S}_1^\sigma} \rho_2 \, \mathrm{d}\mu$, the first term is $(1 - m_\sigma) \log \underline{\rho}_1$. Therefore,

$$\int \rho_2 \log \rho_1^{(\sigma)} \mathrm{d}\mu = \log \underline{\rho}_1 + m_\sigma \left( \log \left( \underline{\rho}_1 + \frac{M_{\mathrm{ex}}^{(1)}}{\mu(\mathcal{S}_1^\sigma)} \right) - \log \underline{\rho}_1 \right) = \log \underline{\rho}_1 + m_\sigma L_\sigma, \quad \text{where} \quad L_\sigma := \log \left( 1 + \frac{M_{\mathrm{ex}}^{(1)}}{\underline{\rho}_1 \mu(\mathcal{S}_1^\sigma)} \right).$$

Combining terms yields the inequality

$$\mathcal{F}_1(\rho_1^{(\sigma)}) \le \tfrac{1}{2} \left( \underline{\rho}_1 \int V_{1|2} \mathrm{d}\mu + M_{\mathrm{ex}}^{(1)}(v_* + \sigma) \right) + \tfrac{1}{2} \left( \log \underline{\rho}_1 + m_\sigma L_\sigma \right) + C(\rho_2). \tag{68}$$

**Step 3: Conclude convergence.** We first analyze the limit of the log-penalty $m_\sigma L_\sigma \to 0$ as $\sigma \to 0$. Since $\rho_2 \in L^1(\mu)$ and $\mu(\mathcal{Z}) < \infty$, the integral is absolutely continuous: for every $\varepsilon > 0$ there exists $\delta(\varepsilon) > 0$ such that for any measurable $\mathcal{A} \subset \mathcal{Z}$ with $\mu(\mathcal{A}) \le \delta(\varepsilon)$,

$$\int_{\mathcal{A}} \rho_2 \, \mathrm{d}\mu \le \varepsilon.$$

Define the modulus of absolute continuity

$$\Psi(t) := \sup \left\{ \int_{\mathcal{A}} \rho_2 \, \mathrm{d}\mu : \ \mathcal{A} \subset \mathcal{Z} \text{ measurable}, \ \mu(\mathcal{A}) \le t \right\}.$$

Since $\rho_2 \in L^1(\mu)$ and $\mu(\mathcal{Z}) < \infty$, we have $\Psi(t) \to 0$ as $t \to 0$. Since $\Psi(t) \to 0$ as $t \to 0$ and $|\log t|^{-2} \to 0$ as $t \to 0$, we may choose a sequence $\delta_\sigma \to 0$ such that

$$\delta_\sigma \le \mu(\mathcal{W}_1^\sigma), \qquad \Psi(\delta_\sigma) \le |\log \delta_\sigma|^{-2}.$$

By non-atomicity of $\mu$, choose $\mathcal{S}_1^\sigma \subseteq \mathcal{W}_1^\sigma$ with $\mu(\mathcal{S}_1^\sigma) = \delta_\sigma$. Then

$$m_\sigma := \int_{\mathcal{S}_1^\sigma} \rho_2 \, \mathrm{d}\mu \le \Psi(\delta_\sigma) \le \frac{1}{|\log \delta_\sigma|^2}.$$

With $L_\sigma = \log\left(1 + M_{\text{ex}}^{(1)}/(\underline{\rho}_1 \, \mu(\mathcal{S}_1^\sigma))\right)$ and $\mu(\mathcal{S}_1^\sigma) = \delta_\sigma$, we have

$$L_\sigma \le \log\left(1 + M_{\text{ex}}^{(1)}/\underline{\rho}_1\right) + |\log \delta_\sigma| =: K + |\log \delta_\sigma|.$$

Therefore,

$$0 \le m_\sigma L_\sigma \le \frac{1}{|\log \delta_\sigma|^2}\Big(K + |\log \delta_\sigma|\Big) = \frac{K}{|\log \delta_\sigma|^2} + \frac{1}{|\log \delta_\sigma|} \xrightarrow{\sigma \to 0} 0.$$

Thus, the additional log-penalty incurred by the spike vanishes.

Now, subtract $\mathcal{F}_*$ (Eq. (67)) from Eq. (68):

$$\mathcal{F}_1(\rho_1^{(\sigma)}) - \mathcal{F}_* \le \tfrac{1}{2}(M_{\text{ex}}^{(1)}\sigma + m_\sigma L_\sigma).$$

Letting $\sigma \to 0$ gives

$$\limsup_{\sigma \to 0} \mathcal{F}_1(\rho_1^{(\sigma)}) \le \mathcal{F}_*.$$

Meanwhile, Step 1 showed $\inf_{\rho_1 \in \mathcal{P}_{\mu,\underline{\rho}_1}(\mathcal{Z})} \mathcal{F}_1(\rho_1) \ge \mathcal{F}_*$. Therefore

$$\inf_{\rho_1 \in \mathcal{P}_{\mu,\underline{\rho}_1}(\mathcal{Z})} \mathcal{F}_1(\rho_1) = \mathcal{F}_* \qquad \text{and} \qquad \lim_{\sigma \to 0} \mathcal{F}_1(\rho_1^{(\sigma)}) = \inf_{\rho_1 \in \mathcal{P}_{\mu,\underline{\rho}_1}(\mathcal{Z})} \mathcal{F}_1(\rho_1),$$

concluding that the infimum is approached by the boundary solutions concentrating on the minima of the effective potential field $V_{1|2} = \frac{1}{\tau} U_{1 \to 2} + \log \rho_2$.

The $\rho_2$-coordinate statement is identical after swapping indices and using $V_{2|1} = \frac{1}{\tau} U_{2 \to 1} + \log \rho_1$. $\qquad\square$

### D.3. Proof of Thm. 4.2

**Theorem 4.2** (Low-temperature multimodal energy consistency). *Assume Asms. 2.1, 3.2 and 4.1. Fix any $(\theta, \phi) \in \Theta \times \Phi$, and assume $q_\theta \ll \mu$, $q_\phi \ll \mu$ with continuous densities $\rho_\theta, \rho_\phi$ bounded below: $\inf_{\mathbf{z} \in \mathcal{Z}} \rho_\theta(\mathbf{z}) \ge \underline{\rho}_\theta > 0$, $\inf_{\mathbf{z} \in \mathcal{Z}} \rho_\phi(\mathbf{z}) \ge \underline{\rho}_\phi > 0$. Let $\varepsilon_{\text{kde}}^{(\theta)}(\tau) := \|\tilde{\rho}_{\theta,\tau} - \rho_\theta\|_\infty$ and $\varepsilon_{\text{kde}}^{(\phi)}(\tau) := \|\tilde{\rho}_{\phi,\tau} - \rho_\phi\|_\infty$. Then there exists $\tau_0(\theta, \phi) > 0$ such that for all $0 < \tau \le \tau_0(\theta, \phi)$,*

$$\left| \mathcal{J}_\tau^{\text{Sym}}(\theta, \phi) - \mathcal{F}_{\tau, \mathbf{U}_{\theta,\phi}}^{\text{Sym}}(\rho_\theta, \rho_\phi) \right| \le \frac{\varepsilon_{\text{kde}}^{(\theta)}(\tau) + \varepsilon_{\text{kde}}^{(\phi)}(\tau)}{\min\{\underline{\rho}_\theta, \underline{\rho}_\phi\}},$$

*where $\mathcal{F}_{\tau, \mathbf{U}_{\theta,\phi}}^{\text{Sym}}(\rho_\theta, \rho_\phi)$ is the intrinsic multimodal energy from Def. 4.3, evaluated on $\mathcal{P}_{\mu,\underline{\rho}_\theta}(\mathcal{Z}) \times \mathcal{P}_{\mu,\underline{\rho}_\phi}(\mathcal{Z})$, with $\mathbf{U}_{\theta,\phi} := (U_{\theta \to \phi}, U_{\phi \to \theta})$. In particular,*

$$\left| \mathcal{J}_\tau^{\text{Sym}}(\theta, \phi) - \mathcal{F}_{\tau, \mathbf{U}_{\theta,\phi}}^{\text{Sym}}(\rho_\theta, \rho_\phi) \right| \to 0, \qquad \text{as } \tau \to 0^+.$$

☞ **Proof intuition.** This proof establishes that in the low-temperature limit, the empirical cross-modal parametric energy mirrors the theoretical, dual-density thermodynamic free energy. The argument relies on an algebraic cancellation and the geometric stability of the kernel smoothing: **(i)** We first decompose both the learned objective and the theoretical free energy into their core components: the cross-modal binding potentials and the internal entropies. When we calculate the exact difference between the two functionals, these complex binding and entropy terms cancel out. The remaining discrepancy is entirely isolated to the KL divergence between the true marginal densities and their kernel-smoothed counterparts. **(ii)** To prove this residual discrepancy vanishes, we rely on the boundary of the true marginals. Because the densities are bounded away from zero, the logarithmic ratio inside the KL divergence operates in a strictly Lipschitz-continuous regime. This allows us to analytically bound the complex integral by the maximum worst-case estimation error (the KDE mismatch) of each modality. **(iii)** Because the kernel smoothing of both the vision and language manifolds independently converges to the true underlying densities as $\tau \to 0^+$, their respective KDE mismatches vanish. This crushes the KL divergence between each modality's raw density and the smoothed density to zero, mathematically guaranteeing that the learned cross-modal energy landscape is indistinguishable from the true thermodynamic landscape.

*Proof.* Fix any $(\theta, \phi) \in \Theta \times \Phi$. Under Asm. 4.1, $s$ is bounded and Borel measurable, hence both cross-modal potentials $U_{\theta \rightarrow \phi}$ and $U_{\phi \rightarrow \theta}$ from Def. 4.1 are Borel measurable and bounded. Moreover, since $\mathcal{Z}$ is compact and $\rho_\theta, \rho_\phi$ are continuous with $\rho_\theta, \rho_\phi \geq \min\{\underline{\rho}_\theta, \underline{\rho}_\phi\} > 0$, both $\log \rho_\theta$ and $\log \rho_\phi$ are bounded, hence $D_{\mathrm{KL}}(\rho_\theta \| \rho_\phi)$ and $D_{\mathrm{KL}}(\rho_\phi \| \rho_\theta)$ are finite. Consequently, the multimodal functional $\mathcal{F}^{\mathrm{Sym}}_{\tau, \mathbf{U}_{\theta, \phi}}(q_\theta, q_\phi)$ is well-defined.

**Step 1: Functional discrepancy reduces to log-ratio perturbations.** Recall the directional multimodal parametric energies from Def. 4.2:

$$\mathcal{J}^{\theta \rightarrow \phi}_\tau(\theta, \phi) = \tfrac{1}{\tau} \mathcal{U}^{\theta \rightarrow \phi}(q_\theta) - H_\times(q_\theta, \tilde{\rho}_{\phi, \tau}), \qquad \mathcal{J}^{\phi \rightarrow \theta}_\tau(\theta, \phi) = \tfrac{1}{\tau} \mathcal{U}^{\phi \rightarrow \theta}(q_\phi) - H_\times(q_\phi, \tilde{\rho}_{\theta, \tau}),$$

and $\mathcal{J}^{\mathrm{Sym}}_\tau = \tfrac{1}{2}(\mathcal{J}^{\theta \rightarrow \phi}_\tau + \mathcal{J}^{\phi \rightarrow \theta}_\tau)$. Using $H_\times(q_1, \rho_2) = H(\rho_1) + D_{\mathrm{KL}}(\rho_1 \| \rho_2)$ with $\rho_1 = \mathrm{d}q_1 / \mathrm{d}\mu$, we can rewrite

$$\mathcal{J}^{\mathrm{Sym}}_\tau(\theta, \phi) = \tfrac{1}{2}\Big(\mathcal{F}_{\tau, U_{\theta \rightarrow \phi}}(\rho_\theta) + \mathcal{F}_{\tau, U_{\phi \rightarrow \theta}}(\rho_\phi)\Big) - \tfrac{1}{2}\Big(D_{\mathrm{KL}}(\rho_\theta \| \tilde{\rho}_{\phi, \tau}) + D_{\mathrm{KL}}(\rho_\phi \| \tilde{\rho}_{\theta, \tau})\Big). \tag{69}$$

On the other hand, by Def. 4.3 with $D^{\mathrm{Sym}}_{\mathrm{KL}}(\rho_\theta, \rho_\phi) = \tfrac{1}{2}(D_{\mathrm{KL}}(\rho_\theta \| \rho_\phi) + D_{\mathrm{KL}}(\rho_\phi \| \rho_\theta))$,

$$\mathcal{F}^{\mathrm{Sym}}_{\tau, \mathbf{U}_{\theta, \phi}}(\rho_\theta, \rho_\phi) = \tfrac{1}{2}\big(\mathcal{F}_{\tau, U_{\theta \rightarrow \phi}}(\rho_\theta) + \mathcal{F}_{\tau, U_{\phi \rightarrow \theta}}(\rho_\phi)\big) - D^{\mathrm{Sym}}_{\mathrm{KL}}(\rho_\theta, \rho_\phi). \tag{70}$$

Subtracting Eq. (70) from Eq. (69), the potential and entropy terms cancel, giving the exact identity

$$\begin{aligned} \mathcal{F}^{\mathrm{Sym}}_{\tau, \mathbf{U}_{\theta, \phi}}(\rho_\theta, \rho_\phi) - \mathcal{J}^{\mathrm{Sym}}_\tau(\theta, \phi) = \\ \tfrac{1}{2}\big(D_{\mathrm{KL}}(\rho_\theta \| \tilde{\rho}_{\phi, \tau}) - D_{\mathrm{KL}}(\rho_\theta \| \rho_\phi) + D_{\mathrm{KL}}(\rho_\phi \| \tilde{\rho}_{\theta, \tau}) - D_{\mathrm{KL}}(\rho_\phi \| \rho_\theta)\big). \end{aligned} \tag{71}$$

Now, use the elementary difference formula: for any $\mu$-densities $p, r_1, r_2$ with $r_1, r_2 > 0$ $\mu$-a.e.,

$$D_{\mathrm{KL}}(p \| r_1) - D_{\mathrm{KL}}(p \| r_2) = \int_{\mathcal{Z}} p \log \frac{r_2}{r_1} \, \mathrm{d}\mu.$$

Since $\kappa_\tau > 0$, $\rho_\theta, \rho_\phi$ are probability densities, and $\tilde{\rho}_{\theta, \tau}, \tilde{\rho}_{\phi, \tau} > 0$ everywhere. Apply it to $(p, r_1, r_2) = (\rho_\theta, \tilde{\rho}_{\phi, \tau}, \rho_\phi)$ and $(\rho_\phi, \tilde{\rho}_{\theta, \tau}, \rho_\theta)$ to obtain

$$\mathcal{F}^{\mathrm{Sym}}_{\tau, \mathbf{U}_{\theta, \phi}}(\rho_\theta, \rho_\phi) - \mathcal{J}^{\mathrm{Sym}}_\tau(\theta, \phi) = \tfrac{1}{2}\Big(\int \rho_\theta \log \frac{\rho_\phi}{\tilde{\rho}_{\phi, \tau}} \, \mathrm{d}\mu + \int \rho_\phi \log \frac{\rho_\theta}{\tilde{\rho}_{\theta, \tau}} \, \mathrm{d}\mu\Big). \tag{72}$$

**Step 2: Nonasymptotic bound by KDE sup-errors.** By the lower bounds $\rho_\theta \geq \underline{\rho}_\theta$ and $\rho_\phi \geq \underline{\rho}_\phi$ and the Lem. B.3 applied separately to $\rho_\theta$ and $\rho_\phi$, there exists $\tau_0(\theta, \phi) > 0$ such that for all $0 < \tau \leq \tau_0(\theta, \phi)$,

$$\varepsilon^{(\theta)}_{\mathrm{kde}}(\tau) = \|\tilde{\rho}_{\theta, \tau} - \rho_\theta\|_\infty \leq \frac{\underline{\rho}_\theta}{2}, \qquad \varepsilon^{(\phi)}_{\mathrm{kde}}(\tau) = \|\tilde{\rho}_{\phi, \tau} - \rho_\phi\|_\infty \leq \frac{\underline{\rho}_\phi}{2}.$$

Hence, for all $\mathbf{z} \in \mathcal{Z}$ and all such $\tau$,

$$\tilde{\rho}_{\theta, \tau}(\mathbf{z}) \geq \frac{\underline{\rho}_\theta}{2}, \qquad \tilde{\rho}_{\phi, \tau}(\mathbf{z}) \geq \frac{\underline{\rho}_\phi}{2}. \tag{73}$$

On $[\underline{\rho}_\phi/2, \infty)$, the map $\log(\cdot)$ is Lipschitz with constant $2/\underline{\rho}_\phi$; thus, for all $\mathbf{z}$,

$$|\log \rho_\phi(\mathbf{z}) - \log \tilde{\rho}_{\phi, \tau}(\mathbf{z})| \leq \frac{2}{\underline{\rho}_\phi} |\rho_\phi(\mathbf{z}) - \tilde{\rho}_{\phi, \tau}(\mathbf{z})| \leq \frac{2\, \varepsilon^{(\phi)}_{\mathrm{kde}}(\tau)}{\underline{\rho}_\phi}.$$

Equivalently,

$$\left|\log \frac{\rho_\phi(\mathbf{z})}{\tilde{\rho}_{\phi, \tau}(\mathbf{z})}\right| \leq \frac{2\, \varepsilon^{(\phi)}_{\mathrm{kde}}(\tau)}{\underline{\rho}_\phi}. \tag{74}$$

Analogously,

$$\left|\log \frac{\rho_\theta(\mathbf{z})}{\tilde{\rho}_{\theta, \tau}(\mathbf{z})}\right| \leq \frac{2\, \varepsilon^{(\theta)}_{\mathrm{kde}}(\tau)}{\underline{\rho}_\theta}. \tag{75}$$

Now, bound Eq. (72) using $\int_{\mathcal{Z}} \rho_\theta \, \mathrm{d}\mu = \int_{\mathcal{Z}} \rho_\phi \, \mathrm{d}\mu = 1$:

$$\left| \mathcal{F}^{\mathrm{Sym}}_{\tau, \mathbf{U}_{\theta,\phi}}(\rho_\theta, \rho_\phi) - \mathcal{J}^{\mathrm{Sym}}_\tau(\theta, \phi) \right| \leq \tfrac{1}{2} \left( \int \rho_\theta \left| \log \frac{\rho_\phi}{\tilde{\rho}_{\phi,\tau}} \right| \mathrm{d}\mu + \int \rho_\phi \left| \log \frac{\rho_\theta}{\tilde{\rho}_{\theta,\tau}} \right| \mathrm{d}\mu \right)$$

$$\leq \tfrac{1}{2} \left( \left\| \log \frac{\rho_\phi}{\tilde{\rho}_{\phi,\tau}} \right\|_\infty + \left\| \log \frac{\rho_\theta}{\tilde{\rho}_{\theta,\tau}} \right\|_\infty \right) \leq \tfrac{1}{2} \left( \frac{2\,\varepsilon^{(\phi)}_{\mathrm{kde}}(\tau)}{\underline{\rho}_\phi} + \frac{2\,\varepsilon^{(\theta)}_{\mathrm{kde}}(\tau)}{\underline{\rho}_\theta} \right) \leq \frac{\varepsilon^{(\theta)}_{\mathrm{kde}}(\tau) + \varepsilon^{(\phi)}_{\mathrm{kde}}(\tau)}{\min\{\underline{\rho}_\theta, \underline{\rho}_\phi\}}.$$

This proves the claimed nonasymptotic bound for all $0 < \tau \leq \tau_0(\theta, \phi)$.

**Step 3: Vanishing discrepancy as $\tau \to 0^+$.** By Lem. B.3 applied to $\rho_\theta$ and $\rho_\phi$ (using Asm. 3.2),

$$\varepsilon^{(\theta)}_{\mathrm{kde}}(\tau) \xrightarrow{\tau \to 0^+} 0, \qquad \varepsilon^{(\phi)}_{\mathrm{kde}}(\tau) \xrightarrow{\tau \to 0^+} 0.$$

Therefore, the bound in Step 2 implies

$$\left| \mathcal{J}^{\mathrm{Sym}}_\tau(\theta, \phi) - \mathcal{F}^{\mathrm{Sym}}_{\tau, \mathbf{U}_{\theta,\phi}}(\rho_\theta, \rho_\phi) \right| \xrightarrow{\tau \to 0^+} 0,$$

for any fixed pair $(\theta, \phi) \in \Theta \times \Phi$. $\qquad\qquad\square$

# E. Experimental Details

This section details the experimental procedures used to empirically validate our theoretical framework. Specifically, we provide comprehensive setups for evaluating: (i) the convergence of finite-batch InfoNCE gradients to their deterministic counterparts; (ii) the Gibbs-shaped concentration and spread behavior of unimodal training under sharp kernels; (iii) the emergence of a persistent, population-level modality gap induced by conditional heterogeneity; and (iv) the real-world implications of cross-modal divergence via probing and fine-tuning CLIP models on MS-COCO.

## E.1. Large-Batch Gradient Consistency Across Critics

This experiment provides a numerical validation of the large-batch consistency established in Thm. 3.1. We verify that as the number of negatives $N$ increases, the stochastic InfoNCE descent direction aligns increasingly well with the deterministic energy gradient. Crucially, since the multimodal symmetric objective (Eq. (2)) is the sum of two directional terms, establishing consistency for the unimodal gradient implies consistency for the full multimodal objective by linearity.

**Data and pairing.** We construct a synthetic unimodal dataset in $\mathcal{X} \subset \mathbb{R}^m$ ($m = 64$) from a mixture of $K = 4$ Gaussians: $\mathbf{x} \sim \sum_{k=1}^{K} \pi_k \mathcal{N}(\boldsymbol{\mu}_k, \sigma^2 \mathbf{I})$. The component means $\{\boldsymbol{\mu}_k\}$ are random directions scaled by a separation factor of 4, with $\sigma = 1$. Positive pairs are generated via an additive noise $\tilde{\mathbf{x}} = \mathbf{x} + \boldsymbol{\varepsilon}$, where $\boldsymbol{\varepsilon} \sim \mathcal{N}(\mathbf{0}, \sigma^2_{\mathrm{aug}} \mathbf{I})$. Negative samples $\{\mathbf{x}^-_j\}_{j=1}^N$ are i.i.d. draws from the same data distribution.

**Encoder and regimes.** We employ a linear encoder $f_{\mathbf{W}}(\mathbf{x})$ with parameters $\mathbf{W} \in \mathbb{R}^{n \times m}$ ($n = 128$) under two distinct geometric regimes:

$$\mathbf{z} = f_{\mathbf{W}}(\mathbf{x}) = \begin{cases} \mathrm{normalize}(\mathbf{W}\mathbf{x}) \in \mathbb{S}^{n-1}, & \text{(Spherical regime)} \\ \tanh(\mathbf{W}\mathbf{x}) \in [-1, 1]^n, & \text{(Compact Euclidean regime).} \end{cases}$$

The spherical case corresponds to the standard feature-normalized setting, while the compact Euclidean case mirrors our theoretical assumption of a compact $\mathcal{Z}$ without enforcing explicit normalization, allowing for distance-based critics.

We evaluate the exponential kernel defined in § 2 with the two canonical critics: the cosine critic $s_{\cos}(\mathbf{z}, \mathbf{w}) = \langle \mathbf{z}, \mathbf{w} \rangle$ and the RBF critic $s_{\mathrm{rbf}}(\mathbf{z}, \mathbf{w}) = -\frac{1}{n} \|\mathbf{z} - \mathbf{w}\|^2$, with temperatures $\tau_{\cos} = 0.1$ and $\tau_{\mathrm{rbf}} = 1.0$, respectively. The RBF critic includes $1/n$ scaling to maintain $O(1)$ logits. We set augmentation noise $\sigma_{\mathrm{aug}} = 0.05$ for the cosine regime and $\sigma_{\mathrm{aug}} = 0.2$ for the RBF regime.

**Loss and gradients.** For each seed, we sample a fixed batch of $B = 64$ anchors/positives and a large "reference" pool of $N_{\text{ref}} = 4096$ negatives. For a given $N$, we compute the loss $\hat{\ell}_{B,N}(\mathbf{W})$ using the first $N$ negatives from this pool:

$$\hat{\ell}_{B,N}(\mathbf{W}) = \frac{1}{B} \sum_{i=1}^{B} \left[ -\frac{1}{\tau} s(\mathbf{z}_i, \tilde{\mathbf{z}}_i) + \log\left( \exp\left( s(\mathbf{z}_i, \tilde{\mathbf{z}}_i)/\tau \right) + \sum_{j=1}^{N} \exp\left( s(\mathbf{z}_i, \mathbf{w}_j^-)/\tau \right) \right) \right].$$

Let $\mathbf{g}_N := \nabla_{\mathbf{W}} \hat{\ell}_{B,N}(\mathbf{W})$ denote the gradient estimate using $N$ negatives, and $\mathbf{g}_{\text{ref}} := \mathbf{g}_{N_{\text{ref}}}$ denote the high-fidelity reference gradient. Fixing the negative pool ensures that variations in $\mathbf{g}_N$ are attributable solely to the sample size $N$.

**Metrics and results.** We sweep $N \in \{4, 8, 16, 32, 64, 128, 256, 512, 1024\}$ and report two metrics: (i) *gradient alignment* $\cos(\mathbf{g}_N, \mathbf{g}_{\text{ref}})$, measuring directional accuracy; and (ii) *relative error* $\|\mathbf{g}_N - \mathbf{g}_{\text{ref}}\|/\|\mathbf{g}_{\text{ref}}\|$, measuring magnitude deviation. Results are averaged over 20 random seeds; error bars indicate $\pm 1$ standard deviation across seeds. As shown in Fig. 3, across both the spherical and RBF regimes, the gradient alignment score increases monotonically with $N$ while the relative error decreases. This confirms our theoretical claims: as the denominator concentrates, the stochastic InfoNCE gradient converges to the deterministic energy gradient. The rapid convergence (alignment $> 0.95$ at $N = 256$) suggests that the large-batch energy landscape is a faithful proxy for practical training dynamics even at moderate batch sizes.

### E.2. Unimodal Gibbs Equilibrium and Low-Temperature Concentration

This experiment provides a numerical verification of the intrinsic unimodal geometry characterized in § 3. Specifically, we validate the variational perspective by demonstrating that: (i) for fixed $\tau > 0$, the functional $\mathcal{F}_{\tau,U}$ admits a unique Gibbs-type equilibrium (Prop. 3.1); and (ii) as $\tau \to 0^+$, this equilibrium concentrates exponentially around the minimizers of $U$, confirming the ground-state convergence predicted in Prop. 3.2.

**Representation space and potential field.** We define the domain as the unit sphere $\mathcal{Z} = \mathbb{S}^2 \subset \mathbb{R}^3$ equipped with surface measure $\mu$ and geodesic distance $d_{\mathbb{S}^2}$. We construct a smooth, non-isotropic potential $U(\mathbf{z})$ using a log-sum-exp mixture of two von-Mises–Fisher-like modes:

$$U(\mathbf{z}) := -\frac{1}{\gamma} \log\left( w \exp(\kappa\langle \mathbf{z}, \mathbf{m}_1\rangle) + (1-w) \exp(\kappa\langle \mathbf{z}, \mathbf{m}_2\rangle) \right), \qquad \mathbf{z} \in \mathbb{S}^2, \tag{76}$$

with $\gamma = 12$, concentration $\kappa = 12$, and mixture weight $w = 0.5$. The modes are centered at $\mathbf{m}_1 = (0, 0, 1)$ and a separated direction $\mathbf{m}_2 = (0.85, 0.15, -0.50)/\|(0.85, 0.15, -0.50)\|$, creating two distinct low-energy basins (Fig. 7, left top).

**Particle training objective.** To emulate minimization of $\mathcal{F}_{\tau,U}(\rho) = \frac{1}{\tau} \int_{\mathcal{Z}} U \rho \, d\mu - H(\rho)$ (Def. 3.3) without committing to a parametric encoder, we represent $\rho$ by $M$ particles $\{\mathbf{z}_i\}_{i=1}^{M} \subset \mathbb{S}^2$ and optimize a standard particle/KDE surrogate. We parameterize unconstrained variables $\{\mathbf{v}_i\}_{i=1}^{M} \subset \mathbb{R}^3$ and project to the sphere $\mathbf{z}_i = \mathbf{v}_i/\|\mathbf{v}_i\|$ at each iteration. We approximate the density at particle locations by a wrapped Gaussian kernel on the sphere,

$$\hat{\rho}_h(\mathbf{z}_i) := \frac{1}{M} \sum_{j=1}^{M} \exp\left( -\frac{d_{\mathbb{S}^2}(\mathbf{z}_i, \mathbf{z}_j)^2}{2h^2} \right), \tag{77}$$

with bandwidth $h = 0.35$. The optimized objective is the particle approximation

$$\widehat{\mathcal{F}}_{\tau,U}(\{\mathbf{z}_i\}) := \frac{1}{\tau} \cdot \frac{1}{M} \sum_{i=1}^{M} U(\mathbf{z}_i) + \frac{1}{M} \sum_{i=1}^{M} \log \hat{\rho}_h(\mathbf{z}_i), \tag{78}$$

where the second term is the KDE-based surrogate for $-H(\rho)$ up to an additive constant. We minimize Eq. (78) using Adam (Kingma & Ba, 2015) for 5000 steps with learning rate $5 \times 10^{-2}$, while injecting Gaussian noise from $\mathcal{N}(\mathbf{0}, \sigma^2 \mathbf{I})$ with $\sigma = 0.06$ at each iteration. This injection approximates a Langevin diffusion process (Welling & Teh, 2011), ensuring that the particles maintain active exploration of the landscape to escape shallow local traps, while the gradient of the entropy term drives the global dispersion.

We sweep $\tau \in \{10, 5.0, 2.5, 1.0, 0.5, 0.2, 0.1\}$ and run 20 seeds. As a reference, we estimate the Gibbs equilibrium $\rho_\tau^\star(\mathbf{z}) \propto \exp(-U(\mathbf{z})/\tau)$ (Prop. 3.1) by importance sampling: we draw $n_{\text{mc}} = 120{,}000$ uniform points on $\mathbb{S}^2$, weight by $\exp(-U/\tau)$, and compute weighted expectations. For visualization, we use a separate pool of 24,000 uniform points and perform weighted sampling without replacement to draw 2400 Gibbs points per $\tau$.

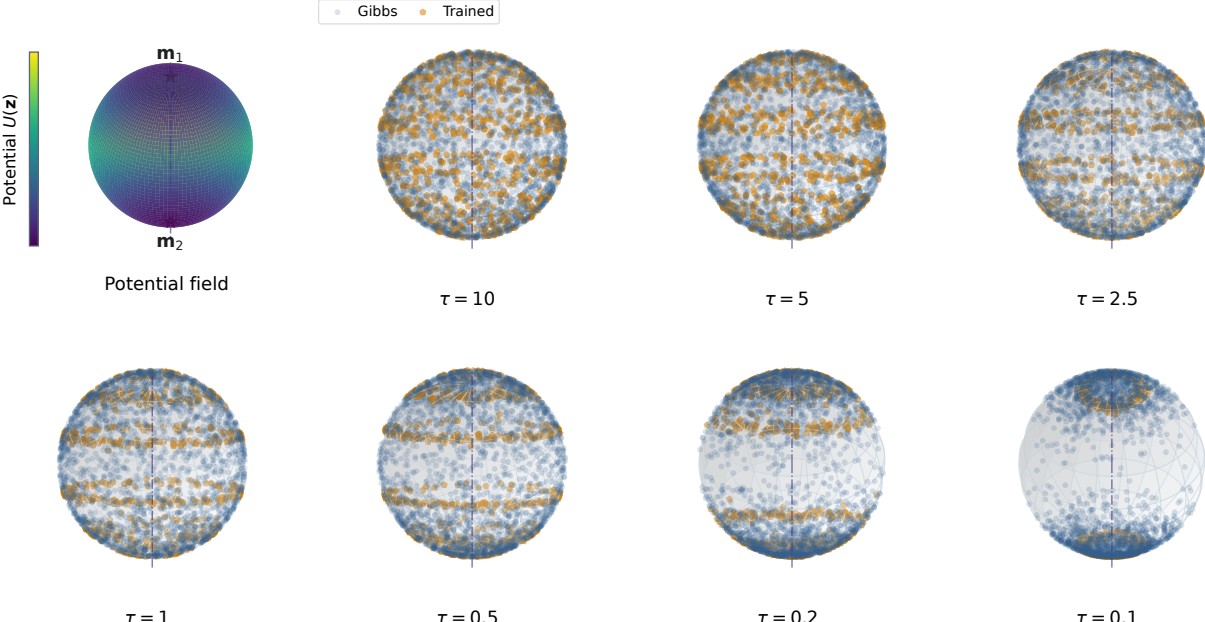

*Figure 7.* Unimodal potential landscape on $\mathbb{S}^2$ and equilibria across temperature $\tau$. Left top: the 2-well potential $U$ in Eq. (76) (colored by value), with minima centers $\mathbf{m}_1, \mathbf{m}_2$. Others: Gibbs samples (blue; importance-resampled) and trained particles (orange; minimizing Eq. (78)) across various temperatures. As $\tau$ decreases, both distributions concentrate around the low-energy wells.

**Evaluation metric.** To quantify low-$\tau$ concentration, we report the *cap mass* within geodesic radius $\varepsilon = 0.50$ around either potential well: $\mathrm{CapMass}_\varepsilon(\rho) := (\rho\,\mu)\big(\mathcal{C}_\varepsilon(\mathbf{m}_1) \cup \mathcal{C}_\varepsilon(\mathbf{m}_2)\big)$, where $\mathcal{C}_\varepsilon(\mathbf{m}) := \{\mathbf{z} \in \mathbb{S}^2 : \ d_{\mathbb{S}^2}(\mathbf{z}, \mathbf{m}) \le \varepsilon\}$. For the trained particles, this is simply the fraction of particles whose distance to $\{\mathbf{m}_1, \mathbf{m}_2\}$ is at most $\varepsilon$. For the Gibbs baseline, we compute the same quantity as a weighted Monte Carlo estimate. We report the mean $\pm$ standard error over the 20 seeds.

**Results.** Fig. 7 visualizes the potential landscape and the learned equilibrium across $\tau$. At high temperature (e.g., $\tau = 10$), both the Gibbs samples and the trained particles spread broadly over the sphere: the entropy term in $\mathcal{F}_{\tau,U}$ is comparatively strong, producing a diffuse equilibrium consistent with the "entropic dispersion" force. As $\tau$ decreases (e.g., $\tau = 0.2, \tau = 0.1$), the potential term $\frac{1}{\tau}\int_{\mathcal{Z}} U\rho\,\mathrm{d}\mu$ dominates, and both distributions increasingly concentrate near the two low-energy basins. Notably, the trained particles exhibit semi-regular lattice-like bands, in contrast to the purely random scatter of the Gibbs samples. This micro-structure is an intrinsic consequence of the optimization: the KDE entropy term induces a repulsive force between particles, causing them to self-organize into low-discrepancy packing configurations to maximize mutual separation. This behavior is a known feature of particle-based variational inference, where it effectively yields a variance-reduced discretization of the target density (Liu & Wang, 2016). Crucially, this microscopic regularity does not violate the macroscopic prediction: at $\tau = 0.1$, the global mass is visibly localized around the wells, matching the qualitative prediction of ground-state concentration (Prop. 3.2).

This trend is quantified in Fig. 2a: the cap mass $\mathrm{CapMass}_\varepsilon(\rho)$ increases as $\tau$ decreases for both the Gibbs baseline and the trained particles. The trained curve closely tracks the trend, with small deviations attributable to finite particle count $M$, finite optimization time, and the fixed KDE bandwidth $h$, which imposes a smoothing scale on the entropy surrogate. Overall, the experiment supports the unimodal message: minimizing the free-energy produces a unique Gibbs-type equilibrium at each $\tau$, and the equilibrium exhibits systematic low-temperature concentration near minimizers of $U$, thereby validating the mechanistic picture used to analyze the unimodal parametric energy in § 3.

### E.3. Multimodal Structural Gap under Controlled Misalignment

This experiment provides a controlled empirical validation of the *multimodal* mechanism developed in § 4, grounded in the well-documented *modality gap* phenomenon observed in CLIP-style training (Liang et al., 2022). Our theory explains this gap as an objective-induced geometric necessity: although symmetric multimodal InfoNCE admits a stable large-batch

deterministic limit, its intrinsic landscape is inherently *cross-coupled*, with equilibrium determined by the interplay of two directional alignment fields. When the two modalities exhibit heterogeneous conditional laws—arising from noise, distinct semantic density, or mismatched semantic structure—these fields generically fail to share a common potential, yielding a persistent negative symmetric divergence. We test this mechanism in a controlled setting by showing that exact marginal matching is a *knife-edge* condition: even mild, structured misalignment introduces a structural wedge that forces the population-level marginals apart.

**Latent variables and paired observations.** We generate paired observations $(\mathbf{x}_1, \mathbf{x}_2)$ representing two modalities that stem from a shared latent angle $\alpha \in (-\pi, \pi]$. To strictly rule out the trivial solution where marginals are uniform (and thus trivially match regardless of noise), we sample the shared latent $\alpha$ from a *non-uniform*, multi-peaked mixture of von Mises–Fisher distributions:

$$\alpha \sim w\,\mathrm{VM}(\mu_1, \kappa) + (1 - w)\,\mathrm{VM}(\mu_2, \kappa), \qquad \text{with } w = 0.7,\ \mu_1 = 0,\ \mu_2 = \pi,\ \kappa = 6.$$

Both modalities observe this angle as a projection onto the 2D unit circle, corrupted by independent Gaussian observation noise $\boldsymbol{\xi}_1, \boldsymbol{\xi}_2 \sim \mathcal{N}(\mathbf{0}, \sigma_{\mathrm{obs}}^2 \mathbf{I})$, where $\sigma_{\mathrm{obs}} = 0.02$. Modality 1 is generated by mapping the shared latent directly to the observation space:

$$\mathbf{x}_1 := [\cos \alpha,\ \sin \alpha]^\top + \boldsymbol{\xi}_1.$$

Modality 2 observes a *misaligned* version of the latent. We introduce an additive angular shift $\eta \sim \mathcal{N}(0, \sigma_{\mathrm{mis}}^2)$ before mapping it to the observation space:

$$\mathbf{x}_2 := [\cos(\alpha + \eta),\ \sin(\alpha + \eta)]^\top + \boldsymbol{\xi}_2.$$

Thus, increasing the variance of the misalignment $\sigma_{\mathrm{mis}}$ increases the *cross-modal conditional mismatch* between $\mathbf{x}_1$ and $\mathbf{x}_2$, while keeping the marginal complexity of both modalities comparable.

**Encoders, learning objective, and protocol.** To isolate the directional alignment dynamics and prevent magnitude from confounding the representation geometry, we apply bias-free linear encoders $f, g : \mathbb{R}^2 \to \mathbb{R}^2$ to the observed modalities. The unnormalized embeddings $\tilde{\mathbf{z}}_1 = f(\mathbf{x}_1)$ and $\tilde{\mathbf{z}}_2 = g(\mathbf{x}_2)$ are then projected onto the unit circle:

$$\mathbf{z}_1 = \frac{\tilde{\mathbf{z}}_1}{\|\tilde{\mathbf{z}}_1\|}, \quad \mathbf{z}_2 = \frac{\tilde{\mathbf{z}}_2}{\|\tilde{\mathbf{z}}_2\|} \in \mathbb{S}^1.$$

We adopt a cosine similarity critic $s(\mathbf{z}, \mathbf{w}) = \langle \mathbf{z}, \mathbf{w} \rangle$ and an corresponding exponential kernel $\kappa_\tau(\mathbf{z}, \mathbf{w}) = \exp\big(s(\mathbf{z}, \mathbf{w})/\tau\big)$ with a temperature of $\tau = 0.07$. This kernel directly instantiates the InfoNCE objective central to our multimodal analysis. Specifically, for a batch of $N$ paired observations, training minimizes the symmetric CLIP loss:

$$\mathcal{L}_{\mathrm{CLIP}}(f, g) = \tfrac{1}{2}\big(\mathcal{L}_{\mathrm{CE}}(\mathbf{S}/\tau, \mathbf{y}_{\mathrm{diag}}) + \mathcal{L}_{\mathrm{CE}}(\mathbf{S}^\top/\tau, \mathbf{y}_{\mathrm{diag}})\big),$$

where $\mathbf{S} \in \mathbb{R}^{N \times N}$ is the pairwise similarity matrix with entries $\mathbf{S}_{ij} := \langle \mathbf{z}_{1,i}, \mathbf{z}_{2,j} \rangle$, $\mathcal{L}_{\mathrm{CE}}$ denotes the standard cross-entropy loss, and the target $\mathbf{y}_{\mathrm{diag}}$ indicates that the matched positive pairs lie strictly on the diagonal ($i = j$).

Using Adam with learning rate $5 \times 10^{-3}$ for 2000 steps and batch size as 256. We sweep eight misalignment scales $\sigma_{\mathrm{mis}} \in \{0, 0.1, 0.2, 0.3, 0.4, 0.5, 0.6, 0.7\}$; larger scales induce progressively more severe cross-modal misalignment, yet all settings retain a meaningful statistical dependence between positive pairs. For each $\sigma_{\mathrm{mis}}$, we repeat training over 20 random seeds (seeds 0–19). The large batch size makes the training dynamics closer to the large-batch deterministic regime in Thm. 4.1, whose proof follows the same Monte-Carlo concentration logic as the asymmetric case but applied to the bidirectional symmetric normalization.

**Evaluation metrics.** After training, we evaluate the learned representations on $n_{\mathrm{eval}} = 8000$ fresh paired samples. For the unit-normalized embeddings $\mathbf{z}_1, \mathbf{z}_2 \in \mathbb{S}^1$, we compute the embedding angles $a_1 = \mathrm{atan2}(\mathbf{z}_{1,y}, \mathbf{z}_{1,x})$ and $a_2 = \mathrm{atan2}(\mathbf{z}_{2,y}, \mathbf{z}_{2,x})$, where the subscripts $x$ and $y$ denote the respective coordinate axes. The following visual diagnostics are considered: *(i) Marginal gap.* We discretize $a_1$ and $a_2$ into 60 bins on $(-\pi, \pi]$ and estimate the two empirical angle marginals $\hat{\rho}_\theta$ and $\hat{\rho}_\phi$. As a scalar summary of marginal mismatch, we report the symmetric KL divergence $D_{\mathrm{KL}}^{\mathrm{Sym}}(\hat{\rho}_\theta, \hat{\rho}_\phi)$ averaged over 20 random seeds (seeds 0–19), with error bars or shadow areas showing $\pm$ standard error. *(ii) Polar marginals.* To visualize *where* the mismatch occurs on the circle, we plot the two estimated angle marginals as polar density curves for each

misalignment scale $\sigma_{\mathrm{mis}}$. *(iii) Joint-angle heatmaps.* To visualize the learned *cross-modal coupling* beyond marginals, we plot a log-scaled 2D histogram, i.e., a $\log(1 + \mathrm{count})$ heatmap, of the joint distribution of $(a_1, a_2)$. Diagonal concentration indicates near-deterministic alignment $a_2 \approx a_1$; off-diagonal spread indicates systematic cross-modal mismatch. *(iv) Angle-shift density.* Finally, we compute the wrapped angular difference $\Delta a := \mathrm{wrap}(a_2 - a_1) \in (-\pi, \pi]$ and plot its empirical density. This directly measures the distribution of residual misalignment in embedding space.

For the scalar marginal-gap curve, we average over 20 random seeds and report the mean with a $\pm$SEM band. For the qualitative diagnostics (polar marginals, joint-angle heatmaps, and angle-shift densities), we display one fixed representative run (seed 0) and reuse the same held-out latent sample across all misalignment scales, matching the animation setup.

**Results.** Fig. 2b shows that the marginal discrepancy $D_{\mathrm{KL}}^{\mathrm{Sym}}(\hat{q}_\theta, \hat{q}_\phi)$ increases as the latent misalignment scale $\sigma_{\mathrm{mis}}$ grows. Since the encoder architecture, critic function, and optimization protocol remain fixed across all runs, this result isolates the conditional mismatch $\sigma_{\mathrm{mis}}$ as the primary driver of the gap. It confirms that as the two modalities become less conditionally compatible, the symmetric InfoNCE objective is forced to sacrifice marginal alignment, inducing a persistent modality separation that cannot be resolved by optimization alone.

The polar marginals in Fig. 8 provide a structural view of *how* this separation manifests geometrically. First, across all settings ($\sigma_{\mathrm{mis}} \geq 0$), the learned angular marginals are clearly *multi-peaked* rather than uniform. The density concentrates in specific angular regions, reflecting the underlying non-uniform latent law (a two-component von Mises–Fisher mixture) and confirming that the encoders successfully recover the latent structure up to a global rotation. Second, the evolution of these peaks reveals the gap mechanism. At $\sigma_{\mathrm{mis}} = 0$, the polar curves for the two modalities nearly overlap, consistent with a compatible pairing law where symmetric InfoNCE can simultaneously achieve alignment and marginal coincidence. However, for any $\sigma_{\mathrm{mis}} > 0$, the curves begin to diverge within the same modal regions: the peaks shift relative to each other, and their probability masses differ. This mismatch becomes visually more pronounced as $\sigma_{\mathrm{mis}}$ increases, tracking the quantitative rise in the gap curve.

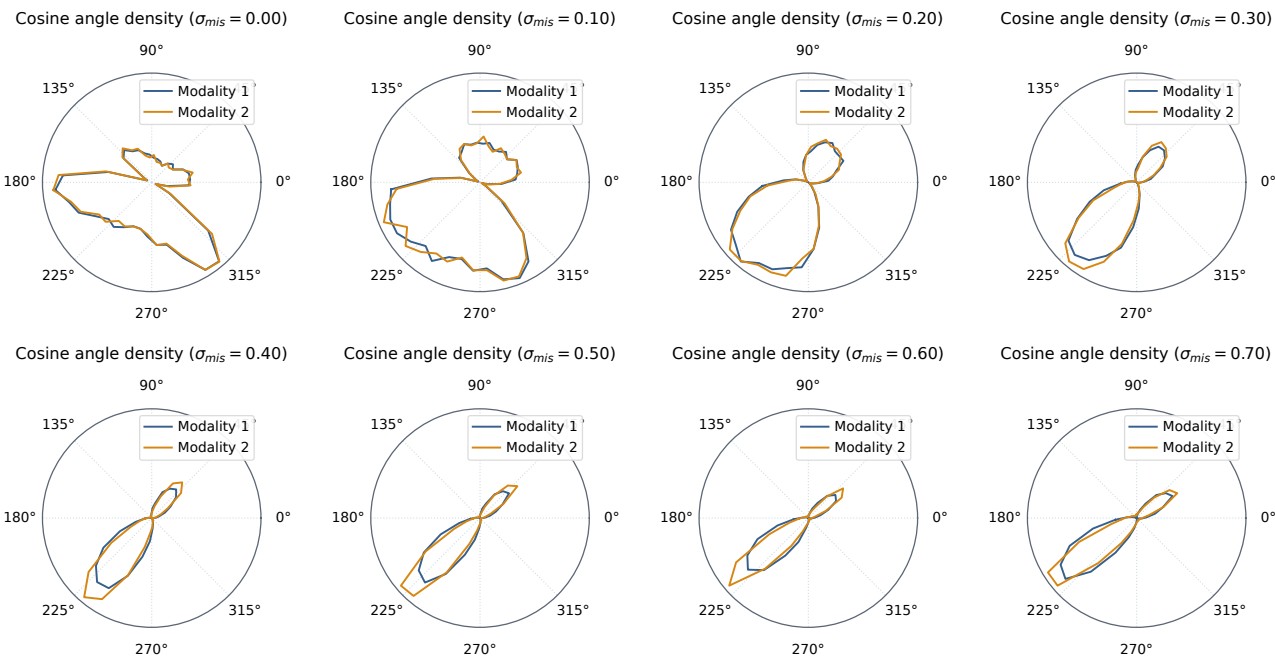

*Figure 8.* Polar marginals across misalignment. Estimated angle marginals of the two modalities on $S^1$. The mismatch becomes visually apparent for $\sigma_{\mathrm{mis}} > 0$ and grows with $\sigma_{\mathrm{mis}}$.

The joint-angle heatmaps in Fig. 9 elucidate the mechanism behind this separation. At $\sigma_{\mathrm{mis}} = 0$, probability mass concentrates sharply along the diagonal $a_2 \approx a_1$, indicating a near-deterministic coupling. Because the latent distribution is bimodal, this diagonal concentration is not uniform but forms distinct bright spots (high-density clusters) corresponding to the latent modes. As $\sigma_{\mathrm{mis}}$ increases, these clusters persist—indicating the model still captures the coarse latent semantics—

but they deform into thickened, elongated ellipses. The diagonal band broadens significantly, reflecting that the conditional law of $a_2$ given $a_1$ has become intrinsically noisy. Crucially, this is not merely a symmetric blur around the diagonal; the deformation is accompanied by the shifts in marginal occupancy observed in Fig. 8. This confirms that under the symmetric objective, broadening conditional alignment structurally necessitates a shift in the marginal supports.

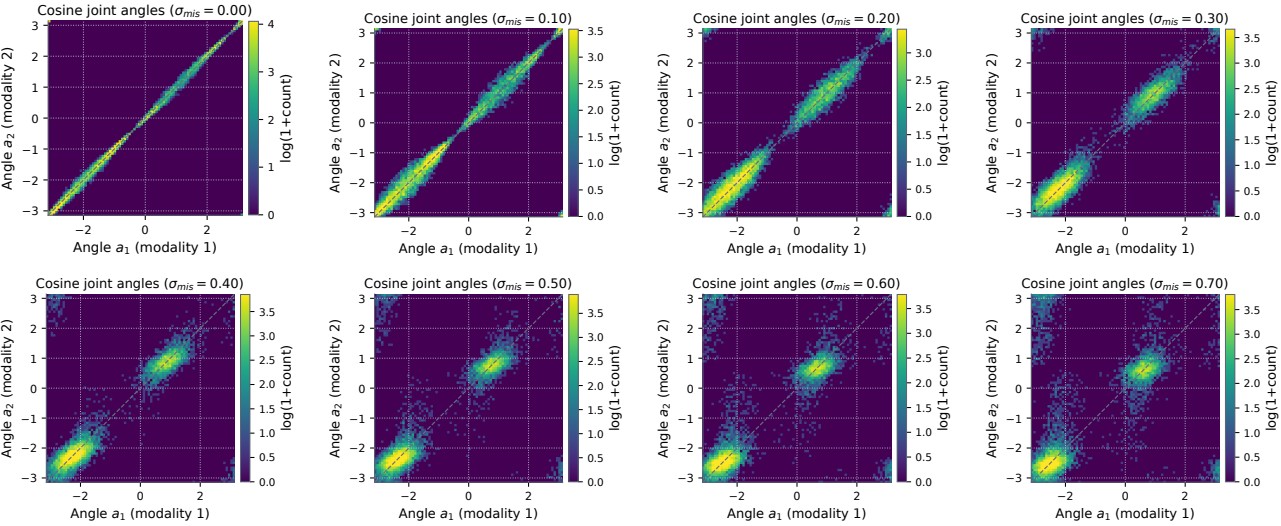

*Figure 9.* Joint-angle coupling across misalignment. 2D histograms of $(a_1, a_2)$ from $\sigma_{\mathrm{mis}} = 0.0$ to $\sigma_{\mathrm{mis}} = 0.7$. Diagonal concentration (small $\sigma_{\mathrm{mis}}$) indicates near-deterministic alignment; increasing off-diagonal spread (large $\sigma_{\mathrm{mis}}$) indicates intrinsically noisy coupling.

Finally, the angle-shift densities in Fig. 10 quantify the residual embedding misalignment $\Delta a = \mathrm{wrap}(a_2 - a_1)$. As $\sigma_{\mathrm{mis}}$ increases, the distribution of $\Delta a$ transitions from a sharp peak at zero to a progressively wider distribution with heavy tails. The residual shift becomes both larger in magnitude and more variable, matching the signature expected from the injection of additive latent noise $\eta \sim \mathcal{N}(0, \sigma_{\mathrm{mis}}^2)$.

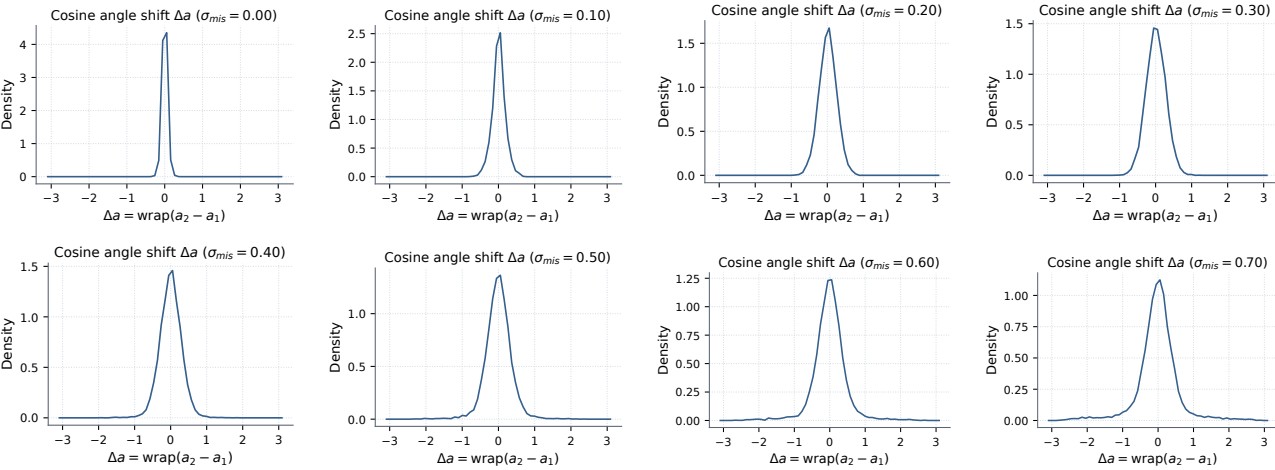

*Figure 10.* Residual embedding-space misalignment. Densities of $\Delta a = \mathrm{wrap}(a_2 - a_1)$ from $\sigma_{\mathrm{mis}} = 0.0$ to $\sigma_{\mathrm{mis}} = 0.7$. The distribution broadens as $\sigma_{\mathrm{mis}}$ increases, indicating larger and more variable residual misalignment in representation space.

In summary, these empirical patterns provide direct validation for the multimodal mechanism characterized in § 4. Our theoretical analysis posits that the symmetric objective consists of two directional alignment fields coupled via their marginals. In this experiment, the controlled perturbation $\alpha + \eta$ creates heterogeneous conditional laws, meaning the optimal alignment from Modality 1 to Modality 2 is no longer the inverse of the alignment from Modality 2 to Modality 1. Consequently, the two directions of the symmetric objective cannot be governed by a single, unified potential function.

*Table 2.* Retrieval performance and cross-modal discrepancy for pretrained OpenCLIP checkpoints evaluated on the first 5,000 images of MS-COCO `val2017`. Retrieval is reported as image-to-text and text-to-image Recall@1. Cross-modal discrepancy is quantified by the energy distance, RBF-MMD$^2$ at the median bandwidth, the centroid gap $\|\mu_I - \mu_T\|$, and centroid cosine similarity. Lower energy distance, MMD$^2$, and centroid gap indicate smaller cross-modal discrepancy.

| Model | I→T R@1 | T→I R@1 | $\mathcal{E}(p_I, p_T)$ | MMD$^2_{\sigma=\text{med}}$ | $\|\mu_I - \mu_T\|$ | $\cos(\mu_I, \mu_T)$ |
|---|---|---|---|---|---|---|
| PE-Core-L-14 | 0.741 | 0.559 | 0.599 | 0.255 | 0.854 | 0.097 |
| PE-Core-B-16 | 0.703 | 0.499 | 0.545 | 0.237 | 0.808 | 0.189 |
| ViT-bigG-14 | 0.670 | 0.506 | 0.288 | 0.129 | 0.598 | 0.384 |
| ViT-H-14 | 0.662 | 0.485 | 0.375 | 0.159 | 0.697 | 0.048 |
| ViT-L-14 | 0.569 | 0.357 | 0.697 | 0.303 | 0.898 | 0.184 |
| ResNet-50x16 | 0.559 | 0.354 | 0.502 | 0.230 | 0.761 | 0.349 |
| RN101 | 0.498 | 0.299 | 0.443 | 0.228 | 0.670 | 0.604 |
| ViT-B-16 | 0.492 | 0.302 | 0.688 | 0.313 | 0.871 | 0.309 |
| ResNet-50 | 0.480 | 0.289 | 0.627 | 0.276 | 0.849 | 0.237 |

Empirically, this manifests as a geometric tug-of-war between the modalities, yielding two distinct observable effects: (i) The broadening diagonal band in Fig. 9 reflects the necessary rise in conditional entropy driven by the mismatched conditionals. (ii) The widening marginal separation in Fig. 2b and Fig. 8 confirms our prediction that marginal coincidence is a fragile, knife-edge property that rapidly destabilizes once misalignment is introduced. Together, these observations demonstrate that the population-level modality gap is neither a finite-sample artifact nor a symptom of under-training. Rather, it is a fundamental consequence of the objective's geometry: when strict pairwise compatibility is violated (here, for any $\sigma_{\text{mis}} > 0$), the symmetric InfoNCE equilibrium inherently cannot maintain identical marginals.

### E.4. MS-COCO Experiments: Pretrained Gap and Controlled Pair Corruption

We complement the synthetic experiments with two studies on MS-COCO. The first is an observational study of pretrained CLIP-like models, designed to assess whether these models exhibit a modality gap and whether improved retrieval performance necessarily corresponds to a smaller cross-modal distributional discrepancy. The second is a controlled intervention study designed to test whether weakening image-text compatibility during training systematically enlarges the modality gap. Together, these studies relate the intrinsic geometric analysis to realistic image-text representations.

E.4.1. PROBING PRETRAINED MODELS ON MS-COCO

**Setup and evaluations.** We evaluate a collection of pretrained CLIP (using the OpenCLIP (Ilharco et al., 2021) official artifacts) checkpoints on MS-COCO `val2017`. For each checkpoint, we use the first 5000 validation images and up to 5 captions per image, yielding one normalized image embedding ($\mathbf{z}_I$) per image and one normalized text embedding ($\mathbf{z}_T$) per caption. Retrieval is evaluated in both directions under the many-captions-to-one-image matching protocol. Image-to-text Recall@$K$ assesses whether any of the top-$K$ retrieved captions is associated with the query image, whereas text-to-image Recall@$K$ assesses whether the correct image appears among the top-$K$ retrieved images for a given caption. We report retrieval performance using Recall@1 (R@1), and define AvgR@1 as the mean of the R@1 scores in the two retrieval directions.

To quantify cross-modal discrepancy, we compute the centroid gap

$$\|\mu_I - \mu_T\|, \qquad \text{with } \mu_I = \mathbb{E}[\mathbf{z}_I], \mu_T = \mathbb{E}[\mathbf{z}_T],$$

and the energy distance

$$\mathcal{E}(p_I, p_T) = 2\mathbb{E}[d(\mathbf{z}_I, \mathbf{z}_T)] - \mathbb{E}[d(\mathbf{z}_I, \mathbf{z}'_I)] - \mathbb{E}[d(\mathbf{z}_T, \mathbf{z}'_T)],$$

where $p_I$ and $p_T$ denote the image and text embedding marginals, $\mathbf{z}_I, \mathbf{z}'_I \sim p_I$ and $\mathbf{z}_T, \mathbf{z}'_T \sim p_T$ are independent laws, and $d(\cdot, \cdot)$ is the Euclidean distance between normalized embeddings. Whereas the centroid gap captures only the separation between the modality means, the energy distance compares cross-modal distances with within-modal distances and thus provides a broader measure of distributional mismatch. Unlike retrieval, which evaluates alignment between matched image-caption pairs, the energy distance operates at the distribution level: it is small when image and text embeddings are well mixed and large when cross-modal distances remain systematically greater than within-modal distances. We additionally report an RBF-MMD diagnostic using a bandwidth selected by the median-distance heuristic.

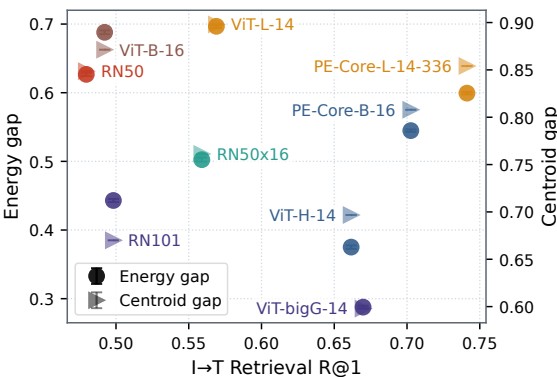 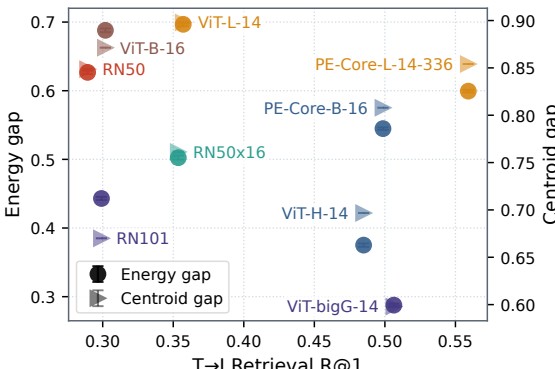

*Figure 11.* Directional retrieval versus cross-modal gap on MS-COCO for pretrained OpenCLIP checkpoints. Left: image-to-text retrieval (I→T R@1) versus gap. Right: text-to-image retrieval (T→I R@1) versus gap. In both panels, the left y-axis shows energy distance, and the right y-axis shows centroid gap. The relation is clearly non-monotone: models with similar directional retrieval can exhibit substantially different cross-modal discrepancy. This confirms that retrieval performance and modality gap are related but not equivalent, and that strong pretrained models may still maintain a substantial structural separation between image and text marginals.

**Results.** As shown in Tab. 2, the results indicate that retrieval and cross-modal discrepancy are only partially aligned. As already seen in the AvgR@1 plot in Fig. 6 (left), better retrieval often corresponds to a smaller gap, but the trend is clearly non-monotone. For instance, PE-Core-L-14 achieves the best retrieval yet still has a large energy distance and centroid gap, whereas ViT-bigG-14 attains slightly lower retrieval but the smallest gap values in the table. The separate $I \to T$ and $T \to I$ versus gap plots (Fig. 11) show that this is not caused by averaging: models with nearly identical directional retrieval can still differ substantially in energy and centroid discrepancy, e.g., ViT-bigG-14 versus PE-Core-B-16 in $T \to I$ retrieval, or ViT-L-14 versus ResNet-50x16 in $I \to T$ retrieval. The RBF-MMD values follow the same broad ordering, confirming that the effect is not metric-specific. Overall, these results show that the modality gap persists even in strong pretrained CLIP variants and is not reducible to retrieval performance alone; instead, it reflects an additional geometric property of the learned image and text marginals.

### E.4.2. CONTROLLED SAME-CATEGORY PAIR CORRUPTION ON MS-COCO

**Setup and evaluations.** To test more directly whether weakening image–text compatibility enlarges the modality gap, we perform a controlled corruption experiment on MS-COCO `train2017`. We first build an index that records, for each image, its caption set, its COCO category labels, and the pool of other images sharing each category. During training, the image is always kept fixed. With probability $1 - p$, we pair it with one of its own captions; with probability $p$, we instead replace the caption by a randomly chosen caption from a *different* image that shares at least one category with the current image. Thus, the corruption preserves coarse semantic relatedness while breaking exact instance-level correspondence. We sweep the corruption probability over $p \in \{0, 0.25, 0.50, 0.75, 1.00\}$.

We fine-tune pretrained OpenCLIP backbones (ResNet-50 and ViT-B-16) using the standard symmetric CLIP loss. To isolate the effect of pair corruption rather than full-model adaptation, we freeze the backbone towers and update only the projection heads together with the logit scale. Training uses AdamW with learning rate $10^{-5}$, weight decay 0.05, batch size 256, and 5000 optimization steps. Evaluation follows the same protocol as in the previous experiment: we encode the first 5000 images of MS-COCO `val2017` together with up to 5 captions per image, report image-to-text and text-to-image Recall@1, and define AvgR@1 as their mean. As the geometric diagnostic, we report the centroid gap $\|\mu_I - \mu_T\|$.

**Results.** As shown in Fig. 6 (right), the effect of corruption is clean for both backbones. As the caption corruption probability $p$ increases, retrieval performance degrades while the centroid gap grows. For ResNet-50, AvgR@1 drops from 0.398 at $p = 0$ to 0.226 at $p = 1$, while the centroid gap increases from 0.845 to 1.081. For ViT-B-16, AvgR@1 drops from 0.484 to 0.297, while the centroid gap rises from 0.851 to 1.148. The same pattern holds in both retrieval directions, as shown in Fig. 12: for ResNet-50, I→T R@1 decreases from 0.489 to 0.272 and T→I R@1 from 0.307 to 0.181; for ViT-B-16, I→T R@1 decreases from 0.564 to 0.366 and T→I R@1 from 0.405 to 0.229.

Several aspects of this trend are important. First, the corruption is not arbitrary: the replacement caption is sampled from a same-category image, so the supervisory signal remains semantically plausible at a coarse level. Nevertheless, this weaker

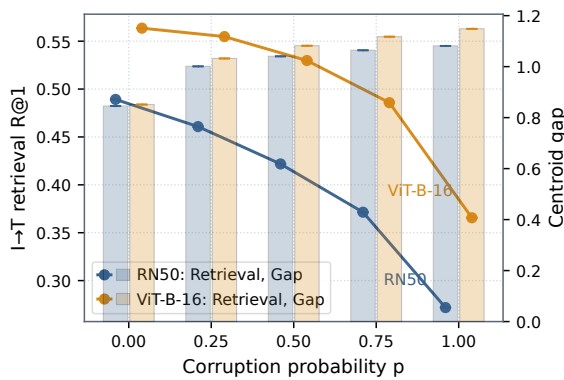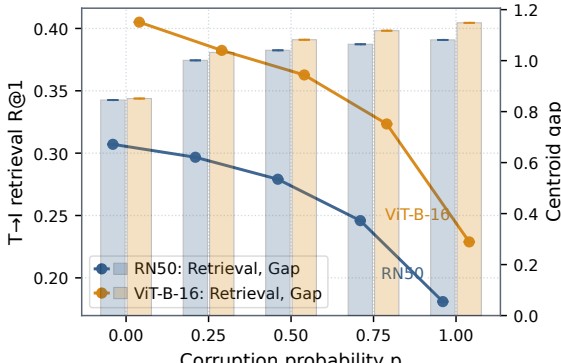

*Figure 12.* Directional retrieval under same-category caption corruption on MS-COCO. Left: image-to-text retrieval (I→T R@1) versus corruption probability $p$. Right: text-to-image retrieval (T→I R@1) versus $p$. Bars show the centroid gap on the right axis. For both RN50 and ViT-B-16, increasing same-category mispairing degrades retrieval in both directions while systematically enlarging the modality gap.

form of mispairing is already sufficient to produce a substantial geometric separation between image and text marginals. Second, the centroid gap reacts very early: even moving from $p = 0$ to $p = 0.25$ causes a sharp increase in gap for both ResNet-50 and ViT-B-16, while the retrieval drop is initially much milder. This suggests that cross-modal distributional separation is highly sensitive to degradation in pairwise compatibility, and may emerge before retrieval collapses. Third, the monotone behavior is consistent across both architectures, indicating that the phenomenon is not tied to one particular backbone family.

Overall, this experiment provides a controlled real-data analogue of the theory developed in the main text. Exact image–text compatibility is progressively weakened, and the learned representations respond in two coupled ways: matched-pair retrieval worsens, and the image/text marginals drift farther apart. This is precisely the behavior predicted by the structural-gap view of symmetric contrastive learning: once pairwise compatibility is degraded, optimization can no longer maintain the same degree of cross-modal coincidence, and the modality gap widens systematically.

## F. Discussion

### F.1. Limitations and Future Directions

We analyze symmetric InfoNCE in a tractable large-batch, low-temperature geometric regime. These assumptions isolate the core mechanism, but they also mark the present boundary of the theory and point to several natural extensions.

**Finite-temperature and finite-batch effects.** Our main results rely on the limits $(\tau \to 0^+)$ and $(N \to \infty)$. Practical training instead uses finite batches and moderate temperatures, introducing stochasticity and entropic smoothing that may soften the predicted modality barriers or permit transitions between metastable states. Characterizing finite-temperature and finite-batch corrections, especially the interaction between batch noise and barrier crossing, is a natural next step toward sharper practical predictions.

**Singularities and boundary behavior.** The multimodal analysis imposes a strict density floor to regularize the logarithmic barrier near the boundary of the probability simplex. Removing this technical constraint would require analyzing the singular geometry of the KL divergence on the open simplex, and is likely essential for understanding singular measures, including distributions that collapse onto lower-dimensional manifolds or discrete sets.

**Geometric homogeneity.** We assume a fixed compact embedding manifold with constant kernel volume. Extending the framework to heterogeneous geometries, where the local kernel volume varies across the space, would connect the present measure-theoretic analysis to unnormalized contrastive objectives, hyperbolic contrastive learning, and related non-contrastive predictive frameworks.

**Objective geometry versus optimization dynamics.** Our theory describes the geometry of the objective rather than the trajectory of a specific optimization algorithm. Understanding how stochastic optimization explores the resulting barrier-shaped landscape, and whether it can reach the exceptional solutions associated with exact marginal matching, remains an important open problem at the interface of geometric analysis and learning theory.

## F.2. Broader Implications

Our analysis suggests that contrastive learning should be understood not only through pointwise alignment, but through the population geometry induced on the representation space. In the large-batch regime, InfoNCE gives rise to deterministic fields and intrinsic energies over $\mathcal{Z}$, clarifying why strong retrieval can coexist with persistent modality-level discrepancies: matching positive pairs does not by itself control the induced marginal geometry. In particular, the multimodal theory shows that improving pairwise alignment alone need not eliminate the modality gap unless the population-level coupling is also reshaped. This points to a practical lesson for both evaluation and objective design: beyond retrieval, one should monitor distributional diagnostics of the learned marginals, and interventions aimed at closing the gap should act on the induced geometry itself, for example, through explicit discrepancy regularization, shared reference fields, or other mechanisms that weaken the barrier-like coupling between modalities.

More broadly, this perspective suggests that intrinsic population-level analyses can be useful for isolating principled properties of representation-learning objectives that are not obvious from parameter-space dynamics alone. In this view, objective geometry, optimization dynamics, parametrization-specific inductive biases, and identifiability guarantees address distinct but complementary questions: the intrinsic geometry clarifies what the loss tends to favor, optimization theory explains how algorithms explore that landscape, and identifiability or generative analyses address when the resulting representations correspond to meaningful latent structure. Our results, therefore, point to a broader methodological lesson: understanding modern representation learning may require combining intrinsic objective-level analysis with complementary theories of optimization, parametrization, and recoverability.

