# OpenReview forum: "The Geometric Mechanics of Contrastive Representation Learning: Alignment Potentials, Entropic Dispersion, and Cross-Modal Divergence"
_ICML.cc/2026/Conference — ICML 2026 regular_

### Official Review · Reviewer_reUT · 2026-02-13

**Soundness:** 3
**Presentation:** 4
**Significance:** 3
**Originality:** 3
**Overall Recommendation:** 4
**Confidence:** 3

**Summary:**

While methods like InfoNCE have achieved significant empirical success by aligning semantically similar data points, the underlying geometric mechanisms governing the evolution of representation distributions have remained undercharacterized. To address this, the authors introduce a rigorous measure-theoretic framework that models the learning process as the evolution of probability measures on the embedding manifold. The core contribution of the work is the identification of a fundamental geometric bifurcation between unimodal and multimodal learning regimes. This finding provides a structural explanation for the modality gap, proving it is a geometric necessity of the objective rather than an artifact of initialization.

**Compliance With Llm Reviewing Policy:**

Affirmed.

**Final Justification:**

I maintain my original assessment of this work.

**Key Questions For Authors:**

1. Your analysis relies heavily on the large-batch limit to establish the deterministic energy landscape. In practical settings with finite batches, the gradient is stochastic. Does the barrier remain robust under significant gradient noise?

2. Your multimodal experiments rely on simple von Mises-Fisher mixtures on spheres, which have constant curvature. Real-world data often lies on manifolds with complex, hierarchical, or non-uniform curvature. How does the geometric necessity of the modality gap hold up when the underlying manifold violates the assumption?

**Limitations:**

Since the paper relies entirely on synthetic data for validation (Appendix D), it would be beneficial to explicitly state the lack of evaluation on real-world, large-scale datasets (e.g., ImageNet, LAION) as a limitation. Bridging the gap between the idealized theoretical dynamics and the messy, high-dimensional statistics of real-world multimodal data remains an important open challenge that could be highlighted.

**Strengths And Weaknesses:**

1. The theoretical analysis appears rigorous, grounded in well-defined measure-theoretic concepts and large-batch limit analysis. The structure of the argument moves from the general framework to the specific cases of unimodal convexity and multimodal coupling.

2. The authors mathematically prove that the modality gap is a structural geometric necessity caused by a negative symmetric divergence term in the objective. This insight redefines the gap as an inherent feature of symmetric InfoNCE, fundamentally shifting the design of future multimodal loss functions.

3. The core results hinge on large-batch ($N \to \infty$) and intrinsic ($\tau \to 0^+$) limits, standard yet potentially divergent from small-batch or high-temperature behaviors. While relevant, discussing finite-sample dynamics or convergence rates would strengthen the claims' practical applicability.

---

> ### Author Rebuttal · Authors · 2026-03-26
>
> We sincerely thank the reviewer for the careful and constructive assessment, and especially for recognizing the rigor of the measure-theoretic analysis and the value of the geometric bifurcation view. We also appreciate the two main suggestions: clarifying the role of finite-batch stochasticity, and discussing how the argument should be interpreted beyond the homogeneous-manifold setting.
>
> ---
>
> **1. Finite-batch stochasticity \& barrier robustness.**
>
> This is an important question. Our current answer is: not as an unchanged theorem under arbitrary gradient noise, but as a leading-order geometric mechanism whose effect should become clearer as batch size grows. Accordingly, the paper’s sharpest claims concern the objective geometry in the large-batch, intrinsic regime, rather than exact barrier preservation under all finite-batch stochastic dynamics. As trend-level evidence, the original manuscript already includes a numerical large-batch validation showing that finite-batch gradients align increasingly well with the deterministic reference as the number of negatives grows, with gradient error decreasing monotonically.
>
> More broadly, we agree that understanding how this mechanism scales with batch size (e.g., whether one can characterize a robustness threshold, convergence rate, or a law governing noise-induced barrier softening) is both significant and nontrivial. We view this as an important open direction for extending the current theory from intrinsic objective geometry to practical learning dynamics.
>
> ---
>
> **2. Non-uniform curvature / heterogeneous geometry.**
>
> This is an important scope question. Our synthetic multimodal experiment is meant to validate the mechanism in the regime analyzed by the theory, not to claim that all real multimodal representation spaces are globally spherical. More precisely, in our setting, the representation manifold is determined by the similarity critic and normalization regime, not by the data source. Under standard feature-normalized InfoNCE with a cosine critic (as in CLIP), the embeddings lie on a sphere, which is exactly the ambient manifold targeted by our analysis. In this sense, the vMF-on-sphere experiment is not an arbitrary choice, but a clean validation of the normalized contrastive regime studied in the paper.
>
> So our answer is the following:
> - Within the normalized InfoNCE + isotropic critic regime, the geometric mechanism we analyze is directly aligned with the induced embedding geometry, so the modality-gap result should be interpreted as a structural property of that regime in the analyzed limit.
> - If the underlying geometry genuinely falls outside this scope—for example, through non-normalized embeddings, strongly varying local volume, or objectives that induce genuinely non-spherical geometry (e.g., hyperbolic constructions)—then our current theorem does not claim the same necessity result unchanged. In that case, additional geometric control is required, and this is part of the intended limitation rather than something we claim to have already solved.
>
> We will clarify this scope more explicitly in the final version.
>
> ---
>
> **3. Grounding beyond synthetic validation.**
>
> We appreciate this suggestion and have acted on it. In the final version, we add two real-data studies on MS-COCO.
>
> - Pretrained-model observational study. Across a range of pretrained OpenCLIP checkpoints, retrieval and cross-modal discrepancy are related but not equivalent: for example, PE-Core-L-14-336 has the best Avg Recall@1 (0.650) yet still shows a relatively large energy distance / centroid gap (0.599 / 0.854), whereas ViT-bigG-14 has slightly lower Avg Recall@1 (0.588) but much smaller gap values (0.288 / 0.598). This supports the view that the modality gap is an additional geometric property, not reducible to retrieval alone.
>
> - Controlled interventional study. We also run a same-category caption-corruption experiment on MS-COCO, which weakens exact image–text compatibility while preserving coarse semantic plausibility. The effect is clean: as the corruption probability $p$ increases, retrieval degrades while the centroid gap grows monotonically; moreover, the gap reacts early even when the retrieval drop is still relatively mild. For ViT-B-16, moving from $p=0$ to $0.25$ changes Avg Recall@1 only from 0.484 to 0.469, but the centroid gap jumps from 0.851 to 1.032. This is exactly the type of signature suggested by the theory.
>
> ---
>
> We will therefore revise the paper in two ways: (i) make the scope clearer by explicitly stating that the sharpest claims concern the large-batch regime, with finite-batch stochastic dynamics and complex geometry as important extensions; and (ii) incorporate the new MS-COCO evidence to connect the theoretical mechanism to realistic multimodal representations.
>
> We are grateful for these suggestions, which we believe meaningfully improve both the positioning and the practical relevance of the paper.

---

> > ### Author Rebuttal · Reviewer_reUT · 2026-04-03
> >
> > The paper's strongest theoretical results are confined to the large-batch, normalized-InfoNCE-with-isotropic-critic regime, and the authors have not yet provided a quantitative bridge to finite-batch practice. While we appreciate the new MS-COCO evidence, it is limited to one architecture and one form of synthetic misalignment, leaving open the question of whether the geometric bifurcation manifests robustly across diverse architectures, similarity functions, and real-world noise distributions.

---

> > > ### Author Response · Authors · 2026-04-03
> > >
> > > Thank you for the thoughtful follow-up. We appreciate your remaining concern about the scope and practical reach of the central claim. We address the two points you raise as directly as possible below.
> > >
> > > ---
> > >
> > > > "The paper’s strongest theoretical results are confined to the large-batch, normalized-InfoNCE-with-isotropic-critic regime, and the authors have not yet provided a quantitative bridge to finite-batch practice."
> > >
> > > We agree that a full **finite-batch quantitative theory** falls outside the scope of the present analysis. The reason is that our main theorems characterize the **intrinsic objective geometry** in the large-batch regime. In contrast, a finite-batch theory would additionally need to model stochastic optimization effects—for example, how gradient noise scales with batch size, whether one can prove robustness thresholds or convergence laws, and how noise-induced barrier softening interacts with the intrinsic landscape. We view this as an important but genuinely nontrivial extension, and it goes beyond what can reasonably be established within a single paper focused on the geometric mechanism itself.
> > >
> > > At the same time, our numerical validation suggests that the large-batch theory is not disconnected from practical training. In the original manuscript, the gradient-consistency experiment shows that as the batch size grows, the finite-batch gradient aligns increasingly well with the deterministic reference; by batch size **1024**, the agreement is already quite strong. So while we do not claim a full finite-batch theorem, the current evidence supports interpreting the large-batch analysis as a meaningful leading-order description for modern contrastive training regimes that typically use large batches.
> > >
> > > ---
> > >
> > > > "While we appreciate the new MS-COCO evidence, it is limited to one architecture and one form of synthetic misalignment, leaving open the question of whether the geometric bifurcation manifests robustly across diverse architectures, similarity functions, and real-world noise distributions."
> > >
> > > We agree that the revised evidence does not establish universality across **all** architectures, similarity functions, or real-world noise models. However, we would like to clarify two points gently.
> > >
> > > First, the new evidence is **not limited to one architecture**. The pretrained-model study spans **9 OpenCLIP checkpoints** across both **CNN and Transformer** families, pretrained on large-scale datasets. Our goal there is not to claim universality, but to show that the geometric signatures highlighted by the theory are visible across multiple strong CLIP-style models rather than in a single architecture. (Because of the response character limit, we refer the reviewer to our response to Reviewer UNd5 for the detailed numerical results.)
> > >
> > > Second, the controlled corruption is not intended as an arbitrary synthetic perturbation. It was designed to mimic a realistic form of **cross-modal translation noise** in human annotation: preserving the main categorical/salient concepts while perturbing some finer-grained semantic details. This is closely aligned with the paper’s prediction that weakening conditional compatibility need not immediately destroy coarse retrieval behavior, but can still enlarge cross-modal **distribution-level** separation. In that sense, the experiment is meant as a targeted probe of the mechanism, not as a claim to exhaust the full space of real-world noise processes.
> > >
> > > ---
> > >
> > > **In summary**, we agree that the strongest theorem-level claim should be read as an objective-level geometric result in the analyzed regime, not yet as a complete theory of finite-batch optimization across all architectures and similarity functions. Our numerical validation and new MS-COCO evidence are intended as supporting evidence that the corresponding signatures already appear in practical CLIP-style settings across both CNN and Transformer models.
> > >
> > > We appreciate this helpful clarification, and in the final version, we will sharpen the wording accordingly so that this objective-level claim and its relation to practical training behavior are stated more precisely.

---

### Official Review · Reviewer_kKAz · 2026-02-21

**Soundness:** 2
**Presentation:** 4
**Significance:** 3
**Originality:** 4
**Overall Recommendation:** 4
**Confidence:** 3

**Summary:**

This paper proposes a measure-theoretic framework to analyze the optimization landscapes of InfoNCE and symmetric (CLIP-style) contrastive objectives. By lifting the analysis from pointwise samples to representation measures on a compact Riemannian manifold $\mathcal{Z}$, the authors derive a deterministic large-batch limit for the objective. The paper identifies a "geometric bifurcation": unimodal InfoNCE is characterized as a strictly convex functional with a unique Gibbs equilibrium, while multimodal InfoNCE is shown to contain a negative symmetric divergence term. This term is credited with enforcing the "modality gap" as a structural necessity of the objective.
This research introduces a rigorous measure-theoretic framework designed to dissect the optimization landscapes inherent in both the Noise-Contrastive Estimation (InfoNCE) objective and its symmetric counterpart, commonly employed in models like CLIP. The core innovation lies in elevating the analysis from discrete, pointwise samples to continuous representation measures defined over a compact Riemannian manifold, denoted as $\mathcal{Z}$. This shift to a continuous domain allows the authors to derive a deterministic, large-batch limit for the contrastive objective, effectively bypassing the stochasticity associated with finite-batch sampling and providing a clear, macroscopic view of the optimization dynamics.

The paper's key finding is the identification of a fundamental geometric bifurcation that separates the behaviors of unimodal and multimodal representation learning.

**Compliance With Llm Reviewing Policy:**

Affirmed.

**Key Questions For Authors:**

if the modality gap truly pushes mass into disjoint sets , how can the negative symmetric divergence  remain a valid, finite part of the functional? Does the "structural necessity" of the gap vanish if you remove the density floor?

your consistency proof assumes negatives are independent. In a CLIP-style  in-batch setting, the negatives are functions of $\theta$. How does their simultaneous movement affect the "alignment potential" field?

regarding the bound in Theorem 3.2: as the representation concentrates into a ground state , $\underline{q}\_\theta$ vanishes. Can you provide the specific rates at which the KDE error  vanishes relative to that minimum density?

you assume a constant kernel volume. If a learned manifold has varying local volumes , could the modality gap just be an artifact of mass migrating to high-volume regions of the geometry?

you base your inheritance conditions (Corollaries 3.1 and 4.1) on sufficient encoder expressivity. However, neural networks utilize smooth activation functions like GELU or SiLU and are often subject to strict Lipschitz constraints via spectral normalization. How do these specific functional constraints interact with a landscape that inherently wants to drive mass toward disjoint, non-Gibbs 'extremal' configurations? Is it possible that the modality gap isn't a 'structural necessity' for the math to work, but rather an artifact of these continuous functions being unable to cross the repulsive logarithmic barriers you've identified during standard gradient descent?

**Limitations:**

Yes

**Strengths And Weaknesses:**

Strengths
Originality: Moving from the discrete alignment-uniformity decomposition to a continuous measure-theoretic potential field analysis is a sophisticated and welcome shift in perspective.
Significance: The mechanistic explanation for the "Mind the Gap" phenomenon, attributing it to the objective's own curvature rather than data noise or initialization, is a high-impact claim that challenges current empirical folklore.
Clarity: The paper is well-structured, with a clear roadmap (Figure 1) and rigorous definitions of the kernel-induced state variables ($\Gamma, \tilde{q}$).

Weaknesses and Major Flaws:

Mischaracterization of Gradient Consistency (Thm 3.1)

The proof of gradient consistency in Appendix C.2 relies on an i.i.d. sampling assumption for negatives. However, modern contrastive learning (SimCLR, CLIP) utilizes in-batch negatives. In this setting, the negative samples $\mathbf{w}\_j $ are themselves functions of the parameters $\theta$. The paper’s derivation neglects the $\nabla_\theta g_\phi(\mathbf{y}\_j)$ terms in the log-sum-exp gradient. This omission ignores the simultaneous co-adaptation of negatives, which is fundamental to the repulsion mechanics in practice.

Singularities in the Multimodal Landscape (Thm 4.3 & Prop 4.3)

The paper claims the modality gap is enforced by a negative symmetric divergence $-D_{\mathrm{S}}(q_\theta, q_\phi)$.
Technical Error: If $q_\theta$ and $q_\phi$ achieve a complete modality gap (disjoint support), the KL-divergence becomes infinite ($D_{\mathrm{KL}}(q_\theta \| q_\phi) = \infty$).
The functional $\mathcal{F}^{\mathrm{mm}}$ is thus undefined at the very equilibrium the authors seek to describe. The authors introduce a "density floor" $\underline{\rho}$ as a patch, but this contradicts the later claim of "extremal collapse." You cannot simultaneously claim the math requires a density floor for stability and then argue the objective drives mass to a boundary where that floor would be violated.

Vacuous Bounds in the Low-Temperature Limit (Thm 3.2)

The consistency bound between the parametric energy $\mathcal{J}\_\tau$ and the intrinsic functional $\mathcal{F}$ is given as $O(\varepsilon\_{\mathrm{kde}}(\tau) / \underline{q}\_\theta)$.

As $\tau \downarrow 0^+$, the paper correctly notes that the equilibrium concentrates into a ground state. This implies $\underline{q}\_\theta$ (the minimum density) vanishes.

Unless the authors can prove that the KDE error $\varepsilon_{\mathrm{kde}}(\tau)$ vanishes faster than the density $\underline{q}\_\theta$ at the manifold boundaries, this bound is vacuous. No such rate analysis is provided.

Homogeneity Assumptions (Asm 2.1)

The assumption of constant kernel volume $\int \kappa\_\tau d\mu = V\_\kappa(\tau)$ is rarely satisfied in the non-linear embedding spaces learned by deep neural networks. If the representation manifold has varying local volume, the "Uniformity" term is corrupted by local geometric artifacts. The paper fails to discuss how local curvature affects the entropic dispersion.

---

> ### Author Rebuttal · Authors · 2026-03-26
>
> We sincerely thank the reviewer for the exceptionally careful and technically engaged reading. Your questions helped us identify several places where the original version was too compressed. We clarify them below.
>
> ---
>
> **1. Gradient consistency: population large-batch regime vs. finite-batch in-batch training.**
>
> Short answer: our theorem is a large-batch population-limit statement, not an exact finite-batch CLIP gradient identity. In particular, Thm. 3.1 is unimodal, so cross-modal co-adaptation is not relevant there. The multimodal co-adaptation issue is handled at the level of the two directional population fields in Thm. 4.1 and App. C.8, which are then combined into the symmetric energy. Thus, the paper does not ignore simultaneous movement of negatives; it treats it at the population-field level rather than through exact finite-$N$ in-batch algebra. We agree, however, that the setup should state more explicitly that i.i.d. marginal negatives are the theorem regime, whereas exact finite-batch in-batch dependence is a separate approximation issue.
>
> ---
>
> **2. Density floor, singularities, and what Prop. 4.2 actually claims.**
>
> Short answer: We do not claim full disjoint support, and the density floor is introduced for well-posedness, not to create the gap. Prop. 4.2 concerns the *excess mass above the floor*.  Writing $\rho_1=\underline{\rho}+\eta$ with $\eta\ge 0$, the proposition states that the excess mass $M_{\mathrm{ex}}:=1-\underline{\rho}\mu(\mathcal{Z})$ is driven toward shrinking near-minimizer sets of the effective field. This is not "complete modality separation". The role of the floor is that, without it, the negative symmetric divergence can make the intrinsic energy ill-posed whenever one density vanishes where the other remains positive. Thus, the floor is not the source of the mechanism; it is what keeps the coordinate best-response problem finite. Removing the floor removes the present well-posed collapse proposition, but not the negative symmetric divergence mechanism itself.
>
> ---
>
> **3. "Vacuous Bounds in the Low-Temperature Limit".**
>
> Short answer: The apparent conflict comes from combining two statements that operate in different regimes. Thm. 3.2 is a pointwise-in-$\theta$ statement: it fixes $\theta$, assumes the induced density $q_\theta$ is continuous and bounded below by a positive constant, and proves low-temperature functional consistency for that fixed $q_\theta$. Here, $\underline{q}\_\theta$ is the lower bound of that fixed density. By contrast, Prop. 3.3 is an intrinsic Gibbs-equilibrium statement: for fixed $U$, the Gibbs minimizer concentrates on low-energy sublevel sets as $\tau\downarrow 0$. This concerns the intrinsic equilibrium family $\rho^\star_\tau$, not the fixed density $q_\theta$ in Thm. 3.2. Therefore, the statement "ground-state concentration implies $\underline{q}_\theta\to 0$, hence Thm. 3.2 becomes vacuous" would apply only to a stronger uniform-along-minimizers statement that we do not claim. We will clarify this distinction more explicitly in the final version.
>
> ---
>
> **4. Constant kernel volume and local volume effects.**
>
> Short answer: No, the multimodal mechanism is not a high-volume migration artifact. The intended regime is the standard feature-normalized setting used in modern contrastive learning: embeddings are normalized to a fixed compact space (e.g., the unit sphere), and similarity is computed by an isotropic critic such as a vMF-style kernel. In this regime, constant kernel volume is natural by symmetry. Even when $V_\kappa$ is not constant, this affects the KDE-control side of the derivation, not the existence of the multimodal negative symmetric divergence. Thus, we believe that the modality gap is not a "high-volume migration artifact". Relaxing Assm. 2.1 would mainly introduce position-dependent normalization terms, rather than changing the sign/coupling structure that drives the multimodal bifurcation.
>
> ---
>
> **5. Objective geometry vs. restricted parametric classes.**
>
> We appreciate this question, because it points to a distinction the original paper did not make explicit enough: our strongest structural claim is an objective-level statement, whereas the corollaries are inheritance statements that require sufficient encoder expressivity.
>
> So our answer is: yes, restricted Lipschitz neural classes can soften how much of the extremal intrinsic geometry is realized in practice; but no, this does not make the modality-gap mechanism merely an artifact of continuous functions being unable to "cross the barrier". The negative symmetric divergence term is already present in the intrinsic multimodal functional itself. What restricted parametric classes may change is the degree of inheritance of the intrinsic geometry, not the existence of the objective-level coupling mechanism.
>
> ---
>
> We are grateful for your careful reading, and we will add short interpretive remarks after the relevant formal statements to make these distinctions clearer.

---

> > ### Author Rebuttal · Reviewer_kKAz · 2026-04-01
> >
> > The rebuttal clarified several technical points and resolved my main concerns regarding internal consistency. However, my remaining concern relates to the interpretation of the central claim that the modality gap is a structural necessity of the objective. While the rebuttal clarifies that the results establish an objective-level geometric mechanism, questions remain about how this translates to practical training behavior and whether the results demonstrate inevitability versus geometric tendency.
> >
> > These concerns relate to the scope and interpretation of the main contribution rather than correctness of the results, and would likely require clarifications or reframing in the paper rather than a short rebuttal response.

---

> > > ### Author Response · Authors · 2026-04-01
> > >
> > > Thank you for the thoughtful follow-up. We are glad that our rebuttal clarified the main technical consistency concerns. We also appreciate that the remaining issue is now, as you put it, about the “scope and interpretation of the main contribution rather than correctness of the results."
> > >
> > > ---
> > >
> > > > “questions remain about how this translates to practical training behavior and whether the results demonstrate inevitability versus geometric tendency.”
> > >
> > > We appreciate this interpretive distinction. To clarify our position as precisely as possible, our intended claim is **not** that every finite-batch training run, optimizer, or architecture must realize the same endpoint in practice. Rather, the strongest result of the paper is an **objective-level geometric statement** in the analyzed regime: in the canonical large-batch, feature-normalized symmetric-InfoNCE setting, the induced intrinsic objective contains a persistent negative symmetric divergence term, so exact cross-modal marginal matching is **not** the generic behavior favored by the objective without additional control. In this sense, the paper identifies a **structural geometric mechanism** toward modality gap in that regime, where "structural" means induced by the learning objective itself rather than merely by initialization or training artifacts.
> > >
> > > At the same time, we agree that the phrase **"structural necessity"** may be read too strongly if interpreted as an unconditional statement about all practical training dynamics. That is not the intended scope. The realized magnitude and form of the gap in practice can depend on finite-batch noise, optimization, encoder parameterization, and architecture-level constraints. This is why the strongest theorem-level claims in the paper are objective-level, while inheritance to parametric models is treated separately under the expressivity conditions.
> > >
> > > To support relevance beyond the idealized setting, in the revised manuscript, we will add real-data evidence on MS-COCO across both CNN and Transformer CLIP-style models, together with a controlled same-category caption-corruption study. These additions are intended as evidence that the corresponding geometric signatures persist in realistic multimodal representations, while still keeping our theoretical claim scoped at the objective level.
> > >
> > > ---
> > >
> > > Thank you again for taking the time to engage so deeply with our work!

---

### Official Review · Reviewer_UNd5 · 2026-03-08

**Soundness:** 3
**Presentation:** 2
**Significance:** 3
**Originality:** 3
**Overall Recommendation:** 4
**Confidence:** 2

**Summary:**

This paper studies contrastive learning by looking at how representation distributions evolve on a fixed embedding manifold. It proves that in the large-batch limit, the stochastic InfoNCE loss tracks a deterministic energy landscape. For unimodal training, this landscape is strictly convex, and "uniformity" just acts as an entropic tie-breaker. For multimodal training like CLIP, the paper proves that a modality gap is structurally required by the symmetric objective. The loss creates a negative symmetric divergence that forces the modalities apart, showing the gap is not just an initialization artifact.

**Compliance With Llm Reviewing Policy:**

Affirmed.

**Final Justification:**

I maintain my original positive assessment of this work.

**Key Questions For Authors:**

Since the modality gap is caused directly by the symmetric InfoNCE objective , what specific changes to the loss function would you suggest to fix this issue in practical multimodal training?

**Limitations:**

Yes

**Strengths And Weaknesses:**

# Strengths

1. The paper gives a solid mathematical reason for the "modality gap" seen in multimodal models. It proves the gap is a structural feature of the loss function itself, rather than an artifact of bad initialization or sampling.
2. The work successfully connects the batch-wise InfoNCE loss that practitioners actually optimize to a clear geometric energy landscape.
3. The appendix includes helpful toy experiments that back up the math, clearly showing gradient consistency and the modality gap in simple, controlled setups.

# Weaknesses

1. **No real-world experiments**: The empirical validations only use synthetic data, such as 2D mixtures of Gaussians. It would be much more convincing if these geometric forces were shown in a standard SSL setup, like a small ResNet trained on CIFAR.
2. **No practical fix**: The paper diagnoses the modality gap as a structural necessity of symmetric InfoNCE , but it stops short of proposing or testing a new loss function to actually solve the problem.
3. Presentation can be difficult for non-theory readers. The measure-theoretic formulation and notation make the paper harder to follow, especially for readers more focused on empirical SSL research.

---

> ### Author Rebuttal · Authors · 2026-03-26
>
> We thank the reviewer for the positive assessment, especially for recognizing the paper’s objective-level explanation of modality gap, its bridge from batch-wise InfoNCE to a geometric landscape, and the value of the controlled validations. We also appreciate the three constructive suggestions regarding real-data evidence, practical fixes, and accessibility for non-theory readers. We address them below.
>
> ---
>
> **1. Added real-world experiments.**
>
> To address the request for evidence beyond synthetic validation, we add real-data experiments directly in the CLIP-like regime on MS-COCO, spanning both CNN and Transformer backbones, since our central claim concerns multimodal symmetric InfoNCE. We report Avg Recall@1 for retrieval, *energy distance* as a two-sample discrepancy between image/text marginals, and *centroid gap* as the distance between the mean image embedding and the mean text embedding.
>
> *(a) Pretrained OpenCLIP checkpoints on MS-COCO*
>
> | Model | Avg R@1 | Energy distance | Centroid gap |
> | --- | ---: | ---: | ---: |
> | PE-Core-L-14-336 | 0.650 | 0.599 | 0.854 |
> | PE-Core-B-16 | 0.601 | 0.545 | 0.808 |
> | ViT-bigG-14 | 0.588 | 0.288 | 0.598 |
> | ViT-H-14 | 0.574 | 0.375 | 0.697 |
> | ViT-L-14 | 0.463 | 0.697 | 0.898 |
> | RN50x16 | 0.457 | 0.502 | 0.761 |
> | RN101 | 0.399 | 0.443 | 0.670 |
> | ViT-B-16 | 0.397 | 0.688 | 0.871 |
> | RN50 | 0.385 | 0.627 | 0.849 |
>
> The key observation is that retrieval and cross-modal discrepancy are related but *not equivalent*: for example, PE-Core-L-14-336 achieves the best Avg R@1 (0.650) yet still shows a relatively large energy distance / centroid gap (0.599 / 0.854), whereas ViT-bigG-14 has slightly lower retrieval (0.588) but much smaller gap values (0.288 / 0.598). This is exactly the kind of real-data signature suggested by the theory.
>
> *(b) Controlled same-category caption corruption*
>
> We also run a controlled intervention that weakens exact image–text compatibility while preserving coarse semantics: captions are increasingly replaced by captions from other images sharing the same coarse category.
>
> | Model | Corruption p | Avg R@1 | Centroid gap |
> | --- | ---: | ---: | ---: |
> | RN50 | 0.00 | 0.398 | 0.845 |
> | RN50 | 0.25 | 0.379 | 1.001 |
> | RN50 | 0.50 | 0.350 | 1.040 |
> | RN50 | 0.75 | 0.309 | 1.064 |
> | RN50 | 1.00 | 0.226 | 1.081 |
> | ViT-B-16 | 0.00 | 0.484 | 0.851 |
> | ViT-B-16 | 0.25 | 0.469 | 1.032 |
> | ViT-B-16 | 0.50 | 0.446 | 1.082 |
> | ViT-B-16 | 0.75 | 0.405 | 1.117 |
> | ViT-B-16 | 1.00 | 0.297 | 1.148 |
>
> Here, the centroid gap reacts early: for ViT-B-16, increasing corruption from $p=0$ to $0.25$ changes Avg R@1 only from 0.484 to 0.469, while the centroid gap jumps from 0.851 to 1.032; RN50 shows the same monotone trend. This matches the theory’s prediction that cross-modal geometric separation can emerge before retrieval degrades severely. In the final version, we will incorporate these results with careful interpretation and plots.
>
> ---
>
> **2. What loss changes does the theory suggest?**
>
> Short answer: augment symmetric InfoNCE with an explicit distribution-level regularizer that penalizes cross-modal marginal mismatch, rather than relying only on stronger pairwise attraction. In our view, this is the most direct practical implication of the theory.
>
> The reason is that the mechanism identified in the paper is population-level: if the symmetric objective induces repulsion between modality marginals, then improving matched-pair similarity alone need not remove the gap. Therefore, the loss should not only reward alignment of positive pairs, but also explicitly control the discrepancy between the induced image and text embedding distributions.
>
> Concretely, the theory suggests three directions:
> - adding a regularization term to the loss that penalizes cross-modal marginal mismatch (e.g., via a symmetric-divergence or IPM-style penalty);
> - weakening the fully coupled cross-modal reference field, e.g., through a shared or tempered reference;
> - analyzing or designing the objective direction-by-direction, since heterogeneous modalities induce different directional potentials.
>
> We did not introduce a new loss in this work because we wanted to keep the central theoretical claim sharp, rather than add an insufficiently studied empirical fix. Designing a low-overhead surrogate for this discrepancy control is an important next step.
>
> ---
>
> **3. Improving accessibility for non-theory readers.**
>
> To improve accessibility for non-theory readers, in the final version, we will make the paper easier to follow by adding interpretation after the main formal statements, expanding the theorem-level intuition, clarifying assumptions more explicitly, and incorporating the MS-COCO evidence above to connect the intrinsic analysis back to practical multimodal training.
>
> ---
>
> Thank you again for these constructive suggestions! We hope these additions make the paper both stronger and more useful to the broader community.

---

> > ### Author Rebuttal · Reviewer_UNd5 · 2026-04-03
> >
> > Thank you to the authors for the thorough response. The clarifications provided for W1 and W2 successfully resolve my initial concerns. I acknowledge that addressing W3 requires more extensive revisions that cannot be fully demonstrated in a short rebuttal format. Trusting the authors to incorporate these presentation improvements in the final version, I am maintaining my initial positive assessment of this work.

---

> > > ### Author Response · Authors · 2026-04-03
> > >
> > > Thank you very much for reviewing our work! Your valuable suggestions have helped us substantially improve the clarity of the manuscript.

---

### Official Review · Reviewer_aN5w · 2026-03-12

**Soundness:** 3
**Presentation:** 2
**Significance:** 2
**Originality:** 2
**Overall Recommendation:** 2
**Confidence:** 3

**Summary:**

The paper presents a theoretical analysis of contrastive representation learning (InfoNCE) using a measure-theoretic framework that models learned representations as probability distributions on an embedding manifold. It shows that in the large-batch limit the stochastic InfoNCE objective converges to a deterministic energy functional over these distributions. In the unimodal case, the induced landscape is convex with a Gibbs-type equilibrium distribution. In the multimodal case, the objective contains a negative symmetric divergence between modalities, which the authors argue leads to a structural “modality gap” where modality distributions remain separated.

**Compliance With Llm Reviewing Policy:**

Affirmed.

**Key Questions For Authors:**

The paper is technically sophisticated but overly abstract and does not clearly produce new insights beyond existing theoretical interpretations of contrastive learning.

**Limitations:**

Yes.

**Strengths And Weaknesses:**

**Weaknesses**

-	Limited conceptual novelty : This paper intends to examine a broad area — the geometry of contrastive learning objectives — but the insights appear largely incremental relative to existing theory (Wang & Isola, 2020, Poole et al., 2019, Wang et al. 2020, Chen et al., 2021). Much of the analysis: 1) reformulates the alignment–uniformity decomposition in a more abstract language, 2) derives properties in limiting regimes (large batch, low temperature) already studied in prior theoretical works on contrastive. As a result, the new conceptual insight is relatively limited despite the heavy formal machinery.

-	The paper introduces substantial mathematical formalism (e.g., measure-theoretic analysis, variational functionals, and manifold assumptions) to study the geometry of contrastive learning objectives. While the framework is technically sophisticated, several of the resulting insights appear conceptually related to phenomena that have already been discussed in prior work. For example, the interpretation of contrastive learning in terms of equilibrium distributions and entropy–energy tradeoffs connects to earlier views of contrastive objectives as energy-based models or density-ratio estimators (e.g., Gutmann & Hyvärinen, 2010; Oord et al., 2018; Poole et al., 2019). Similarly, analyses of the population-level behavior of contrastive learning and the role of alignment versus dispersion have been explored in works such as Arora et al. (2019) and Wang & Isola (2020). In the multimodal setting, the observation that joint training may lead to persistent discrepancies between modality distributions is also related to the empirically observed “modality gap” discussed in Liang et al. (2022) and subsequent studies. In this sense, the main contribution of the paper appears to be the development of a geometric-measure formulation that unifies these perspectives within a common theoretical framework. While this perspective is interesting, it would be helpful for the paper to clarify more explicitly what additional insights or predictive implications this formalism provides beyond existing theoretical and empirical analyses. For instance, how do the authors would remove the modality grap given the current analysis ?

**Strengths**

-	Clear theoretical objective: The paper provides a principled theoretical analysis of contrastive learning by modeling representation learning as the evolution of probability measures on an embedding manifold.
-	Unified analysis of unimodal and multimodal contrastive learning: The framework studies both settings within the same formalism and highlights structural differences between them.
-	Theoretical explanation for the modality gap: The analysis suggests that multimodal contrastive learning may induce a persistent gap between modality distributions due to a negative symmetric divergence term in the objective.

---

> ### Author Rebuttal · Authors · 2026-03-26
>
> We thank the reviewer for recognizing the paper’s clear theoretical objective, its unified treatment of unimodal and multimodal contrastive learning, and its theoretical explanation of the modality gap. We also appreciate the central concern that the original manuscript did not make the paper’s novelty and concrete implications sufficiently clear. We address this directly below.
>
> ---
>
> **1. Novelty beyond unification.**
>
> We are grateful to the reviewer for recognizing our intrinsic geometric framework. Our intended claim, however, is stronger than unification alone: the formulation is **analytically substantive**, because it yields theorem-level consequences that are not directly established by the prior viewpoints cited in the review.
>
> - **Relative to critic-optimality / MI-style views** (e.g., Gutmann & Hyvärinen, 2010; Oord et al., 2018; Poole et al., 2019), our contribution is not another density-ratio interpretation of the optimal critic. We derive a large-batch **value-and-gradient bridge** from stochastic InfoNCE to deterministic population energies, formulated directly at the level of normalized representation measures.
>
> - **Relative to alignment–uniformity analyses** (e.g., Wang & Isola, 2020), our contribution is not merely a decomposition of the loss into alignment and dispersion terms. We show that, in the unimodal regime, the induced intrinsic free-energy geometry has a *unique Gibbs equilibrium*, which **sharpens “uniformity” into an entropic tie-breaker within the aligned basin** rather than a force acting over the entire manifold.
>
> - **Relative to empirical modality-gap papers** (e.g., Liang et al., 2022), our contribution is not simply to document that the gap exists, but to derive an **objective-level mechanism** for it: in the multimodal regime, the induced objective contains a persistent negative symmetric divergence term, which yields a geometric bifurcation under conditional heterogeneity.
>
> Thus, while unification is one part of the paper, the main contribution is that this intrinsic formulation makes these three consequences derivable within a single analytical path.
>
> ---
>
> **2. Predictive and practical implications.**
>
> The framework is intended to be mechanistic, not merely descriptive. In particular, regarding the reviewer’s question on how the modality gap should be mitigated, our answer is: pairwise alignment alone is insufficient; one needs explicit control of cross-modal discrepancy at the distribution level. From this perspective, the theory yields two predictive signatures: (i) in the unimodal regime, low temperature leads to concentration *within* an aligned basin, sharpening the usual alignment–uniformity intuition; (ii) in the multimodal regime, weakening cross-modal conditional compatibility enlarges *distribution-level* separation, and this need not be faithfully reflected by retrieval alone.
>
> A key empirical implication is therefore that multimodal evaluation should include simple probes of the modality marginals (e.g., centroid gap or energy distance), not only matched-pair retrieval. To support this point, we ran two additional real-data studies on MS-COCO, which will be included in the revised manuscript. Across pretrained OpenCLIP checkpoints, retrieval and cross-modal discrepancy are related but *not equivalent*: for example, PE-Core-L-14-336 has the best Avg Recall@1 (0.650) yet still shows a relatively large energy distance / centroid gap (0.599 / 0.854), whereas ViT-bigG-14 has slightly lower Avg Recall@1 (0.588) but much smaller gap values (0.288 / 0.598). In a controlled same-category caption-corruption experiment, the centroid gap reacts early: for ViT-B-16, increasing the corruption probability from $p=0$ to $0.25$ changes Avg Recall@1 only from 0.484 to 0.469, while the centroid gap jumps from 0.851 to 1.032. This matches the theory’s prediction that cross-modal geometric separation can emerge before retrieval degrades severely.
>
> ---
>
> **3. Why the presentation might have felt abstract.**
>
> We appreciate this point, and we believe it reflects a real tradeoff in the paper. The goal is to analyze *objective-induced geometry* in an intrinsic regime, rather than directly study architecture-specific parametric dynamics. The natural cost is a higher level of abstraction; the benefit is that it isolates the principal mechanism—large-batch population energy, Gibbs structure in the unimodal case, and divergence-driven bifurcation in the multimodal case—rather than only describing optimal critics or empirical retrieval behavior.
>
> We agree, however, that the original version could have made these takeaways clearer. In the revised manuscript, we will add a brief interpretation around the main formal statements to clarify the need for this intrinsic analysis and its implications, making the presentation more accessible to the broader community.
>
> ---
>
> We hope these clarifications make the paper’s novelty, predictive implications, and practical relevance more concrete.

---

> > ### Author Rebuttal · Reviewer_aN5w · 2026-04-03
> >
> > I thank the authors for their response. However, I disagree with them on two key points.
> > First, the only genuinely new prediction made by the theory concerns the modality gap in CLIP (let’s call a cat a cat). It does not provide new insights for the unimodal case, which was already well characterized geometrically by [Wang & Isola, 2020] in the limit N→∞.
> >
> > Second, the claim that “the natural cost is a higher level of abstraction” is not a sufficient justification for the 20 pages of proofs presented. These proofs combine functional analysis, measure theory, and optimization in a way that cannot realistically be carefully verified within the constraints of an ICML review process.
> >
> > It would be much clearer and more impactful to focus specifically on the modality gap phenomenon—which, in my view, is the primary point of interest shared by all reviewers—to validate the theory on toy or small-scale examples, and to study how the gap can be efficiently mitigated. I disagree with the authors’ claim that such a focus would not “keep the central theoretical claim sharp”; on the contrary, it would strengthen it.
> >
> > In its current form, the manuscript requires substantial revisions that cannot be adequately addressed within the rebuttal. Therefore, I am not changing my score.

---

> > > ### Author Response · Authors · 2026-04-04
> > >
> > > We thank the reviewer for the continued engagement. We respectfully disagree with two technical points that we believe are central to the current assessment.
> > >
> > > ---
> > >
> > > > **Claim 1: The unimodal case "was already well characterized geometrically by [1] in the limit $ N\to\infty$."**
> > >
> > > We would like to identify four precise technical distinctions between our unimodal analysis and [1]:
> > >
> > > **(1) Value convergence vs. gradient consistency.** Theorem 1 in [1] proves that the normalized contrastive loss converges in value to an alignment term plus a uniformity term as $N\to\infty$. No gradient is mentioned in their theorem statement or proof. Our Theorems 3.1 and 4.1 prove that both the value and the gradient of stochastic InfoNCE converge to those of the deterministic parametric energy, uniformly over the compact parameter space. This distinction is not stylistic: gradient consistency is what supports mechanistic conclusions about optimization trajectories. Without it, one can characterize optima, but not the descent direction that gradient-based training follows on the representation measures.
> > >
> > > **(2) Characterization of global optima vs. landscape geometry.**  [1] identifies conditions that globally optimal encoders must satisfy: perfect alignment and the uniform distribution on the hypersphere. Our Proposition 3.2 proves the intrinsic functional $\mathcal{F}_{\tau,U}(\rho)$ is strictly convex over the space of probability densities for all $\tau>0$, implying a unique Gibbs equilibrium $\rho^*(\mathbf{z})\propto \exp(-U(\mathbf{z})/\tau)$ at each $\tau$, tying back to the parametric model up to a KDE mismatch. Proposition 3.3 then derives ground-state concentration as a consequence of this landscape structure. [1] has no analogue of either result.
> > >
> > > **(3) The role of uniformity is different in the two papers.** [1] treats alignment and uniformity as two global effects, decomposes the loss additively, and proposes optimizing both as separate objectives. Our Remark 3.7 and Proposition 3.3 establish something qualitatively different: given the strict convexity and Gibbs structure, entropy acts as an *entropic tie-breaker within the alignment basin*, not a global competing force across the full manifold. This reinterpretation has concrete implications: interventions targeting global uniformity without respecting the alignment potential are not what the objective actually enforces.
> > >
> > > **(4) Generality and parametric inheritance.** [1] studies $\mathbb{S}^{m-1}$ with cosine similarity. Our framework covers general compact Riemannian manifolds with exponential kernels, and Corollaries 3.1 and 4.1 further show that, under sufficient encoder expressiveness, learned representations inherit the intrinsic geometric properties. Moreover, the unimodal intrinsic analysis is the analytical route that later enables the multimodal bifurcation results. Furthermore, the multimodal theorems themselves have no precursor in [1].
> > >
> > > ---
> > >
> > > > **Claim 2: "The proofs ... cannot realistically be carefully verified within the constraints of an ICML review process."**
> > >
> > > We note that the reviewer's concern is about verification cost rather than an identified error, and the two are distinct. Each proof is self-contained and relies on well-established mathematical tools standard in measure theory and statistical machine learning: regularity via compactness, concentration via the law of large numbers, and convergence via dominated convergence. The novelty lies not in the mathematical tools themselves, but in their utilization to derive the value-and-gradient bridge, the intrinsic Gibbs characterization of unimodal behavior, and the multimodal bifurcation mechanism, none of which appear in the cited prior work.
> > >
> > > ---
> > >
> > > **Summary.** The four distinctions above—gradient consistency vs. value convergence, landscape geometry vs. characterization of optima, entropy as tie-breaker vs. global competing force, and parametric inheritance vs. none—are each directly verifiable by comparing the two papers. These distinctions are **already stated in the manuscript itself (e.g., Introduction, pp. 1–2; Remark 3.1, p. 4; Remark 3.7, p. 6; see also Cor. 3.1)**. We believe they collectively establish that the unimodal contribution is not already contained in [1], and that the multimodal contribution has no precursor in the cited literature.
> > >
> > > ---
> > >
> > > [1] Wang T, Isola P. Understanding contrastive representation learning through alignment and uniformity on the hypersphere. ICML, 2020.

---

### Decision · Program_Chairs · 2026-04-30

**Decision:**

Accept (regular)

**Comment:**

This paper seeks to understand the geometry of the representation space in contrastive learning.

Reviewer aN5w argues that although the specific results of the paper are new, they are similar to existing and well-studied results in the literature. The paper does not add to this body of work beyond describing the questions and results in a more abstract language. These are valid concerns.

Reviewer UNd5 commented about the lack of real-world evaluation. The authors provided an interesting experiment in the rebuttal on cross-modal Info NCE objectives. Reviewer kKAz raised several technical points which have been clarified by the authors. The reviewer is on board with some of these clarifications but they have argued that the paper needs better framing to reflect the narrower scope.

Reviewer reUT had comments about studying these questions in finite-batch settings (which is somewhat out of scope of this paper). They also requested more experimental evidence, which was partially provided by the authors.

Based on these reviews, I would suggest that the authors minimize jargon and focus this paper on isolating new predictions regarding contrastive learning. In this context, the results on multi-modal contrastive learning are perhaps most interesting part of the paper. It is worthwhile to investigate new ways to describe old ideas, but it is important to argue what new insight this program leads to. For example, while distribution-level regularizers to control cross-modal mismatch are of course better than pairwise objectives, they are also much more expensive to implement. It would be interesting to characterize the benefits of the former in terms of the amount of samples required to reach the same performance. The experiments on MS-COCO with CNNs and transformers that were suggested by the authors could also improve the paper significantly.